# Learning Mixtures of Gaussians Using the DDPM Objective

**Kulin Shah**
UT Austin
kulinshah@utexas.edu

**Sitan Chen**
Harvard University
sitan@seas.harvard.edu

**Adam Klivans**
UT Austin
klivans@cs.utexas.edu

## Abstract

Recent works have shown that diffusion models can learn essentially any distribution provided one can perform score estimation. Yet it remains poorly understood under what settings score estimation is possible, let alone when practical gradient-based algorithms for this task can provably succeed.

In this work, we give the first provably efficient results along these lines for one of the most fundamental distribution families, Gaussian mixture models. We prove that gradient descent on the denoising diffusion probabilistic model (DDPM) objective can efficiently recover the ground truth parameters of the mixture model in the following two settings:

1. We show gradient descent with random initialization learns mixtures of two spherical Gaussians in $d$ dimensions with $1/\text{poly}(d)$-separated centers.
2. We show gradient descent with a warm start learns mixtures of $K$ spherical Gaussians with $\Omega(\sqrt{\log(\min(K, d))})$-separated centers.

A key ingredient in our proofs is a new connection between score-based methods and two other approaches to distribution learning, the expectation-maximization (EM) algorithm and spectral methods.

## 1 Introduction

In recent years diffusion models [SSDK+20, SDWMG15, SE19] have emerged as a powerful framework for generative modeling and now form the backbone of notable image generation systems like DALL·E 2 [RDN+22], Imagen [SCS+22], and Stable Diffusion [RBL+22]. At the heart of this framework is a reduction from *distribution learning* to *denoising* or *score estimation*. That is, in order to generate new samples from a data distribution $q$ given a collection of independent samples, it suffices to learn the score function, i.e., the gradient of the log-density of the data distribution when convolved with varying levels of noise (see Section 1.3). A popular and well-studied objective for score matching is the *denoising diffusion probabilistic model (DDPM) objective* due to [HJA20]. Optimizing this objective amounts to solving the following type of problem: given a noisy observation $\widetilde{x}$ of a sample $x$ from $q$, estimate the mean of the posterior distribution over $x$.

While a number of theoretical works [DBTHD21, BMR22, CLL22, DB22, LLT22, LWYL22, Pid22, WY22, CCL+23b, CDD23, LLT23, CCL+23a, LWCC23, BDD23] have established rigorous convergence guarantees for diffusion models under mild assumptions on the data distribution, these works assume the existence of an oracle for score estimation and leave open whether one can actually provably implement such an oracle for interesting families of data distributions. In practice, the algorithm of choice for score estimation is simply to train a student network via gradient descent (GD) to fit a set of examples $(x, \widetilde{x})$. We thus ask:

*Are there natural data distributions under which GD provably achieves accurate score estimation?*

37th Conference on Neural Information Processing Systems (NeurIPS 2023).

In this work, we consider the setting where $q$ is given by a *mixture of Gaussians*. Concretely, we assume that there exist centers $\mu_1^*, \ldots, \mu_K^* \in \mathbb{R}^d$ such that

$$q = \frac{1}{K} \sum_{i=1}^{K} \mathcal{N}(\mu_i^*, \mathrm{Id}).$$

We answer the above question in the affirmative for this class of distributions:

**Theorem 1** (Informal, see Theorems 7 and 13). *Gradient descent on the DDPM objective with random initialization efficiently learns the parameters of an unknown mixture of two spherical Gaussians with $1/poly(d)$-separated centers.*

**Theorem 2** (Informal, see Theorem 16). *When there is a warm start of the centers, gradient descent on the DDPM objective efficiently learns the parameters an unknown mixture of $K$ spherical Gaussians with $\Omega(\sqrt{\log(\min(K,d))})$-separated centers.*

The DDPM objective is described in Algorithm 1. The term "efficiently" above means that both the running time and sample complexity of our algorithm is polynomial in the dimension $d$, the inverse accuracy $1/\varepsilon$, and the number of components $K$. In the informal discussion, we often work with population gradients for simplicity, but in our proofs we show that empirical estimates of the gradient suffice (full details can be found in the Appendix).

---

**Algorithm 1:** GMMDENOISER($t, \{\mu_i^{(0)}\}_{i=1}^K, H$)

**Input:** Noise scale $t$, initialization $\{\mu_i^{(0)}\}_{i=1}^K$, number of gradient descent steps $H$

1 Initialize the parameters for the score estimate at $\theta_t^{(0)} = \{\mu_{i,t}^{(0)}\}_{i=1}^K$ (see Eq. (9) for how the estimate $s_\theta$ depends on the parameters $\theta$, and Eq. (8) for the definition of $\mu_{i,t}^{(0)}$)

2 Run gradient descent on the DDPM objective $L_t(s_{\theta_t})$ for $H$ steps where

$$L_t(s_{\theta_t}) = \mathbb{E}\left[\left\| s_{\theta_t}(X_t) + \frac{Z_t}{\sqrt{1 - \exp(-2t)}} \right\|^2 \right],$$

3 **return** $\theta_t^{(H)} = \{\mu_{i,t}^{(H)}\}_{i=1}^K$ where $\theta_t^{(H)}$ denotes the parameters after $H$ steps of GD.

---

We refer to Section 1.3 for a formal description of the quantities used in Algorithm 1. Note that there are by now a host of different algorithms for provably learning mixtures of Gaussians (see Section 1.1). For instance, it is already known that expectation-maximization (EM) achieves the quantitative guarantees of Theorems 1 and 2 [DTZ17, XHM16, KC20, SN21], and in fact even stronger guarantees are known via the method of moments. Unlike works based on the method of moments however, our algorithm is practical. And unlike works based on EM, it is based on an approach which is empirically successful for a wide range of realistic data distributions. Furthermore, as we discuss in Section 1.2, the analysis of Algorithm 1 leverages an intriguing and, to our knowledge, novel connection from score estimation to EM, as well as to another notable approach for learning mixture models, namely spectral methods. Roughly speaking, at large noise levels, the gradient updates in Algorithm 1 are essentially performing a type of power iteration, while at small noise levels, the gradient updates are performing the "M" step in the EM algorithm.

## 1.1 Related work

**Theory for diffusion models.** A number of works have given convergence guarantees for DDPMs and variants [DBTHD21, BMR22, CLL22, DB22, LLT22, LWYL22, Pid22, WY22, CCL$^+$23b, CDD23, LLT23, LWCC23, BDD23, CCL$^+$23a]. These results show that, given an oracle for accurate score estimation, diffusion models can learn essentially any distribution over $\mathbb{R}^d$ (e.g. [CCL$^+$23b, LLT23, CLL22] show this for arbitrary compactly supported distributions). Additionally, two recent works [EAMS22, MW23] have used Eldan's stochastic localization [Eld13, Eld20], which is a reparametrization in time and space of the reverse SDE for DDPMs, to give sampling algorithms for certain distributions arising in statistical physics. As we discuss next, these works are end-to-end in that they also give provable algorithms for score estimation via approximate message passing, though the statistical task they address is not distribution learning.

**Provable score estimation.**   There is a rich literature giving Bayes-optimal algorithms for various natural denoising problems via methods inspired by statistical physics, like approximate message passing (AMP) (e.g. [MV21, CFM21, BM11, Kab03, DMM09, DMM10]) and natural gradient descent (NGD) on the TAP free energy [CFM21, EAMS22, Cel22]. The abovementioned works [EAMS22, MW23] (see also [Cel22]) build on these techniques to give algorithms for the denoising problems that arise in their implementation of stochastic localization. These works on denoising via AMP or NGD are themselves part of a broader literature on variational inference, a suitable literature review would be beyond the scope of this work, see e.g. [BKM17, WJ+08, MM09].

We are not aware of any provable algorithms for score estimation explicitly in the context of *distribution learning*. That said, it may be possible to extract a distribution learning result from [EAMS22]. While their algorithm was for sampling from the Sherrington-Kirkpatrick (SK) model given the Hamiltonian rather than training examples as input, if one is instead given training examples drawn from the SK measure, then at sufficiently high temperature one can approximately recover the Hamiltonian [AG22]. In this case, a suitable modification [EAMS22] should be able to yield an algorithm for approximately generating fresh samples from the SK model given training examples.

**Learning mixtures of Gaussians.**   The literature on provable algorithms for learning Gaussian mixture models is vast, dating back to the pioneering work of Pearson [Pea94], and we cannot do justice to it here. We mention only works whose quantitative guarantees are closest in spirit to ours and refer to the introduction of [LL22] for a comprehensive overview of recent works in this direction. For mixtures of identity-covariance Gaussians in high dimensions, the strongest existing guarantee is a polynomial-time algorithm [LL22] for learning the centers as long as their pairwise separation slightly exceeds $\Omega(\sqrt{\log K})$ based on a sophisticated instantiation of method of moments inspired by the quasipolynomial-time algorithms of [DKS18, HL18, KSS18]. By the lower bound in [RV17], this is essentially optimal. In contrast, our Theorem 2 only applies given one initializes in a neighborhood of the true parameters of the mixture. We also note the exponential-time spectral algorithm of [SOAJ14] and quasipolynomial-time tensor-based algorithm of [DK20], which achieve *density estimation* even in the regime where the centers are arbitrarily closely spaced and learning the centers is information-theoretically impossible.

A separate line of work has investigated the "textbook" algorithm for learning Gaussian mixtures, namely the EM algorithm [BWY17, DS07, DTZ17, XHM16, YYS17, ZLS20, KC20, SN21]. Notably, for balanced mixtures of two Gaussians with the same covariance, [DTZ17] showed that finite-sample EM with random initialization converges exponentially quickly to the true centers. For mixtures of $K$ Gaussians with identity covariance, [KC20, SN21] showed that from an initialization sufficiently close to the true centers, finite-sample EM converges exponentially quickly to the true centers as long as their pairwise separation is $\Omega(\sqrt{\log K})$. In particular, [SN21] establish this local convergence as long as every center estimate is initialized at distance at most $\Delta/2$ away from the corresponding true center, where $\Delta$ is the minimum separation between any pair of true centers; this radius of convergence is provably best possible for EM.

Lastly, we note that there are many works giving parameter recovery algorithms mixtures of Gaussians with general mixing weights and covariances, all of which are based on method of moments [KMV10, HP15, Kan21, BS15, MV10, LM23, BDJ+22, DHKK20]. Unfortunately, for general mixtures of $K$ Gaussians, these algorithms run in time at least $d^{O(K)}$, and there is strong evidence [DKS17, BRST21] that this is unavoidable for computationally efficient algorithms.

## 1.2   Technical overview

We begin by describing in greater detail the algorithm we analyze in this work. For the sake of intuition, in this overview we will focus on the case of mixtures of two Gaussians ($K = 2$) where the centers are well-separated and symmetric about the origin, that is, the data distribution is given by

$$q = \frac{1}{2}\mathcal{N}(\mu^*, \mathrm{Id}) + \frac{1}{2}\mathcal{N}(-\mu^*, \mathrm{Id}). \tag{1}$$

At the end of the overview, we briefly discuss the key challenges for handling smaller separation and general $K$.

**Loss function, architecture of the score function and student network.**   The algorithmic task at the heart of score estimation is that of denoising. Formally, for some noise level $t > 0$, we are given

a noisy sample

$$X_t = \exp(-t)X_0 + \sqrt{1 - \exp(-2t)}Z_t\,,$$

where $X_0$ is a clean sample drawn from the data distribution $q$, and $Z_t \sim \mathcal{N}(0, \mathrm{Id})$. Conditioning on $X_t$ induces some posterior distribution over the noise $Z_t$, and our goal is to form an estimate $s$ for the mean of this posterior which achieves small error on average over the randomness of $X_0$ and $Z_t$. That is, we would like to minimize the *DDPM objective*, which up to rescaling is given by[1]

$$L_t(s) = \mathbb{E}_{X_0, Z_t}\|s(X_t) - Z_t\|^2\,.$$

As discussed in the introduction, the algorithm of choice for minimizing this objective in practice is gradient descent on some student network. To motivate our choice of architecture, note that when the data distribution is given by (1), the true minimizer of $L_t$ is, up to scaling,

$$\tanh(\langle \mu_t^*, x\rangle)\mu_t^* - x\,, \quad \text{where } \mu_t^* \triangleq \mu^* \exp(-t)\,. \tag{2}$$

See Appendix A for the derivation. Notably, Eq. (2) is exactly a two-layer neural network with $\tanh$ activation. As a result, we use the same architecture for our student network when running gradient descent. That is, given weights $\mu \in \mathbb{R}^d$, our student network is given by $s_\mu(x) \triangleq \tanh(\mu^\top x)\mu - x$. The exact gradient updates on $\mu$ are given in Lemma C.2.

As we discuss next, depending on whether the noise level $t$ is large or small, this update closely approximates the update in one of two well-studied algorithms for learning mixtures of Gaussians: power method and EM respectively.

**Learning mixtures of two Gaussians.** We first provide a brief overview of the analysis and then go into the details of the analysis. We start with mixtures of two Gaussians of the form (1) where $\|\mu^*\|$ is $\Omega(1)$. In this case, we analyze the following two-stage algorithm. We first use gradient descent on the DDPM objective with large $t$ starting from random initialization. We show that gradient descent in this "high noise" regime resembles a type of power iteration and gives $\mu$ that has a nontrivial correlation with $\mu_t^*$. Starting from this $\mu$, we then run gradient descent with small $t$. We show that the gradient descent in this "small noise" regime corresponds to the EM algorithm and converges exponentially quickly to the ground truth.

**Large noise level: connection to power iteration.** When $t$ is large, we show that gradient descent on the DDPM objective is closely approximated by power iteration. More precisely, in this regime, the negative gradient of $L_t(s_\mu)$ is well-approximated by

$$-\nabla_\mu L_t(s_\mu) \approx (2\mu_t^* \mu_t^{*\top} - r\mathrm{Id})\,\mu\,,$$

where $r$ is a scalar that depends on $\mu$ (See Lemma 8). So the result of a single gradient update with step size $\eta$ starting from $\mu$ is given by

$$\mu' \triangleq \mu - \eta\nabla_\mu L_t(s_\mu) \approx ((1 - \eta r)\,\mathrm{Id} + 2\eta\mu_t^*\mu_t^{*\top})\mu\,. \tag{3}$$

This shows us that each gradient step can be approximated by one step of power iteration (without normalization) on the matrix $(1 - \eta r)\,\mathrm{Id} + 2\eta\mu_t^*\mu_t^{*\top}$. It is know that running enough iterations of the latter from a random initialization will converge in angular distance to the top eigenvector, which in this case is given by $\mu_t^*$. This suggests that if we can keep the approximation error in (3) under control, then gradient descent on $\mu$ will also allow us to converge to a neighborhood of the ground truth. We implement this strategy in Lemma 10. Next, we argue that once we are in a neighborhood of the ground truth, we can run GD on the DDPM objective at *low noise level* to refine our estimate.

**Low noise level: connection to the EM algorithm.** When $t$ is small, we show that gradient descent on the DDPM objective is closely approximated by EM. Here, our analysis uses the fact that $\mu^*$ is sufficiently large and requires that we initialize $\mu$ to have sufficiently large correlation with the true direction $\mu_t^*$. We can achieve the latter using the large-$t$ analysis in the previous section.

Provided we have this, when $t$ is small it turns out that the negative gradient is well-approximated by

$$-\nabla_\mu L_t(s_\mu) \approx \mathbb{E}_{X \sim \mathcal{N}(\mu_t^*, \mathrm{Id})}[\tanh(\langle \mu, X\rangle)X] - \mu\,.$$

---

[1]The real DDPM objective is slightly different, see (5). The latter is what we actually consider in this paper, but this distinction is unimportant for the intuition in this overview.

Note that the expectation is precisely the "M"-step in the EM algorithm for learning mixtures of two Gaussians (see e.g. Eq. (2.2) of [DTZ17]). We conclude that a single gradient update with step size $\eta$ starting from $\mu$ is given by mixing the old weights $\mu$ with the result of the "M"-step in EM:

$$\mu' \triangleq \mu - \eta \nabla_\mu L_t(s_\mu) \approx (1-\eta)\mu + \eta \underbrace{\mathbb{E}_{X \sim \mathcal{N}(\mu_t^*, \mathrm{Id})}[\tanh(\langle \mu, X \rangle)X]}_{\text{"M" step in the EM algorithm}} \,.$$

[XHM16] and [DTZ17] showed that EM converges exponentially quickly to the ground truth $\mu_t^*$ from a warm start, and we leverage ingredients from their analysis to prove the same guarantee for gradient descent on the DDPM objective at small noise level $t$ (see Lemma 12).

**Extending to small separation.** Next, suppose we instead only assume that $\|\mu^*\|$ is $\Omega(1/\mathrm{poly}(d))$, i.e. the two components in the mixture may have small separation. The above analysis breaks down for the following reason: while it is always possible to show that gradient descent at large noise level converges in *angular distance* to the ground truth, if $\|\mu^*\|$ is small, then we cannot translate this to convergence in Euclidean distance.

We circumvent this as follows. Extending the connection between gradient descent at large $t$ and power iteration, we show that a similar analysis where we instead run *projected* gradient descent over the ball of radius $\|\mu^*\|$ yields a solution arbitrarily close to the ground truth, even without the EM step.[2] The projection step can be thought of as mimicking the normalization step in power iteration.

It might appear to the reader that this projected gradient-based approach is strictly superior to the two-stage algorithm described at the outset. However, in addition to obviating the need for a projection step when separation is large, our analysis for the two-stage algorithm has the advantage of giving much more favorable statistical rates. Indeed, we can show that the sample complexity of the two-stage algorithm has optimal dependence on the target error ($1/\varepsilon^2$), whereas we can only show a suboptimal dependence ($1/\varepsilon^8$) for the single-stage algorithm.

**Extending to general $K$.** The connection between gradient descent on the DDPM objective at small $t$ and the EM algorithm is sufficiently robust that for general $K$, our analysis for $K = 2$ can generalize once we replace the ingredients from [XHM16] and [DTZ17] with the analogous ingredients in existing analyses for EM with $K$ Gaussians. For the latter, it is known that if the centers of the Gaussians have separation $\Omega(\sqrt{\log \min(K, d)})$, then EM will converge from a warm start [KC20, SN21]. By carefully tracking the error in approximating the negative gradient with the "M"-step in EM, we are able to show that gradient descent on the DDPM objective at small $t$ achieves the same guarantee.

## 1.3 Preliminaries

**Diffusion models.** Throughout the paper, we use either $q$ or $q_0$ to denote the data distribution and $X$ or $X_0$ to denote the corresponding random variable on $\mathbb{R}^d$. The two main components in diffusion models are the *forward process* and the *reverse process*. The forward process transforms samples from the data distribution into noise, for instance via the *Ornstein-Uhlenbeck (OU) process*:

$$\mathrm{d}X_t = -X_t \, \mathrm{d}t + \sqrt{2} \, \mathrm{d}W_t \quad \text{with} \quad X_0 \sim q_0 \,,$$

where $(W_t)_{t \geq 0}$ is a standard Brownian motion in $\mathbb{R}^d$. We use $q_t$ to denote the law of the OU process at time $t$. Note that for $X_t \sim q_t$,

$$X_t = \exp(-t)X_0 + \sqrt{1 - \exp(-2t)}Z_t \quad \text{with} \quad X_0 \sim q_0, \ Z_t \sim \mathcal{N}(0, \mathrm{Id}) \,.$$

The reverse process then transforms noise into samples, thus performing generative modeling. Ideally, this could be achieved by running the following stochastic differential equation for some choice of terminal time $T$:

$$\mathrm{d}X_t^{\leftarrow} = \{X_t^{\leftarrow} + 2\nabla_x \ln q_{T-t}(X_t^{\leftarrow})\} \, \mathrm{d}t + \sqrt{2} \, \mathrm{d}W_t \quad \text{with} \quad X_0^{\leftarrow} \sim q_T \,,$$

where now $W_t$ is the reversed Brownian motion. In this reverse process, the iterate $X_t^{\leftarrow}$ is distributed acccording to $q_{T-t}$ for every $t \in [0, T]$, so that the final iterate $X_T^{\leftarrow}$ is distributed according to the

---

[2]Note that although $\mu^*$ is unknown, we can estimate its norm from samples.

data distribution $q_0$. The function $\nabla_x \ln q_t$ is called the *score function*, and because it depends on $q$ which is unknown, in practice one estimates it by minimizing the *score matching loss*

$$\min_{s_t} \quad \mathbb{E}_{X_t \sim q_t}[\|s_t(X_t) - \nabla_x \ln q_t(X_t)\|^2]. \tag{4}$$

A standard calculation (see e.g. Appendix A of [CCL$^+$23b]) shows that this is equivalent to minimizing the *DDPM objective* in which one wants to predict the noise $Z_t$ from the noisy observation $X_t$, i.e.

$$\min_{s_t} \quad L_t(s_t) = \mathbb{E}_{X_0, Z_t}\left[\left\|s_t(X_t) + \frac{Z_t}{\sqrt{1 - \exp(-2t)}}\right\|^2\right]. \tag{5}$$

While we have provided background on diffusion models for context, in this work we focus specifically on the optimization problem (5).

**Mixtures of Gaussians.** We consider the case of learning mixtures of $K$ equally weighted Gaussians:

$$q = q_0 = \frac{1}{K}\sum_{i=1}^{K} \mathcal{N}(\mu_i^*, \mathrm{Id}), \tag{6}$$

where $\mu_i^*$ denotes the mean of the $i^{\text{th}}$ Gaussian component. We define $\theta^* = \{\mu_1^*, \mu_2^* \ldots, \mu_K^*\}$. For the mixtures of two Gaussians, we can simplify the data distribution as

$$q = q_0 = \frac{1}{2}\mathcal{N}(\mu^*, \mathrm{Id}) + \frac{1}{2}\mathcal{N}(-\mu^*, \mathrm{Id}). \tag{7}$$

Note that distribution in Eq. (7) is equivalent to the distribution Eq. (6) with $K = 2$ because shifting the latter by its mean will give the former distribution, and furthermore the necessary shift can be estimated from samples. The following is immediate:

**Lemma 3.** *If $q_0$ is a mixture of $K$ Gaussians as in Eq. (6), then for any $t > 0$, $q_t$ is the mixture of $K$ Gaussians given by*

$$q_t = \frac{1}{K}\sum_{i=1}^{K} \mathcal{N}(\mu_{i,t}^*, \mathrm{Id}) \quad where \quad \mu_{i,t}^* \triangleq \mu_i^* \exp(-t). \tag{8}$$

See Appendix A for a proof of this fact. We can see that the means of $q_t$ get rescaled according to the noise level $t$. We also define $\theta_t^* = \{\mu_{1,t}^*, \mu_{2,t}^*, \ldots, \mu_{K,t}^*\}$.

**Lemma 4.** *The score function for distribution $q_t$, for any $t > 0$, is given by*

$$\nabla_x \ln q_t(x) = \sum_{i=1}^{K} w_{i,t}^*(x)\mu_{i,t}^* - x, \qquad where \qquad w_{i,t}^*(x) = \frac{\exp(-\|x - \mu_{i,t}^*\|^2/2)}{\sum_{j=1}^{K} \exp(-\|x - \mu_{j,t}^*\|^2/2)}.$$

*For a mixture of two Gaussians, the score function simplifies to*

$$\nabla_x \log q_t(x) = \tanh(\mu_t^{*\top} x)\mu_t^* - x, \qquad where \qquad \mu_t^* \triangleq \mu^* \exp(-t)$$

See Appendix A for the calculation.

Recall that $\nabla_x \log q_t(x)$ is the minimizer for the score-matching objective given in Eq. (4). Therefore, we parametrize our student network architecture similarly to the optimal score function. Our student architecture for mixtures of $K$ Gaussians is

$$s_{\theta_t}(x) = \sum_{i=1}^{K} w_{i,t}(x)\mu_{i,t} - x, \qquad where \qquad w_{i,t}(x) \triangleq \frac{\exp(-\|x - \mu_{i,t}\|^2/2)}{\sum_{j=1}^{K} \exp(-\|x - \mu_{j,t}\|^2/2)} \tag{9}$$

$$\mu_{i,t} \triangleq \mu_i \exp(-t).$$

where $\theta_t = \{\mu_{1,t}, \mu_{2,t}, \ldots, \mu_{K,t}\}$ denotes the set of parameters at the noise scale $t$. For mixtures of two Gaussians, we simplify the student architecture as follows:

$$s_{\theta_t}(x) = \tanh(\mu_t^\top x)\mu_t - x, \quad where \quad \mu_t \triangleq \mu \exp(-t).$$

As $\theta_t$ only depends on $\mu_t$ in the case of mixtures of two Gaussians, we simplify the notation of the score function from $s_{\theta_t}(x)$ to $s_{\mu_t}(x)$ in that case. We use $\hat{\mu}_t$ and $\hat{\mu}_t^*$ to denote the unit vector along the direction of $\mu_t$ and $\mu_t^*$ respectively. Note that we often use $\mu_t$ (or $\theta_t$) to denote the current iterate of gradient descent on the DDPM objective and $\mu_t'$ to denote the iterate after taking a gradient descent step from $\mu_t$.

**Expectation-Maximization (EM) algorithm.** The EM algorithm is composed of two steps: the E-step and the M-step. For mixtures of Gaussians, the E-step computes the expected log-likelihood based on the current mean parameters and the M-step maximizes this expectation to find a new estimate of the parameters.

**Fact 5** (See e.g., [DTZ17, YYS17, KC20] for more details). *When $X$ is the mixture of $K$ Gaussian and $\{\mu_1, \mu_2, \ldots, \mu_K\}$ are current estimates of the means, the population EM update for all $i \in \{1, 2, \ldots, K\}$ is given by*

$$\mu_i' = \frac{\mathbb{E}_X[w_i(X)X]}{\mathbb{E}_X[w_i(X)]}, \quad where \ \ w_i(X) = \frac{\exp(-\|X - \mu_i\|^2/2)}{\sum_{j=1}^K \exp(-\|X - \mu_j\|^2/2)}.$$

*The EM update for mixtures of two Gaussians given in Eq. (7) simplifies to*

$$\mu' = \mathbb{E}_{X \sim \mathcal{N}(\mu^*, \mathrm{Id})}[\tanh(\mu^\top X)X].$$

An analogous version of the EM algorithm, called the gradient EM algorithm, takes a gradient step in the direction of the M-step instead of optimizing the objective in the M-step fully.

**Fact 6** (See e.g., [YYS17, SN21] for more details). *For all $i \in \{1, 2, \ldots, K\}$, the gradient EM-update for mixtures of $K$ Gaussian is given by*

$$\mu_i' = \mu_i + \eta \, \mathbb{E}_X[w_i(X)(X - \mu_i)],$$

*where $\eta$ is the learning rate.*

## 2 Warmup: mixtures of two Gaussians with constant separation

In this section, we formally state our result for learning mixtures of two Gaussians with constant separation. This case highlights the main proof techniques, namely viewing gradient descent on the DDPM objective as power iteration and as the EM algorithm.

### 2.1 Result and algorithm

**Theorem 7.** *There is an absolute constant $c > 0$ such that the following holds. Suppose a mixture of two Gaussians with the mean parameter $\mu^*$ satisfies $\|\mu^*\| > c$. Then, for any $\varepsilon > 0$, there is a procedure that calls Algorithm 1 at two different noise scales $t$ and outputs $\tilde{\mu}$ such that $\|\tilde{\mu} - \mu^*\| \leq \varepsilon$ with high probability. Moreover, the algorithm has time and sample complexity $\mathrm{poly}(d)/\varepsilon^2$ (see Theorem C.1 for more precise quantitative bounds).*

**Algorithm.** The algorithm has two stages. In the first stage we run gradient descent on the DDPM objective described in Algorithm 1 from a random Gaussian initialization and noise scale $t_1$ for a fixed number of iterations $H$ where $t_1 = \Theta(\log d)$ ("high noise") and $H = \mathrm{poly}(d, 1/\varepsilon)$. In the second stage, the procedure uses the output of the first step as initialization and runs Algorithm 1 at a "low noise" scale of $t_2 = \Theta(1)$.

### 2.2 Proof outline of Theorem 7

We provide a proof sketch of correctness of the above algorithm and summarize the main technical lemmas here. All proofs of the following lemmas can be found in Appendix C.

**Part I: Analysis of high noise regime and connection to power iteration.** We show that in the large noise regime, the negative gradient $-\nabla L_t(s_t)$ is well-approximated by $2\mu_t^* \mu_t^{*\top} \mu_t - 3\|\mu_t\|^2 \mu_t$. Recall that this result is the key to showing the resemblance between gradient descent and power iteration. Concretely, we show the following lemma:

**Lemma 8** (See Lemma C.3 for more details). *For $t = \Theta(\log d)$, the gradient descent update on the DDPM objective $L_t(s_t)$ can be approximated with $2\mu_t^* \mu_t^{*\top} \mu_t - 3\|\mu_t\|^2 \mu_t$:*

$$\left\| \left( -\nabla L_t(s_t) \right) - \left( 2\mu_t^* \mu_t^{*\top} \mu_t - 3\|\mu_t\|^2 \mu_t \right) \right\| \leq \mathrm{poly}(1/d).$$

From Lemma 8, it immediately follows that $\mu' t$, the result of taking a single gradient step starting from $\mu_t$, is well-approximated by the result of taking a single step of power iteration for a matrix whose leading eigenvector is $\mu_t^*$:

$$\mu_t' = \mu_t - \eta \nabla L_t(s_\mu) \approx (\mathrm{Id}(1 - 3\eta\|\mu_t\|^2) + 2\mu_t^*\mu_t^{*\top})\mu_t .$$

The second key element is to show that as a consequence of the above power iteration update, the gradient descent converges in *angular distance* to the leading eigenvector. Concretely, we show the following lemma:

**Lemma 9** (Informal, see Lemma C.5 for more details)**.** *Suppose $\mu_t'$ is the iterate after one step of gradient descent on the DDPM objective from $\mu_t$. Denote the angle between $\mu_t$ and $\mu_t^*$ to be $\theta$ and between $\mu_t'$ and $\mu_t^*$ to be $\theta'$. In this case, we show that*

$$\tan\theta' = \max\left(\kappa_1 \tan\theta, \kappa_2\right),$$

*where $\kappa_1 < 1$ and $\kappa_2 \leq 1/\mathrm{poly}(d)$.*

Note $\tan\theta' < \tan\theta$ implies that $\theta' < \theta$ or equivalently $\langle\hat{\mu}_t', \hat{\mu}_t^*\rangle > \langle\hat{\mu}_t, \hat{\mu}_t^*\rangle$. Thus, the above lemma shows that by taking a gradient step in the DDPM objective, the angle between $\mu_t$ and $\mu_t^*$ decreases. By iterating this, we obtain the following lemma:

**Lemma 10** (Informal, see Lemma C.6 for more details)**.** *Running gradient descent from a random initialization on the DDPM objective $L_t(s_\mu)$ for $t = O(\log d)$ gives $\mu_t$ for which $\langle\hat{\mu}_t, \hat{\mu}_t^*\rangle$ is $\Omega(1)$.*

Note that we cannot keep running gradient descent at this high noise scale and hope to achieve $\mu$ such that $\|\mu - \mu^*\|$ is $O(\varepsilon)$. This is because Lemma 9 can only guarantee that the angle between $\mu_t$ and $\mu_t^*$ is $O(\varepsilon)$, but this does not imply $\|\mu - \mu^*\|$ is $O(\varepsilon)$. Instead, as described in Part II, we will proceed with a smaller noise scale.

**Part II: Analysis of low noise regime and connection to EM.**  In the low noise regime, we run Algorithm 1 using the output from Part I as our initialization. Our analysis here shows that whenever the initialization $\mu_t$ satisfies the condition of $\langle\hat{\mu}_t, \hat{\mu}_t^*\rangle$ being $\Omega(1)$, $\|\mu_t - \mu_t^*\|$ contracts after every gradient step. To start with, we show that the result of a *population* gradient step on the DDPM objective $L_t(s_\mu)$ results in the following:

$$\mu_t' = (1 - \eta)\mu_t + \eta\,\mathbb{E}_{x\sim\mathcal{N}(\mu_t^*,\mathrm{Id})}[\tanh(\mu_t^\top x)x] + \eta G(\mu_t, \mu_t^*),$$

where $\mu_t'$ is the parameter after a gradient step, $\eta$ is the learning rate, and function $G$ is given by

$$G(\mu, \mu^*) = \mathbb{E}_{x\sim\mathcal{N}(\mu^*,\mathrm{Id})}[-\frac{1}{2}\tanh''(\mu^\top x)\|\mu\|^2\,x + \tanh'(\mu^\top x)\mu^\top xx - \tanh'(\mu^\top x)\mu].$$

Note we use the population gradient here only for simplicity; in the Appendix we show that empirical estimates of the gradient suffice. After some calculation, we can show that

$$\|\mu_t' - \mu_t^*\| \leq (1 - \eta)\|\mu_t - \mu_t^*\| + \eta\|\mathbb{E}_{x\sim\mathcal{N}(\mu_t^*,\mathrm{Id})}[\tanh(\mu_t^\top x)x] - \mu_t^*\| + \eta\|G(\mu_t, \mu_t^*)\| . \quad (10)$$

Using Fact 5, we know that $\mathbb{E}_{x\sim\mathcal{N}(\mu_t^*,\mathrm{Id})}[\tanh(\mu_t^\top x)x]$ is precisely the result of one step of EM starting from $\mu_t$, and it is known [DTZ17] that the EM update contracts the distance between $\mu_t$ and $\mu_t^*$ as follows:

$$\|\mathbb{E}_{x\sim\mathcal{N}(\mu_t^*,\mathrm{Id})}[\tanh(\mu_t^\top x)x] - \mu_t^*\| \leq \lambda_1\|\mu_t - \mu_t^*\| \quad \text{for some } \lambda_1 < 1 \quad (11)$$

It remains to control the second term in Eq. (10), for which we prove the following:

**Lemma 11** (Informal, see Lemma C.9 for more details)**.** *When $\|\mu^*\| = \Omega(1)$ and the noise scale $t = \Theta(1)$, then for every $\mu$ with $\langle\hat{\mu}, \hat{\mu}^*\rangle$ being $\Omega(1)$, the following inequality holds:*

$$\|G(\mu_t, \mu_t*)\| \leq \lambda_2\|\mu_t - \mu_t^*\| \quad \text{for some } \lambda_2 < 1 .$$

Combining Eq. (11) and Lemma 11 with Eq. (10), we have

$$\|\mu_t' - \mu_t^*\| \leq (1 - \eta(1 - \lambda_1 - \lambda_2))\|\mu_t - \mu_t^*\| . \quad (12)$$

We can set parameters to ensure that $\lambda_1 + \lambda_2 < 1$ and therefore that $\|\mu_t - \mu_t^*\|$ contracts with each gradient step. Applying Lemma 11 and Eq. (12), we obtain the following lemma summarizing the behavior of gradient descent on the DDPM objective in the low noise regime.

**Lemma 12** (Informal). *For any $\varepsilon > 0$ and for the noise scale $t = \Theta(1)$, starting from an initialization $\mu_t$ for which $\langle \hat{\mu}_t, \hat{\mu}_t^* \rangle = \Omega(1)$, running gradient descent on the DDPM objective $L_t(s_\mu)$ will give us mean parameter $\tilde{\mu}$ such that $\|\tilde{\mu} - \mu^*\| \leq O(\varepsilon)$.*

Combining Lemma 10 and Lemma 12, we obtain our first main result, Theorem 7, for learning mixtures of two Gaussians with constant separation. For the full technical details, see Appendix C.

## 3 Extensions: small separation and more components

### 3.1 Mixtures of two Gaussians with small separation

In this section, we briefly sketch how the ideas from Section 2 can be extended to give our second main result, namely on learning mixtures of two Gaussians even with *small separation*. We defer the full technical details to Appendix D.

**Theorem 13.** *Suppose a mixture of two Gaussians has mean parameter $\mu^*$ that satisfies $\|\mu^*\| = \Omega(\frac{1}{\text{poly}(d)})$. Then, for any $\varepsilon > 0$, there exists a modification of Algorithm 1 that provides an estimate $\mu$ such that $\|\mu - \mu^*\| \leq O(\varepsilon)$ with high probability. Moreover, the algorithm has time and sample complexity $\text{poly}(d)/\varepsilon^8$ (see Theorem D.1 for more precise quantitative bounds).*

**Algorithm modification.** The algorithm that we analyze runs *projected* gradient descent on the DDPM objective but only in the high noise scale regime where $t = O(\log d)$. At each step, we project the iterate $\mu$ to the ball of radius $R$, where $R$ is an empirical estimate for $\|\mu^*\|$ obtained by drawing samples $x_1, \ldots, x_n$ from the data distribution and forming $R \triangleq (\frac{1}{n} \sum_{i=1}^{n} \|x_i\|^2 - d)^{1/2}$.

**Proof sketch.** Lemma 9 and Lemma 10 apply even when the components of the mixture have small separation, and they show that running gradient descent on the DDPM objective results in $\mu_t$ and $\mu_t^*$ being $O(1)$ close in angular distance. Although our analysis can be extended to show that gradient descent can achieve $O(\varepsilon)$ angular distance, this does not guarantee that $\|\mu_t - \mu_t^*\|$ is $O(\varepsilon)$. If in addition to being $O(\varepsilon)$ close in angular distance, we also have that $\|\mu_t\| \approx \|\mu_t^*\|$, then it is easy to see that $\|\mu_t - \mu_t^*\|$ is indeed $O(\varepsilon)$.

Observe that if $R$ is approximately equal to $\|\mu_t^*\|$, then the projection step in our algorithm ensures that our final estimate $\mu_t$ satisfies this additional condition of $\|\mu_t\| \approx \|\mu_t^*\|$. It is not hard to show that $R^2$ is an unbiased estimate of $\|\mu_t^*\|^2$, so standard concentration shows that taking $n = \text{poly}(d, \frac{1}{\varepsilon})$ suffices to ensure that $R$ is sufficiently close to $\|\mu_t^*\|$.

### 3.2 Mixtures of $K$ Gaussians, from a warm start

In this section, we state our third main result, namely for learning mixtures of $K$ Gaussians given by Eq. (6) from a warm start, and provide an overview of how the ideas from Section 2 can be extended to obtain this result.

**Assumption 14.** *(Separation) For a mixture of $K$ Gaussians given by Eq. (6), for every pair of components $i, j \in \{1, 2, \ldots, K\}$ with $i \neq j$, we assume that the separation between their means $\|\mu_i^* - \mu_j^*\| \geq C\sqrt{\log(\min(K, d))}$ for sufficiently large absolute constant $C > 0$.*

**Assumption 15.** *(Initialization) For each component $i \in \{1, 2, \ldots, K\}$, we have an initialization $\mu_i^{(0)}$ with the property that $\|\mu_i^{(0)} - \mu_i^*\| \leq C'\sqrt{\log(\min(K, d))}$ for sufficiently small absolute constant $C' > 0$.*

**Theorem 16.** *Suppose a mixture of $K$ Gaussians satisfies Assumption 14. Then, for any $\varepsilon = \Theta(1/\text{poly}(d))$, running gradient descent on the DDPM objective (Algorithm 1) at low noise scale $t = O(1)$ and with initialization satisfying Assumption 15 results in mean parameters $\{\mu_i\}_{i=1}^{K}$ such that with high probability, the mean parameters satisfy $\|\mu_i - \mu_i^*\| \leq O(\varepsilon)$ for each $i \in \{1, 2, \ldots, K\}$. Additionally, the runtime and sample complexity of the algorithm is $\text{poly}(d, 1/\varepsilon)$ (see Theorem E.1 for more precise quantitative bounds).*

We provide a brief overview of the proof here. The full proof can be found in Appendix E.

**Proof sketch.** For learning mixtures of two Gaussians, we have already established the connection between gradient descent on the DDPM objective and the EM algorithm. For mixtures of $K$ Gaussians, however, in a local neighborhood around the ground truth parameters $\theta^*$, we show an equivalence between *gradient* EM (recall gradient EM performs one-step of gradient descent on the "M" step objective) and gradient descent on the DDPM objective. In particular, our main technical lemma (Lemma E.4) shows that for noise scale $t = \Theta(1)$ and for any $\mu_i$ that satisfies $\|\mu_i - \mu_i^*\| \leq O(\sqrt{\log(\min(K, d))})$, we have

$$-\nabla_{\mu_{i,t}} L_t(s_{\theta_t}) \approx \mathbb{E}_{X_t}[w_{i,t}(X_t)(X_t - \mu_{i,t})].$$

Therefore, the iterate $\mu'_{i,t}$ resulting from a single gradient step on the DDPM objective $L_t(s_{\theta_t})$ with learning rate $\eta$ is given by

$$\mu'_{1,t} = \mu_{1,t} - \eta \nabla_{\mu_{1,t}} L_t(s_{\theta_t}) \approx \mu_{1,t} + \eta \, \mathbb{E}_{X_t}[w_{1,t}(X_t)(X_t - \mu_{1,t})]. \tag{13}$$

Comparing Fact 6 with Eq. (13), we see the correspondence in this regime between gradient descent on the DDPM objective to gradient EM. Using this connection and an existing local convergence guarantee from the gradient EM literature [SN21, KC20], we obtain our main theorem for mixtures of $K$ Gaussians. Full details can be found in Appendix E.

## Acknowledgments

SC would like to thank Sinho Chewi, Khashayar Gatmiry, Frederic Koehler, and Holden Lee for enlightening discussions on sampling and score estimation. KS and AK are supported by the NSF AI Institute for Foundations of Machine Learning (IFML). SC is supported by NSF Award 2103300.

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
