**Roadmap.** In Appendix A, we provide proofs of some simple lemmas from Section 1.3 and some basic inequalities. In Appendix B we give additional notation and preliminaries. In Appendix C, we provide the proof details for Theorem 7, our result on learning mixtures of two Gaussians with constant separation. In Appendix D, we extend this analysis to give a proof of Theorem 13, our result on learning mixtures of two Gaussians with small separation. In Appendix E, we provide the proof details for Theorem 16, our result on learning mixtures of $K$ Gaussians. In Appendix F, we give further deferred proofs. Finally, in Appendix G, we provide experiments to confirm our theoretical results.

# A    Proofs from Section 1.3

## A.1    $X_t$ is a mixture of Gaussians

*Proof of Lemma 3.* Suppose $X_0$ is mixture of $K$ Gaussians with density function given by

$$q_0 = \frac{1}{K} \sum_{i=1}^{K} \mathcal{N}(\mu_{i,0}^*, \mathrm{Id})$$

We know that $X_t = \exp(-t)X_0 + \sqrt{1 - \exp(-2t)}Z_t$ where $Z_t \sim \mathcal{N}(0, \mathrm{Id})$. Then, by change of variable of probability density, we have

$$\text{pdf of } \exp(-t)X_0 = \frac{1}{K} \sum_{i=1}^{K} \mathcal{N}(\mu_{i,0}^* \exp(-t), \exp(-2t) \cdot \mathrm{Id})$$

$$\text{pdf of } \sqrt{1 - \exp(-2t)}Z_t = \mathcal{N}(0, (1 - \exp(-2t)) \cdot \mathrm{Id}).$$

Combining these, we have

$$q_t(X_t) = \frac{1}{K} \sum_{i=1}^{K} \mathcal{N}(\mu_{i,t}^*, I) \qquad \text{where} \qquad \mu_{i,t}^* = \mu_{i,0}^* \exp(-t),$$

as claimed. $\qquad\qquad\qquad\qquad\qquad\qquad\qquad\qquad\qquad\qquad\qquad\qquad\qquad\qquad\qquad\qquad\qquad$ □

## A.2    Derivation of score function

*Proof of Lemma 4.* For mixtures of $K$ Gaussians in the form of Eq. (6), the score function at time $t$ is given by

$$\nabla \log q_t(x) = -\frac{\sum_{i=1}^{K} e^{-\frac{\|x - \mu_{i,t}^*\|^2}{2}} (x - \mu_{i,t}^*)}{\sum_{j=1}^{K} e^{-\frac{\|x - \mu_{j,t}^*\|^2}{2}}}$$

$$= \sum_{i=1}^{K} w_{i,t}^*(x) \mu_{i,t}^* - x \quad \text{where} \quad w_{i,t}^*(x) = \frac{e^{-\frac{\|x - \mu_{i,t}^*\|^2}{2}}}{\sum_{j=1}^{K} e^{-\frac{\|x - \mu_{j,t}^*\|^2}{2}}}.$$

For mixtures of two Gaussians in the form of Eq. (7), the score function is given by

$$\nabla \log q_t(x) = w_{1,t}^*(x) \mu_{1,t}^* + w_{2,t}^*(x) \mu_{2,t}^* - x$$
$$= w_{1,t}^*(x) \mu^* - (1 - w_{1,t}^*(x)) \mu^* - x$$
$$= (2 w_{1,t}^*(x) - 1) \mu^* - x \qquad\qquad\qquad\qquad\qquad (\text{A.1})$$

By simplifying $w_{1,t}^*(x)$, we obtain

$$w_{1,t}^*(x) = \frac{1}{1 + \exp\left(\frac{\|x - \mu^*\|^2}{2} - \frac{\|x + \mu^*\|^2}{2}\right)}$$
$$= \frac{1}{1 + \exp(-2\mu^{*\top} x)}$$
$$= \sigma(2\mu^{*\top} x) \qquad\qquad\qquad\qquad\qquad\qquad\qquad (\text{A.2})$$

where $\sigma(\cdot)$ denotes the sigmoid function. Using Eq. (A.2) in Eq. (A.1), we obtain

$$\nabla \log q_t(x) = \tanh(\mu^{*\top} x)\mu^* - x.$$

$\square$

## B    Additional notations and preliminaries

In this section, we provide additional notations and preliminaries for the proofs to follow. Recall that we use $L_t(s_{\theta_t})$ to denote the population denoising loss at noise scale $t$.

$$L_t(s_{\theta_t}) = \mathbb{E}\left[\left\|s_{\theta_t}(X_t) + \frac{Z_t}{\sqrt{1 - \exp(-2t)}}\right\|^2\right].$$

We use $L_t(s_{\theta_t}(x_0, z_t))$ to denote the denoising loss at noise scale $t$ on a sample $x_0$ from the data distribution and $z_t$ from the standard Gaussian distribution:

$$L_t(s_{\theta_t}(x_0, z_t)) = \left\|s_{\theta_t}(x_t) + \frac{z_t}{\sqrt{1 - \exp(-2t)}}\right\|^2,$$

where $x_t = \exp(-t)x_0 + \sqrt{1 - \exp(-2t)}z_t$. We use $\alpha_t$ as shorthand notation for $\exp(-t)$ and $\beta_t$ as shorthand notation for $\sqrt{1 - \exp(-2t)}$.

For mixtures of two Gaussians, we use $B$ to denote the upper bound on $\|\mu^*\|^2$, that is,

$$\|\mu^*\|^2 \leq B.$$

Throughout, we assume that $B = \text{poly}(d)$.

For any vector $v$, we use $\hat{v}$ to denote the unit vector along the direction of $v$. For a vector $v$, we use $[v]_i$ to denote the $i^{th}$ coordinate of $v$. Similarly, for a matrix $X$, we use $[X]_i$ to denote the $i^{th}$ row of the matrix. For any positive integer $n$, we use $[n]$ to denote the set $\{1, 2, \ldots, n\}$. We use $\mathcal{N}(\mu, \sigma^2 \cdot \text{Id})$ to denote the standard Gaussian with mean $\mu$ and covariance $\sigma^2 \cdot \text{Id}$. Sometimes, we use a shorter notation $\mathcal{N}_\mu$ to denote $\mathcal{N}(\mu, \text{Id})$. For any two quantities $X$ and $Y$ that are both implicitly functions of some parameter $a$ over $\mathbb{R}_{\geq 0}$, we use the shorthand $X \lesssim Y$ and $X = O(Y)$ interchangeably to denote that there exists absolute constant $C > 0$ such that for all $a$ sufficiently large, $X(a) \leq CY(a)$. We also use the shorthand $X \gtrsim Y$ and $X = \Omega(Y)$, defined in the obvious way.

Finally, we will use the following standard bounds.

**Lemma B.1** (Sub-Gaussian norm, see e.g. [Ver])**.** *The sub-Gaussian norm of a random variable $X \in \mathbb{R}$, denoted by $\|X\|_{\psi_2}$ is defined as*

$$\|X\|_{\psi_2} = \inf\{t > 0 \ : \ \mathbb{E}[\exp(X^2/t^2)] \leq 2\}.$$

*The sub-Gaussian norm has the following properties:*

1. *(Bounded): Any bounded random variable $X$ (i.e., there is a finite $A$ for which $|X| \leq A$ with probability 1) is sub-Gaussian:*

$$\|X\|_{\psi_2} \leq \frac{A}{\sqrt{\ln 2}}$$

2. *(Centering): If $X$ is a sub-Gaussian random variable, then $X - \mathbb{E}[X]$ is also a sub-Gaussian random variable. Specifically, the following holds for some absolute constant $C$.*

$$\|X - \mathbb{E}[X]\|_{\psi_2} \leq C\|X\|_{\psi_2}$$

3. *(Moment generating function bound): If $X$ is a sub-Gaussian random variable with $E[X] = 0$, then*
$$\mathbb{E}[\exp(\lambda X)] \leq \exp(C\lambda^2\|X\|_{\psi_2}^2) \quad \text{for all } \lambda \in \mathbb{R},$$
*where $C$ is some absolute constant.*

4. *(Sum of sub-Gaussian random variables): If $X_1$ and $X_2$ are mean zero sub-Gaussian random variables, then*

$$\|X_1 + X_2\|_{\psi_2} \leq \|X_1\|_{\psi_2} + \|X_2\|_{\psi_2}.$$

5. *(Product with a bounded random variable): If $X$ is a sub-Gaussian random variable and $Y$ is a bounded random variable $Y \in [0,1]$, then*

$$\|XY\|_{\psi_2} \leq \|X\|_{\psi_2}.$$

**Lemma B.2** (Sub-exponential norm, see e.g. [Ver]). *The sub-exponential norm of a random variable $X \in \mathbb{R}$, denoted by $\|X\|_{\psi_1}$ is defined as*

$$\|X\|_{\psi_1} = \inf\{t > 0 \; : \; \mathbb{E}[\exp(|X|/t)] \leq 2\}.$$

*The sub-exponential norm has the following properties:*

1. *(Sum of sub-exponential distributions): If $X_1$ and $X_2$ are mean-zero sub-exponential random variables, then $X_1 + X_2$ is also a mean-zero sub-exponential variable. Specifically,*

$$\|X_1 + X_2\|_{\psi_1} \leq \sqrt{2}(\|X_1\|_{\psi_1} + \|X_2\|_{\psi_1}).$$

2. *(Centering) If $X$ is a sub-exponential random variable, then $X - \mathbb{E}[X]$ is sub-exponential with*

$$\|X - \mathbb{E}[X]\|_{\psi_1} \leq C\|X\|_{\psi_1},$$

*where $C$ is some absolute constant.*

*Proof.* The proof follows from following the equivalent definition of a sub-exponential random variable: If any random variable $X$ satisfies

$$\mathbb{E}[\exp(\lambda X)] \leq \exp(C\|X\|_{\psi_1}^2 \lambda^2) \; \text{ for all } \lambda \text{ such that } |\lambda| \leq \frac{1}{C\|X\|_{\psi_1}^2},$$

for some constant $C$, then $X$ is sub-exponential random variable with sub-exponential norm $\|X\|_{\psi_1}$. Then, for any $|\lambda| \leq \frac{1}{2C \max(\|X_1\|_{\psi_1}^2, \|X_2\|_{\psi_1}^2)}$, the MGF of $X_1 + X_2$ is given by

$$\begin{aligned}
\mathbb{E}[\exp(\lambda(X_1 + X_2))] &\leq \mathbb{E}[\exp(2\lambda X_1)]^{1/2}\mathbb{E}[\exp(2\lambda X_2)]^{1/2} \\
&\leq \exp(C\|X_1\|_{\psi_1}^2 2\lambda^2)\exp(C\|X_2\|_{\psi_1}^2 2\lambda^2) \\
&\leq \exp(C\lambda^2(2\|X_1\|_{\psi_1}^2 + 2\|X_2\|_{\psi_1}^2)). \quad\quad\quad \square
\end{aligned}$$

Using $\|X_1\|_{\psi_1} + \|X_2\|_{\psi_1} \geq \max(\|X_1\|_{\psi_1}, \|X_2\|_{\psi_1})$, we know that above inequality is true for any $\lambda$ with $|\lambda| \leq \frac{1}{2C(\|X_1\|_{\psi_1} + \|X_2\|_{\psi_1})^2} \leq \frac{1}{2C \max(\|X_1\|_{\psi_1}^2, \|X_2\|_{\psi_1}^2)}$. This completes the proof.

**Lemma B.3** (Corollary 2.8.4 in [Ver]). *(Bernstein's inequality for sub-exponential random variable) Let $X_1, X_2, \ldots, X_N$ be independent, mean zero, sub-exponential random variables. Then, for every $\varepsilon \geq 0$, we have*

$$\Pr\left[\left|\frac{1}{N}\sum_{i=1}^{N} X_i\right| \geq \varepsilon\right] \leq 2\exp\left[-cN\min\left(\frac{\varepsilon}{\max_i \|X_i\|_{\psi_1}}, \frac{\varepsilon^2}{(\max_i \|X_i\|_{\psi_1})^2}\right)\right]$$

*where $c > 0$ is some absolute constant.*

## C   Learning mixtures of two Gaussians with constant separation

In this section, we provide the details and proofs for learning mixtures of two Gaussians with constant separation. Our results in this section can be summarized in the following theorem statement.

**Theorem C.1** (Formal version of Theorem 7). *Let $q$ be a mixture of two Gaussians (in the form of Eq. (7)) with mean parameter $\mu^*$ satisfying $\|\mu^*\| > c$ for some absolute constant $c > 0$. Recalling that $B$ denotes an* a priori *upper bound on $\|\mu^*\|$, we have that for any $\varepsilon \le \varepsilon'$ where $\varepsilon' \lesssim \frac{1}{d^2 B^9}$, there exists a procedure satisfying the following. If the procedure is run for at least $\Omega(B^6 \log(d/\varepsilon))$ iterations with at least $\mathrm{poly}(d, B)/\varepsilon^2$ samples from $q$, then it outputs $\tilde{\mu}$ such that $\|\tilde{\mu} - \mu^*\| \le \varepsilon$ with high probability.*

As described earlier, the procedure first runs gradient descent on the DDPM objective described in Algorithm 1 from a random Gaussian initialization in a high noise scale regime with noise scale $t_1 = O(\log d)$. It then uses the output of the first step as initialization and runs the Algorithm 1 in a low noise scale regime with noise scale $t_2 = O(1)$.

We begin by calculating the form of the gradient updates:

**Lemma C.2.** *For any noise scale $t > 0$, the gradient update for the mixture of two Gaussians on the DDPM objective is given by*

$$-\nabla_{\mu_t} L_t(s_{\mu_t}) = \mathbb{E}_{x \sim \mathcal{N}(\mu_t^*, \mathrm{Id})} \Big[ \big( \tanh(\mu_t^\top x) - \frac{1}{2} \tanh''(\mu_t^\top x)\|\mu_t\|^2 + \tanh'(\mu_t^\top x)\mu_t^\top x \big) x \Big]$$
$$- \mu_t - \mathbb{E}_{x \sim \mathcal{N}(\mu_t^*, \mathrm{Id})} \Big[ \tanh'(\mu_t^\top x)\mu_t \Big] .$$

The proof of Lemma C.2 is given in Appendix F.1.

## C.1 High noise regime–connection to power iteration

Here we show that running population gradient descent on the DDPM objective at *high* noise scale behaves like power iteration on the covariance matrix of the data and thus reaches an iterate $\mu$ with constant correlation with $\mu^*$.

**Lemma C.3.** *For any noise scale $t > t'$ and number of samples $n > n'$ where $t' \lesssim \log d$ and $n' = \Theta\big(\frac{d^4 B^3}{\varepsilon^2}\big)$, with high probability, the negative gradient of the diffusion model objective $L_t(s_t)$ can be approximated by $2\mu_t^* \mu_t^{*\top} \mu_t - 3\|\mu_t\|^2 \mu_t$. More precisely, given independent samples $\{x_{i,t}\}_{i=1,\ldots,n}$ from $q_t$ generated using noise vectors $\{z_{i,t}\}_{i=1,\ldots,n}$ sampled from $\mathcal{N}(0, \mathrm{Id})$, we have*

$$\Big\| -\nabla\Big(\frac{1}{n}\sum_{i=1}^n L_t(s_{\mu_t}(x_{i,t}, z_{i,t}))\Big) - \big(2\mu_t^*\mu_t^{*\top}\mu_t - 3\|\mu_t\|^2\mu_t\big) \Big\| \le 250\sqrt{d}\|\mu_t\|^5 + 10\|\mu_t\|^3\|\mu^*\|^2 + \varepsilon .$$

*Proof.* Recall that the population gradient update on the DDPM objective is given by

$$-\nabla L_t(s_{\mu_t}) = \mathbb{E}_{x \sim \mathcal{N}(\mu_t^*, \mathrm{Id})}\big[ \tanh(\mu_t^\top x)x - \frac{1}{2}\tanh''(\mu_t^\top x)\|\mu_t\|^2 x + \tanh'(\mu_t^\top x)\mu_t^\top xx \big]$$
$$- \mu_t - \mathbb{E}_{x \sim \mathcal{N}(\mu_t^*, \mathrm{Id})}[\tanh'(\mu_t^\top x)\mu_t]$$
$$= \mathbb{E}_{x \sim \mathcal{N}(\mu_t^*, \mathrm{Id})}\big[ \tanh(\mu_t^\top x)x - \frac{1}{2}\tanh''(\mu_t^\top x)\|\mu_t\|^2 x + \tanh'(\mu_t^\top x)\mu_t^\top x\mu_t^*$$
$$+ \tanh''(\mu_t^\top x)\mu_t^\top x\mu_t \big] - \mu_t ,$$

where the last equality follows from the Stein's lemma on $\mathbb{E}_{x \sim \mathcal{N}(\mu_t^*, \mathrm{Id})}[\tanh'(\mu_t^\top x)\mu_t^\top xx]$, as

$$\mathbb{E}_{x \sim \mathcal{N}(\mu_t^*, \mathrm{Id})}[\tanh'(\mu_t^\top x)\mu_t^\top xx] = \mathbb{E}_{x \sim \mathcal{N}(\mu_t^*, \mathrm{Id})}[\tanh'(\mu_t^\top x)\mu_t^\top x\mu_t^* + \tanh'(\mu_t^\top x)\mu_t + \tanh''(\mu_t^\top x)\mu_t^\top x\mu_t].$$

Using Taylor's theorem, we know that

$$\tanh(\mu_t^\top x) = \mu_t^\top x - \frac{2}{3}(\mu_t^\top x)^3 + O(\xi(x)^5) \qquad \text{where } \xi(x) \in [0, \mu_t^\top x]$$
$$\implies \tanh(\mu^\top x)x = \mu^\top xx - \frac{2}{3}(\mu_t^\top x)^3 x + O(\xi(x)^5 x)$$
$$\implies \Big\| \mathbb{E}_{x \sim \mathcal{N}(\mu_t^*, \mathrm{Id})}[\tanh(\mu_t^\top x)x] - \mathbb{E}_{x \sim \mathcal{N}(\mu_t^*, \mathrm{Id})}\big[\mu_t^\top xx - \frac{2}{3}(\mu_t^\top x)^3 x\big] \Big\| \le \|\mathbb{E}[\xi(x)^5 x]\| \lesssim \sqrt{d}\|\mu_t\|^5$$

where the last inequality follows from $\left\|\mathbb{E}[\xi(x)^5 x]\right\| \leq \mathbb{E}[|\mu_t^\top x|^5\|x\|] \leq \left(\mathbb{E}[|\mu_t^\top x|^{10}]\right)^{1/2}\left(\mathbb{E}[\|x\|^2]\right)^{1/2} \lesssim \|\mu_t\|^5\sqrt{d+\|\mu_t^*\|^2} \lesssim \sqrt{d}\|\mu_t\|^5$. Similarly, using Taylor's theorem, we get

$$\tanh''(\mu_t^\top x) = -2\mu_t^\top x + O(\xi(x)^3) \qquad \text{where } \xi(x) \in [0, \mu_t^\top x]$$

$$\implies \tanh''(\mu_t^\top x)\left(-\frac{1}{2}\|\mu_t\|^2 x + \mu_t^\top x\mu_t\right) = \left(-2\mu_t^\top x + O(\xi(x)^3)\right)\left(-\frac{1}{2}\|\mu_t\|^2 x + \mu_t^\top x\mu_t\right)$$

$$\implies \left\|\mathbb{E}[\tanh''(\mu_t^\top x)\left(-\frac{1}{2}\|\mu_t\|^2 x + \mu_t^\top x\mu_t\right)] - \mathbb{E}\left[-2\mu_t^\top x\left(-\frac{1}{2}\|\mu_t\|^2 x + \mu_t^\top x\mu_t\right)\right]\right\|$$

$$\leq \left\|-\frac{1}{2}\|\mu_t\|^2 \,\mathbb{E}_{x\sim\mathcal{N}(\mu_t^*, I)}[O(\xi(x)^3)x] + \mathbb{E}_{x\sim\mathcal{N}(\mu_t^*, I)}[O(\xi(x)^3)\mu_t^\top x\mu_t]\right\|$$

$$\leq \frac{1}{2}\|\mu_t\|^2\,\mathbb{E}[|\mu_t^\top x|^3\|x\|] + \|\mu_t\|\,\mathbb{E}[|\mu_t^\top x|^4]$$

$$\leq \frac{1}{2}\|\mu_t\|^2\,\sqrt{\mathbb{E}[|\mu_t^\top x|^6]\mathbb{E}[\|x\|^2]} + \|\mu_t\|\,\mathbb{E}[|\mu_t^\top x|^4]$$

$$\leq 10\|\mu_t\|^5\,\sqrt{d} + 6\|\mu_t\|^5$$

Using Taylor's theorem for $\tanh'$, we get

$$\tanh'(\mu_t^\top x) = 1 - (\mu_t^\top x)^2 + O(\xi(x)^4) \qquad \text{where } \xi(x) \in [0, \mu_t^\top x]$$

$$\implies \tanh'(\mu_t^\top x)\mu_t^\top x\mu_t^* = \mu_t^\top x\mu_t^* - (\mu_t^\top x)^3\mu_t^* + O(\xi(x)^4\mu_t^\top x\mu_t^*) \qquad \text{where } \xi(x) \in [0, \mu_t^\top x]$$

$$\implies \left\|\mathbb{E}[\tanh'(\mu_t^\top x)\mu_t^\top x\mu_t^*] - \mathbb{E}[\mu_t^\top x\mu_t^* - (\mu_t^\top x)^3\mu_t^*]\right\| \leq \left\|\mathbb{E}[\xi(x)^4(\mu_t^\top x)\mu_t^*]\right\|$$

$$\leq \mathbb{E}[|\mu_t^\top x|^5]\|\mu_t^*\| \lesssim \|\mu_t^*\|\|\mu_t\|^5$$

Additionally, we have

$$\mathbb{E}_{x\sim\mathcal{N}(\mu_t^*, \mathrm{Id})}[xx^\top\mu_t(1+\|\mu_t\|^2) - \frac{2}{3}(\mu_t^\top x)^3 x - 2\mu_t(\mu_t^\top x)^2 + \mu_t^\top x\mu_t^* - (\mu_t^\top x)^3\mu_t^*]$$

$$= (I + \mu_t^*\mu_t^{*\top})\mu_t(1+\|\mu_t\|^2) - \frac{5}{3}\mathbb{E}[(\mu_t^\top x)^3\mu_t^*] + \mu_t^*\mu_t^{*\top}\mu_t - 4\mathbb{E}[\mu_t(\mu_t^\top x)^2]$$

$$= (I + \mu_t^*\mu_t^{*\top})\mu_t(1+\|\mu_t\|^2) - \frac{5\mu_t^*}{3}((\mu_t^\top\mu_t^*)^3 + 3(\mu_t^\top\mu_t^*)\|\mu_t\|^2)$$
$$\quad + \mu_t^*\mu_t^{*\top}\mu_t - 4\mu_t(\|\mu_t\|^2 + (\mu_t^\top\mu_t^*)^2)$$

$$= \mu_t^*\mu_t^{*\top}\mu_t(2 - 4\|\mu_t\|^2) + \mu_t(1 - 3\|\mu_t\|^2) - \frac{5\mu_t^*(\mu_t^\top\mu_t^*)^3}{3} - 4\mu_t(\mu_t^\top\mu_t^*)^2$$

where the second equality uses Stein's lemma on $\mathbb{E}[(\mu_t^\top x)^3 x]$ and $\mathbb{E}[xx^\top] = \mathrm{Id} + \mu_t^*\mu_t^{*\top}$ and the third equality uses Gaussian moments for $\mathbb{E}[(\mu_t^\top x)^2]$ and $\mathbb{E}[(\mu_t^\top x)^3]$. Putting it all together and using triangle inequality, we obtain the desired bound on $\|-\nabla L_t(s_{\mu_t}) - (2\mu_t^*\mu_t^{*\top}\mu_t - 3\|\mu_t\|^2\mu_t)\|$.

$$\|-\nabla L_t(s_{\mu_t}) - (2\mu_t^*\mu_t^{*\top}\mu_t - 3\|\mu_t\|^2\mu_t)\|$$

$$\leq \left\|-\nabla L_t(s_{\mu_t}) - \mathbb{E}[xx^\top\mu_t(1+\|\mu_t\|^2) - \frac{2}{3}(\mu_t^\top x)^3 x - 2\mu_t(\mu_t^\top x)^2 + \mu_t^\top x\mu_t^* - (\mu_t^\top x)^3\mu_t^* - \mu_t]\right\|$$

$$\quad + \left\|\mathbb{E}[xx^\top\mu_t(1+\|\mu_t\|^2) - \frac{2}{3}(\mu_t^\top x)^3 x - 2\mu_t(\mu_t^\top x)^2 + \mu_t^\top x\mu_t^* - (\mu_t^\top x)^3\mu_t^* - \mu_t]\right.$$
$$\quad\left. - \left(2\mu_t^*\mu_t^{*\top}\mu_t - 3\|\mu_t\|^2\mu_t\right)\right\|$$

$$\leq \left(200\sqrt{d}\|\mu_t\|^5 + 10\|\mu_t\|^5\sqrt{d} + 6\|\mu_t\|^5 + 20\|\mu_t^*\|\|\mu_t\|^5\right) + 10\|\mu_t\|^3\|\mu_t^*\|^2$$

$$\leq 250\sqrt{d}\|\mu_t\|^5 + 10\|\mu_t\|^3\|\mu_t^*\|^2$$

Using Lemma E.7 and triangle inequality, we obtain the result. □

We will use the following simple bound on the correlation between the ground truth and a random initialization:

**Lemma C.4.** *A randomly initialized $\mu_0 \sim \mathcal{N}(0, \mathrm{Id})$ satisfies that $\left|\langle \hat{\mu}_0, \hat{\mu}^* \rangle\right| \geq \frac{1}{2d}$ with probability at least $1 - O(d^{-1/2})$.*

*Proof.* For $\mu_0 \sim \mathcal{N}(0, I)$, we know that $\langle \mu_0, \hat{\mu}^* \rangle \sim \mathcal{N}(0, I)$. Using Gaussian anti-concentration, with probability at least $1 - 1/\sqrt{d}$, we have $\left|\langle \mu_0, \hat{\mu}^* \rangle\right| \geq 1/\sqrt{d}$. Because the $L_2$ norm of a Gaussian vector is sub-exponential, with probability at least $1 - \exp(-\Omega(d))$, we have $\|\mu_0\| \leq 2\sqrt{d}$. Using the norm bound, with probability at least $1 - 1/\sqrt{d} - \exp(-O(d)) = 1 - O(d^{-1/2})$, we obtain the claimed bound on $\left|\langle \hat{\mu}_0, \hat{\mu}^* \rangle\right|$. $\qquad\square$

We can now track the correlation between the iterates of gradient descent and the ground truth:

**Lemma C.5.** *Suppose that the vector $\mu_t$ satisfies $|\langle \hat{\mu}_t, \hat{\mu}_t^* \rangle| \geq \frac{1}{2d}$, and let $\mu_t'$ denote the iterate resulting from a single empirical gradient step with learning rate $\eta$ starting from $\mu_t$. Suppose that the empirical gradient and the population gradient differ by at most $\varepsilon$. Denote the angle between $\mu_t$ (resp. $\mu_t'$) and $\mu_t^*$ by $\theta$ (resp. $\theta'$). Then*

$$\tan \theta' = \max\left(\kappa_1 \tan \theta, \kappa_2\right)$$

*for*

$$\kappa_1 = \frac{1 - 3\eta\|\mu_t\|^2}{1 - 3\eta\|\mu_t\|^2 + \eta(\|\mu_t^*\|^2 - 500\sqrt{d^3}\|\mu_t\|^4 - 20d\|\mu_t\|^2\|\mu_t^*\|^2 - \eta\tilde{\varepsilon})}\ ,$$

$$\kappa_2 = \frac{500\eta\sqrt{d^3}\|\mu_t\|^4 + 20\eta d\|\mu_t\|^2\|\mu_t^*\|^2 + \eta\tilde{\varepsilon}}{\|\mu_t^*\|^2} \quad \text{and} \quad \tilde{\varepsilon} \lesssim \frac{d\varepsilon}{\|\mu_t\|}\ .$$

*Proof.* Define $\hat{\mu}_t^{*\perp}$ as the orthogonal vector to $\mu_t^*$ in the plane of $\mu_t$ and $\mu_t^*$. Note that $\mu_t'$ still lies in this plane, so the orthogonal vector to $\mu_t^*$ in the plane of $\mu_t'$ and $\mu_t^*$ is also given by $\hat{\mu}_t^{*\perp}$.

We have

$$
\begin{aligned}
\tan \theta' &= \frac{\langle \hat{\mu}^{*\perp}, \hat{\mu}_t' \rangle}{\langle \hat{\mu}_t^*, \hat{\mu}_t' \rangle} = \frac{\langle \hat{\mu}_t^{*\perp}, \mu_t' \rangle}{\langle \hat{\mu}_t^*, \mu_t' \rangle} \\
&= \frac{\langle \hat{\mu}_t^{*\perp}, \mu_t + \eta F(\mu_t, \mu_t^*) \rangle + \langle \hat{\mu}_t^{*\perp}, -\eta \nabla L_t(s_t) - \eta F(\mu_t, \mu_t^*) \rangle + \eta\varepsilon}{\langle \hat{\mu}_t^*, \mu_t + \eta F(\mu_t, \mu_t^*) \rangle + \langle \hat{\mu}_t^{*\perp}, -\eta \nabla L_t(s_t) - \eta F(\mu_t, \mu_t^*) \rangle - \eta\varepsilon} \\
&\qquad\qquad\qquad\qquad\text{where}\quad F(\mu, \mu^*) = \left(2\mu_t^* \mu_t^{*\top} \mu_t - 3\|\mu_t\|^2 \mu_t\right) \\
&\leq \frac{\sigma_2 \langle \hat{\mu}_t^{*\perp}, \mu_t \rangle + \eta \|\nabla L_t(s_t) + F(\mu_t, \mu_t^*)\| + \eta\varepsilon}{\sigma_1 \langle \hat{\mu}_t^*, \mu_t \rangle - \eta \|\nabla L_t(s_t) + F(\mu_t, \mu_t^*)\| - \eta\varepsilon} \qquad\qquad\text{(C.1)}
\end{aligned}
$$

where $\sigma_1$ and $\sigma_2$ are the first and second eigenvalues of $\mathrm{Id} + F(\mu_t, \mu_t^*) = (1 - 3\eta\|\mu_t\|^2)\mathrm{Id} + 2\eta\mu_t^*\mu_t^{*\top}$, given by

$$
\begin{aligned}
\sigma_1 &= 1 + \eta(2\|\mu_t^*\|^2 - 3\|\mu_t\|^2) \\
\sigma_2 &= 1 - 3\eta\|\mu_t\|^2\ .
\end{aligned}
$$

The last inequality (C.1) follows from the fact that

$$
\begin{aligned}
\langle \hat{\mu}_t^*, \mu_t + \eta F(\mu_t, \mu_t^*) \rangle &= \hat{\mu}_t^{*\top}((1 - 3\eta\|\mu_t\|^2)\mathrm{Id} + 2\eta\mu_t^*\mu_t^{*\top})\mu_t \\
&= \mu_t^\top((1 - 3\eta\|\mu_t\|^2)\mathrm{Id} + 2\eta\mu_t^*\mu_t^{*\top})\hat{\mu}_t^* = \sigma_1 \mu_t^\top \hat{\mu}_t^*
\end{aligned}
$$

because $\hat{\mu}^*$ is the first eigenvector of $(1 - 3\eta\|\mu_t\|^2)\mathrm{Id} + 2\eta\mu_t^*\mu_t^{*\top}$. Recall from Lemma C.3 that the deviation between the negative population gradient and the power iteration update $F(\mu_t, \mu_t^*)$ is bounded by

$$\frac{\|\nabla L_t(s_t) + F(\mu_t, \mu_t^*)\|}{\langle \mu_t, \hat{\mu}_t^* \rangle} \leq \frac{250\eta\sqrt{d}\|\mu_t\|^4 + 10\eta\|\mu_t\|^2\|\mu_t^*\|^2}{\langle \hat{\mu}_t, \hat{\mu}_t^* \rangle} \leq 500\eta\sqrt{d^3}\|\mu_t\|^4 + 20d\eta\|\mu_t\|^2\|\mu_t^*\|^2\ .$$

Substituting this into Eq. (C.1), we get

$$\tan\theta' \le \frac{\sigma_2\langle\hat{\mu}_t^{*\perp},\mu_t\rangle + \eta\|\nabla L_t(s_t) + F(\mu_t,\mu_t^*)\| + \eta\varepsilon}{\langle\hat{\mu}_t^*,\mu_t\rangle(\sigma_1 - 500\eta\sqrt{d^3}\|\mu_t\|^4 - 20d\eta\|\mu_t\|^2\|\mu_t^*\|^2 - \eta\tilde{\varepsilon})} \quad \text{where} \quad \tilde{\varepsilon} \lesssim \frac{d\varepsilon}{\|\mu\|}$$

$$\le \frac{\sigma_2}{\tilde{\sigma}_1}\tan\theta + \frac{1}{\tilde{\sigma}_1}\left(500\eta\sqrt{d^3}\|\mu\|^4 + 20d\eta\|\mu\|^2\|\mu_t^*\|^2 + \eta\tilde{\varepsilon}\right)$$

$$\text{where} \quad \tilde{\sigma}_1 \triangleq \sigma_1 - 500\eta\sqrt{d^3}\|\mu\|^4 - 20d\eta\|\mu\|^2\|\mu_t^*\|^2 - \eta\tilde{\varepsilon}$$

$$\le \left(1 - \frac{\eta\|\mu_t^*\|^2}{\tilde{\sigma}_1}\right)\frac{\sigma_2}{\tilde{\sigma}_1 - \eta\|\mu_t^*\|^2}\tan\theta + \left(\frac{\eta\|\mu_t^*\|^2}{\tilde{\sigma}_1}\right)\frac{500\eta\sqrt{d^3}\|\mu_t\|^4 + 20d\eta\|\mu_t\|^2\|\mu_t^*\|^2 + \eta\tilde{\varepsilon}}{\eta\|\mu_t^*\|^2}$$

$$\le \max\left(\frac{\sigma_2}{\tilde{\sigma}_1 - \eta\|\mu_t^*\|^2}\tan\theta, \frac{500\eta\sqrt{d^3}\|\mu_t\|^4 + 20\eta d\|\mu_t\|^2\|\mu_t^*\|^2 + \eta\tilde{\varepsilon}}{\|\mu_t^*\|^2}\right)$$

where the last inequality uses the fact that convex combinations of two values is less than the maximum of two values. $\qquad\square$

Finally, we obtain the following bound on the correlation between the ground truth and the final iterate of gradient descent:

**Lemma C.6.** *For any $h \in \mathbb{N}$, let $\mu_t^{(h)}$ denote the iterate after $h$ empirical gradient steps with learning rate $\eta = 1/20$ starting from random initialization, where the empirical gradients are estimated from at least $\Theta(\frac{d^4 B^3}{\varepsilon^2})$ samples. Let $\theta^{(h)}$ denote the angle between $\mu_t^{(h)}$ and $\mu_t^*$. For any $\varepsilon \lesssim \frac{1}{d^2 B^9}$, there exists $H' \lesssim B^6 \log d$ such that for any $H \ge H'$, if $\frac{1}{B^3} \le \|\mu_t^*\| \le \frac{1}{B^2}$, we have*

$$\tan\theta^{(H)} \lesssim 1.$$

*Proof.* Denote the $h$-th iterate of gradient descent by $\mu_t^{(h)}$. In Lemma C.7 we show that $\left\|\mu_t^{(h)}\right\| \le \frac{1}{B^2}$ for all $h$. We would like to apply the bound in Lemma C.5 to argue that the angle with $\mu_t^*$ decreases when going from $\mu_t^{(h)}$ to $\mu_t^{(h+1)}$. Using that $\frac{1}{B^3} \le \|\mu_t^*\| \le \frac{1}{B^2}$ and $\|\mu_t\| \le \frac{1}{B^2}$, we can bound the quantity $\kappa_1$ that appears in Lemma C.5 by

$$\kappa_1 \le \frac{1 - 3\eta\|\mu_t\|^2}{1 - 3\eta\|\mu_t\|^2 + \frac{\eta}{B^6}\left(1 - \frac{500\sqrt{d^3}}{B^2} - \frac{20d}{B^2} - \varepsilon dB^9\right)}$$

$$\le \frac{1}{1 + \frac{\eta}{B^6}\left(1 - \frac{500\sqrt{d^3}}{B^2} - \frac{20d}{B^2} - \varepsilon dB^9\right)} \le \frac{1}{1 + \frac{\eta}{2B^6}}.$$

On the other hand, for $B$ a sufficiently large polynomial in $d$, we can again use that $\frac{1}{B^3} \le \|\mu_t^*\| \le \frac{1}{B^2}$ and $\|\mu_t\| \le \frac{1}{B^2}$ to bound the quantity $\kappa_2$ that appears in Lemma C.5 by

$$\kappa_2 \le \frac{500\eta\sqrt{d^3}}{B^2} + \frac{20\eta d}{B^4} + B^9\eta d\varepsilon \lesssim \frac{\eta}{d}.$$

As $\left|\langle\hat{\mu},\hat{\mu}^*\rangle\right| \ge \frac{1}{2d}$, this implies $|\tan\theta^{(h)}| \le 2d$. Without loss of generality assume that $\tan\theta^{(h)} \le 2d$.

By Lemma C.5, for any $h$ we either have $\tan\theta^{(h)} \lesssim \eta/d \ll 1$, in which case we are done as this bound will also hold for subsequent iterates, or $\tan\theta^{(h)} \lesssim (1 + \frac{\eta}{2B^6})^{-1}\tan\theta^{(h-1)}$. If the latter happens consecutively for $H \ge \frac{\log d}{\log(1+\frac{\eta}{2B^6})}$ steps, then because $(1 + \frac{\eta}{2B^6})^{-H} = \frac{1}{d}$, the angle $\theta$ will satisfy $\tan\theta \le 2d \cdot (1/d) \lesssim 1$. The proof is complete because, by hypothesis, $H \ge \frac{4B^6\log d}{\eta} \ge \frac{\log d}{\log(1+\frac{\eta}{2B^6})}$ (the last inequality follows from $\log(1 + x) \ge \frac{x}{2}$ for any $0 < x < 1$). $\qquad\square$

**Lemma C.7.** *When parameter $\mu_t$ satisfies $\|\mu_t\| \le \frac{1}{B^2}$ for the noise scale $t = O(\log d)$ and $\mu_t'$ is the new parameter after performing a gradient descent update on the DDPM objective at noise scale $t = O(\log d)$, then parameter $\mu_t'$ satisfies $\|\mu_t'\| \le \frac{1}{B^2}$.*

*Proof.* When $\|\mu_t\| \le 0.9\|\mu_t^*\| \le \frac{0.9}{B^2}$, we have

$$\|\mu_t'\| \le \|\mu_t + \eta F(\mu_t, \mu_t^*)\| + \eta\|(-\nabla L_t(s_{\mu_t}) - F(\mu, \mu^*))\| + \eta\varepsilon \le (1 + 2\eta\|\mu_t^*\|^2)\|\mu_t\| + \frac{1}{dB^9}$$

$$\le 1.05\|\mu_t\| + \frac{1}{dB^9} \le \frac{1}{B^2}.$$

When $\|\mu_t\| \ge 0.9\|\mu_t^*\|$, then maximum eigenvalue of $F(\mu_t, \mu_t^*)$ is negative. Therefore, $\|\mu_t'\|$ is less than $\frac{1}{B^2}$. Specifically, we have

$$\|\mu_t'\| \le \|\mu_t + \eta F(\mu_t, \mu_t^*)\| + \eta\|(-\nabla L_t(s_{\mu_t}) - F(\mu, \mu^*))\| + \eta\varepsilon$$

$$\le (1 + \eta(2\|\mu_t^*\|^2 - 3\|\mu_t\|^2))\|\mu_t\| + \frac{1}{dB^9} \le (1 - 0.01\|\mu_t^*\|^2)\|\mu_t\| + \frac{1}{dB^9} \le \frac{1}{B^2}. \quad \square$$

## C.2 Low noise regime - connection to EM algorithm

In the previous section we showed how to obtain a warm start by running gradient descent on the DDPM objective at high noise. We now focus on proving the contraction of $\|\mu_t - \mu_t^*\|$ starting from this warm start, by running gradient descent at *low* noise. We first prove the contraction for population gradient descent and then, we argue that the empirical gradient descent concentrates well around the population gradient descent.

As before, we denote $\mu_t$ as the current iterate and $\mu_t'$ as the next iterate obtained by performing (population) gradient descent on the DDPM objective with step size $\eta$. We upper bound $\|\mu_t' - \mu_t^*\|$ as follows:

$$\|\mu_t' - \mu_t^*\| = \|\mu_t - \eta\nabla_{\mu_t} L_t(s_{\mu_t}) - \mu_t^*\|$$

$$= \left\| (1 - \eta)(\mu_t - \mu_t^*) + \eta\, \mathbb{E}_{x\sim\mathcal{N}(\mu_t^*,1)}\left[ \left(\tanh(\mu_t^\top x) - \frac{1}{2}\tanh''(\mu_t^\top x)\|\mu_t\|^2 \right.\right.\right.$$

$$\left.\left.\left. + \tanh'(\mu_t^\top x)\mu_t^\top x\right)x\right] - \eta\, \mathbb{E}_{x\sim\mathcal{N}(\mu_t^*,1)}[\tanh'(\mu_t^\top x)\mu_t] - \eta\mu_t^* \right\|$$

$$\le (1 - \eta)\|\mu_t - \mu_t^*\| + \eta\left\|\mathbb{E}_{x\sim\mathcal{N}(\mu_t^*,1)}[\tanh(\mu_t^\top x)x] - \mu_t^*\right\| + \eta\|G(\mu_t, \mu_t^*)\|,$$

where

$$G(\mu_t, \mu_t^*) \triangleq \mathbb{E}_{x\sim\mathcal{N}(\mu_t^*,\mathrm{Id})}\left[ -\frac{1}{2}\tanh''(\mu_t^\top x)\|\mu_t\|^2 x + (\tanh'(\mu_t^\top x)\mu_t^\top x)x - \tanh'(\mu_t^\top x)\mu_t \right].$$

Recall that $\mathbb{E}_{x\sim\mathcal{N}(\mu_t^*,1)}[\tanh(\mu_t^\top x)x]$ is the EM update for mixtures of two Gaussians (See Fact 5). If we can show that the $G(\mu_t, \mu_t^*)$ term above is "contractive" in the sense that it is decreasing in $\|\mu_t - \mu_t^*\|$, then we can invoke existing results on convergence of EM to show that the distance between the current iterate and $\mu_t^*$ contracts in a single gradient step [DTZ17, XHM16]. Our goal is thus to control $G(\mu_t, \mu_t^*)$.

For this, we start with the 1D case in Lemma C.8. We then extend to the multi-dimensional case in Lemma C.9.

**Lemma C.8** (One-dimensional version). *Let $\mu, \mu^* > 0$, and consider $\mu \in [c, \frac{4\mu^*}{3}]$ for some constant $c$. In this one-dimensional case, the function $G$ specializes to*

$$G(\mu, \mu^*) = \mathbb{E}_{x\sim\mathcal{N}(\mu^*,1)}\left[ -\frac{1}{2}\tanh''(\mu x)\mu^2 x + \tanh'(\mu x)\mu x^2 - \tanh'(\mu x)\mu \right], \quad \text{(C.2)}$$

*and we have*

$$G(\mu, \mu^*) \le 0.01|\mu - \mu^*|$$

The proof uses the fact that the function $G$ only contains first or higher-order derivatives of the $\tanh$ function and all the derivatives of $\tanh$ decay exponential quickly as $\mu$ increases. Therefore, when $\mu$ is at least a constant, we obtain the result. The complete proof of lemma C.8 is given in Appendix F.2.

**Lemma C.9** (Multi-dimensional version). *For any noise scale $t$, when the current parameter at noise scale $t$, $\mu_t$, satisfies $\|\mu_t\| \in [c, \frac{4\langle\hat{\mu}_t, \mu_t^*\rangle}{3}]$ for some sufficiently large constant $c$, then the following inequality holds:*

$$\left\|G(\mu_t, \mu_t^*)\right\| \le 0.01\|\mu_t - \mu_t^*\|$$

*Proof.* Suppose $\{v_1, v_2, \ldots, v_d\}$ are $d$ orthonormal directions such that $v_1 = \hat{\mu}_t$ and $v_2$ is either of the two unit vectors $\hat{\mu}_t^\perp$ which are orthogonal to $\hat{\mu}_t$ in the plane of $\mu_t$ and $\mu_t^*$. Recall that

$$
\begin{aligned}
G(\mu_t, \mu_t^*) &= \mathbb{E}_{x \sim \mathcal{N}(\mu_t^*, \mathrm{Id})}\Big[ -\frac{1}{2}\tanh''(\mu_t^\top x)\|\mu_t\|^2\, x + (\tanh'(\mu_t^\top x)\mu_t^\top x)x - \tanh'(\mu_t^\top x)\mu_t \Big] \\
&= \mathbb{E}_{x \sim \mathcal{N}(0,I)}\Big[ -\frac{1}{2}\tanh''(\mu_t^\top(x + \mu_t^*))\|\mu_t\|^2\,(x + \mu_t^*) \\
&\qquad + \tanh'(\mu_t^\top(x + \mu_t^*))(\mu_t^\top(x + \mu_t^*))(x + \mu_t^*) - \tanh'(\mu_t^\top(x + \mu_t^*))\mu_t \Big] \\
&= \mathbb{E}_{\alpha_1,\alpha_2,\ldots,\alpha_d \sim \mathcal{N}(0,1)}\Big[ -\frac{1}{2}\tanh''(\|\mu_t\|\,(\alpha_1 + \hat{\mu}_t^\top\mu_t^*))\|\mu_t\|^2\,\big(\textstyle\sum_i \alpha_i v_i + \mu_t^*\big) \\
&\qquad + \tanh'(\|\mu_t\|\,(\alpha_1 + \hat{\mu}_t^\top\mu_t^*))\|\mu_t\|\,(\alpha_1 + \hat{\mu}_t^\top\mu_t^*)\big(\textstyle\sum_i \alpha_i v_i + \mu_t^*\big) \\
&\qquad - \tanh'(\|\mu_t\|\,(\alpha_1 + \hat{\mu}_t^\top\mu_t^*))\mu_t \Big],
\end{aligned}
$$

where in the last equality we rewrote $x \sim \mathcal{N}(0, I)$ as $\sum_{i=1}^d \alpha_i v_i$ for $\alpha_i \sim \mathcal{N}(0,1)$. Therefore, we have

$$
\begin{aligned}
&\langle \hat{\mu}_t, G(\mu_t, \mu_t^*) \rangle \\
&= \mathbb{E}_{\alpha_1,\alpha_2,\ldots,\alpha_d \sim \mathcal{N}(0,I)}\Big[ -\frac{1}{2}\tanh''(\|\mu_t\|\,(\alpha_1 + \hat{\mu}_t^\top\mu_t^*))\|\mu_t\|^2\,(\alpha_1 + \hat{\mu}_t^\top\mu_t^*) \\
&\qquad + \tanh'(\|\mu_t\|\,(\alpha_1 + \hat{\mu}_t^\top\mu_t^*))\|\mu_t\|\,(\alpha_1 + \hat{\mu}_t^\top\mu_t^*)^2 - \tanh'(\|\mu_t\|\,(\alpha_1 + \hat{\mu}_t^\top\mu_t^*))\|\mu_t\| \Big] \\
&= \mathbb{E}_{\alpha_1 \sim \mathcal{N}(\hat{\mu}_t^\top\mu_t^*,1)}\Big[ -\frac{1}{2}\tanh''(\|\mu_t\|\alpha_1)\|\mu_t\|^2\,\alpha_1 + \tanh'(\|\mu_t\|\,\alpha_1)\|\mu_t\|\,\alpha_1^2 - \tanh'(\|\mu_t\|\,\alpha_1)\|\mu_t\| \Big].
\end{aligned}
$$

By taking $\|\mu_t\|$ to be $\mu$ and $\langle \hat{\mu}_t, \mu_t^* \rangle$ to be $\mu^*$, we observe the similarity between the right side of the above equation and the one-dimensional definition of $G$ defined in Eq. (C.2). Using Lemma C.8 and if $\|\mu_t\| \in [c, \frac{4\langle \hat{\mu}_t, \mu_t^* \rangle}{3}]$, we have

$$
\langle \hat{\mu}_t, G(\mu_t, \mu_t^*) \rangle \leq 0.01\big|\langle \hat{\mu}_t, \mu_t \rangle - \langle \hat{\mu}_t, \mu_t^* \rangle\big|
$$

Taking the dot product of $G(\mu_t, \mu_t^*)$ with $v_2 = \hat{\mu}_t^\perp$, we have

$$
\begin{aligned}
\langle \hat{\mu}_t^\perp, G(\mu_t, \mu_t^*) \rangle &= \mathbb{E}_{\alpha_1,\alpha_2,\ldots,\alpha_d \sim \mathcal{N}(0,1)}\Big[ -\frac{1}{2}\tanh''(\|\mu_t\|\,(\alpha_1 + \hat{\mu}_t^\top\mu_t^*))\|\mu_t\|^2\,(\alpha_2 + \langle \hat{\mu}_t^\perp, \mu_t^* \rangle) \\
&\qquad + \tanh'(\|\mu_t\|\,(\alpha_1 + \hat{\mu}_t^\top\mu_t^*))\|\mu_t\|\,(\alpha_1 + \hat{\mu}_t^\top\mu_t^*)(\alpha_2 + \langle \hat{\mu}_t^\perp, \mu_t^* \rangle) \Big] \\
&= \mathbb{E}_{\alpha_1 \sim \mathcal{N}(\hat{\mu}_t^\top\mu_t^*,1)}\Big[ -\frac{1}{2}\tanh''(\|\mu_t\|\,\alpha_1)\|\mu_t\|^2\,\langle \hat{\mu}_t^\perp, \mu_t^* \rangle \\
&\qquad + \tanh'(\|\mu_t\|\,\alpha_1)\|\mu_t\|\,\alpha_1\langle \hat{\mu}_t^\perp, \mu_t^* \rangle \Big] \\
&= \langle \hat{\mu}_t^\perp, \mu_t^* \rangle\, \mathbb{E}_{\alpha_1 \sim \mathcal{N}(\hat{\mu}_t^\top\mu_t^*,1)}\Big[ -\frac{1}{2}\tanh''(\|\mu_t\|\,\alpha_1)\|\mu_t\|^2 + \tanh'(\|\mu_t\|\,\alpha_1)\|\mu_t\|\,\alpha_1 \Big].
\end{aligned}
$$

In Lemma F.5 below, we show that when $\|\mu_t\| \in [c, \frac{4\langle \hat{\mu}_t, \mu_t^* \rangle}{3}]$, the expectation in the last expression is upper bounded by 0.01. Therefore, we have

$$
\big|\langle \hat{\mu}_t^\perp, G(\mu_t, \mu_t^*) \rangle\big| \leq 0.01\big|\langle \hat{\mu}_t^\perp, \mu_t^* \rangle\big| \implies \big|\langle \hat{\mu}_t^\perp, G(\mu_t, \mu_t^*) \rangle\big| \leq 0.01\big|\langle \hat{\mu}_t^\perp, \mu_t - \mu_t^* \rangle\big|
$$

Observe that for $i = 3, \ldots, d$, $\langle G(\mu_t, \mu_t^*), v_i \rangle = 0$. Therefore, we have

$$
\big\|G(\mu_t, \mu_t^*)\big\|^2 = \sum_{i=1}^d \langle v_i, G(\mu_t, \mu_t^*) \rangle^2 \leq 0.01^2\|\mu_t - \mu_t^*\|^2. \qquad \square
$$

The next Lemma ensures that the parameter $\mu_t$ after a few steps of gradient descent on the DDPM objective stays in the region where the function $G$ satisfies $\big\|G(\mu_t, \mu_t^*)\big\| \leq 0.01\|\mu_t - \mu_t^*\|$. Recall that the condition of the Lemma is satisfied because we initialize at the warm start obtained by gradient descent in the high noise regime.

**Lemma C.10.** *Suppose the angle between initialization $\hat{\mu}_t^{(0)}$ and optimal parameter $\mu_t^*$ is $\Theta(1)$, then for any $h$, we have $\|\mu_t^{(h)}\| \in [c, \frac{4\langle\hat{\mu}_t^{(h)}, \mu_t^*\rangle}{3}]$.*

The proof of Lemma C.10 is given in Appendix F.3. Finally, we are ready to prove the main result of this section:

*Proof of Theorem C.1.* To obtain the contraction of $\|\mu_t^{(h)} - \mu_t^*\|$ after a gradient descent step on the DDPM objective, we write $\|\mu_t^{(h+1)} - \mu_t^*\|$ in terms of $\|\mu_t^{(h)} - \mu_t^*\|$ as follows:

$$\|\mu_t^{(h+1)} - \mu_t^*\| = \|\mu_t^{(h)} - \eta\nabla L_t(s_{\mu_t^{(h)}}) - \mu_t^*\| + \eta\left\|\left(\frac{1}{n}\sum_{i=1}^n \nabla L_t(s_{\mu_t^{(h)}}(x_i, z_i))\right) - \nabla L_t(s_{\mu_t^{(h)}})\right\|$$

$$\leq (1-\eta)\|\mu_t^{(h)} - \mu_t^*\| + \eta\left\|\mathbb{E}_{x\sim\mathcal{N}(\mu_t^*,1)}[(\tanh(\mu_t^{(h)\top}x))x] - \mu_t^*\right\| + \eta\|G(\mu_t^{(h)}, \mu_t^*)\| + \eta\varepsilon,$$

where in the last step we used Lemma E.7 below to bound the distance between the population and empirical gradient.

Recall that gradient descent in the low noise regime was initialized using the output of the gradient descent in the high noise regime. Therefore, $\langle\hat{\mu}_t^{(0)}, \hat{\mu}_t^*\rangle \gtrsim 1$. Using Lemma C.10, we know that the condition on Lemma C.8 is always satisfied. Using the contractivity of $G$ established in Lemma C.8 combined with [DTZ17, Theorem 2], and choosing $\eta = 0.05$, we conclude that the distance to the ground truth contracts:

$$\|\mu_t^{(h+1)} - \mu_t^*\| \leq (1 - 0.05)\|\mu_t^{(h)} - \mu_t^*\| + 0.01\|\mu_t^{(h)} - \mu_t^*\| + 0.01\|\mu_t^{(h)} - \mu_t^*\| + \eta\varepsilon$$

$$\leq 0.97\|\mu_t^{(h)} - \mu_t^*\| + \eta\varepsilon.$$

Applying the above for all $h \in [H]$, we obtain

$$\|\mu_t^{(H)} - \mu_t^*\| \leq 0.97^H\|\mu_t^{(0)} - \mu_t^*\| + 50\varepsilon.$$

The choice of $H$ given in the Theorem statement proves the result. $\qquad\square$

## D Learning mixtures of two Gaussians with small separation

In this section, we extend the analysis for learning mixtures of two Gaussians with constant separation, provided in Section C, to the low-separation regime and prove the following:

**Theorem D.1** (Formal version of Theorem 13)**.** *For any $\mathcal{L} > 0$, let $q$ be a mixture of two Gaussians (in the form of Eq. (7)) with mean parameter $\mu^*$ satisfying $\|\mu^*\| > \mathcal{L}$. Recalling that $B$ denotes an a priori upper bound on $\|\mu^*\|$, we have that for any $\varepsilon \leq \varepsilon'$, where $\varepsilon' \lesssim \frac{1}{d^2B^9}$, there exists a procedure satisfying the following. If the procedure is run for at least $\text{poly}(d, B, \frac{1}{\mathcal{L}})\frac{1}{\varepsilon^3}$ iterations with at least $\text{poly}(d, B, \frac{1}{\mathcal{L}}) * \frac{1}{\varepsilon^8}$ samples from $q$, then it outputs $\tilde{\mu}$ such that $\|\tilde{\mu} - \mu^*\| \leq \varepsilon$ with high probability.*

As described in Section 1.2, the algorithm is a simple modification of Algorithm 1 in which gradient descent is replaced by projected gradient descent. We start in Lemma D.2 by showing that the projection step in the algorithm ensures that the norm of the current iterate $\mu_t$ is approximately that of $\mu_t^*$. Then in Lemma D.3, we extend the analysis of Lemma C.5 to show that every projected gradient step contracts the distance to the ground truth. Combined with Lemma D.2, this allows us to conclude the proof of Theorem 13.

**Lemma D.2.** *Let $x_1, \ldots, x_n$ be independent samples from $q$, and define radius parameter $R$ by $R^2 \triangleq \frac{1}{n}\sum_{i=1}^n \|x_i\|^2 - d$. For any $\varepsilon > 0$, provided that $n \gtrsim \frac{B^4 + d^2}{\varepsilon^2\mathcal{L}^2}$, we have $|R - \|\mu^*\|| \leq \varepsilon$ with high probability.*

*Proof.* Observe that we can write the random variable corresponding to the mixture of two Gaussians $X_0 = X = Z + p\mu^*$ where $Z \sim \mathcal{N}(0, I)$ and $p$ is a Rademacher random variable. Using Theorem 3.1.1 (concentration of norms) from [Ver], we know that $\|\|Z\| - \sqrt{d}\|_{\psi_2} \lesssim 1$. Therefore, sub-Gaussian norm $\|\|X_0\|\|_{\psi_2} \lesssim \|\|Z\|\|_{\psi_2} + \|\|p\mu^*\|\|_{\psi_2} \lesssim B + \sqrt{d}$. Using Lemma 2.7.4 from [Ver],

we have $\left\|\|X_0\|^2\right\|_{\psi_1} \lesssim \left\|\|X_0\|\right\|_{\psi_2}^2 \lesssim B^2 + d$. Therefore, using number of samples $n$ specified in the Lemma statement, with high probability, we have

$$\left|\frac{1}{n}\sum_{i=1}^n \|x_i\|^2 - \mathbb{E}[\|X_0\|^2]\right| \le \varepsilon\mathcal{L} \implies \left|\|\mu\|^2 - \|\mu^*\|^2\right| \le \varepsilon\mathcal{L} \implies \left|\|\mu\| - \|\mu^*\|\right| \le \varepsilon$$

where the penultimate implication uses the fact that $\mathbb{E}_{X_0}[\|X_0\|^2] = \mathbb{E}[\|Z\|^2 + \|\mu^*\|^2] = d + \|\mu^*\|^2$. $\qquad\square$

**Lemma D.3.** *Assume that $\mathcal{L} \le \|\mu^*\| \le B$. Then, for any small $\varepsilon > 0$, running projected GD on diffusion models with step size $\eta = \frac{1}{20}$ at noise scale $t = \log\frac{d}{\varepsilon}$ for number of steps $H > H'$ and number of samples $n > n'$ steps will achieve*

$$\left\|\mu^{(H)} - \mu^*\right\| \lesssim d^2 B^4 \varepsilon,$$

*where $H' = \frac{d^2}{\mathcal{L}^2 \varepsilon^3}$ and $n' = \frac{d^{10} B^3}{\varepsilon^8 \mathcal{L}^6}$.*

*Proof.* Recalling that $\mu_t^* = \mu_0^* \exp(-t)$, note that for $t = \log\frac{d}{\varepsilon}$, $\frac{\varepsilon\mathcal{L}}{d} \le \|\mu_t^*\| \le \frac{\varepsilon B}{d}$. We would like to apply Lemma C.5. Note that we may apply this even though it is only stated for gradient descent (without projection). The reason is that it bounds the change in angle between the iterate and the ground truth after a single gradient step, and this angle is unaffected by projection.

Suppose we take one projected gradient step with learning rate $\eta$ starting from an iterate $\mu_t$. As $\mu_t$ was the result of a projection, by Lemma D.2 we have $\frac{\varepsilon\mathcal{L}}{d} \lesssim \|\mu_t^{(h)}\| \lesssim \frac{\varepsilon B}{d}$.

We now bound $\kappa_2$ in Lemma C.5:

$$\kappa_2 = \frac{500\eta\sqrt{d^3}\|\mu_t\|^4 + 20\eta d\|\mu_t\|^2\|\mu_t^*\|^2 + \eta\tilde{\varepsilon}}{\|\mu_t^*\|^2}$$

$$\lesssim 500\eta\sqrt{d^7}\|\mu_t\|^2 + 20\eta d\|\mu_t\|^2 + \frac{d^2\varepsilon}{\|\mu_t^*\|^3}$$

$$\le 550 d^{7/2} B^2 \exp(-2t) + \frac{d^5\varepsilon}{\varepsilon^3\mathcal{L}^3}$$

$$\lesssim d^2 B^2 \varepsilon,$$

where the last inequality follows by choosing population gradient estimation error parameter $\varepsilon = \frac{\varepsilon^4\mathcal{L}^3}{d^3}$ with the number of samples $n' = \frac{d^{11} B^6}{\varepsilon^8\mathcal{L}^6}$. Additionally, $\kappa_1$ in Lemma C.5 is given by

$$\kappa_1 = \frac{1 - 3\eta\|\mu_t\|^2}{(1 - 3\eta\|\mu_t\|^2) + \eta(\|\mu_t^*\|^2 - 500\sqrt{d^3}\|\mu_t\|^4 - 20d\|\mu_t\|^2\|\mu_t^*\|^2 - \tilde{\varepsilon})}$$

$$= \frac{1 - 3\eta\|\mu_t\|^2}{(1 - 3\eta\|\mu_t\|^2) + \eta\|\mu_t^*\|^2(1 - \kappa_2)}$$

$$\lesssim \frac{1 - 3\eta\|\mu_t^{(h)}\|^2}{(1 - 3\eta\|\mu_t^{(h)}\|^2) + \eta\|\mu_t^*\|^2(1 - d^2 B^2 \varepsilon)}$$

$$\le \frac{1}{1 + \frac{\mathcal{L}^2\varepsilon^2}{20d^2}(1 - d^2 B^2 \varepsilon)}.$$

Using bounds on $\kappa_1$ and $\kappa_2$ and Lemma C.5, we conclude that if $\theta$ (resp. $\theta'$) is the angle between $\mu_t$ (resp. the next iterate of projected gradient descent after $\mu_t$) and $\mu_t^*$

$$\tan\theta' \le \max\left(\frac{1}{1 + \frac{\mathcal{L}^2\varepsilon^2}{20d^2}(1 - B^2\varepsilon)}\tan\theta, d^2 B^2 \varepsilon\right).$$

Doing projected gradient descent for $H = \frac{20d^2}{\mathcal{L}^2\varepsilon^3}$ steps, if $\theta^{(h)}$ denotes the angle between the $h$-th iterate and $\mu_t^*$, we obtain

$$\tan\theta^{(H)} \leq \tan\theta^{(h+1)} \leq \max\left(\left(\frac{1}{1+\frac{\mathcal{L}^2\varepsilon^2}{20d^2}(1-d^2B^2\varepsilon)}\right)^H \tan\theta^{(0)}, d^2B^2\varepsilon\right)$$

$$\leq \max\left(\frac{\tan\theta^{(0)}}{1+\frac{H\mathcal{L}^2\varepsilon^2}{20d^2}(1-B^2\varepsilon)}, d^2B^2\varepsilon\right) \leq d^2B^2\varepsilon\,,$$

where the last inequality uses $1+\frac{H\mathcal{L}^2\varepsilon^2}{20d^2}(1-B^2\varepsilon) \geq \frac{1}{\varepsilon}$ for $\varepsilon \lesssim \frac{1}{B^3}$. Additionally, for a random initialization, Lemma C.4 shows that $\cos\theta^{(0)} \geq \frac{1}{2d}$ which implies $\tan\theta^{(0)} \leq \sqrt{\sec^2\theta^{(0)}-1} \lesssim d$. Using Lemma D.2, we have $\|\mu^{(H)}\| \geq \|\mu^*\| - \varepsilon$ which implies $-2\|\mu^{(H)}\|\|\mu^*\|\cos\theta^{(H)} \leq -2\|\mu^*\|^2\cos\theta^{(H)} + 2B\varepsilon$ and $\|\mu^{(H)}\|^2 \leq \|\mu^*\|^2 + 3B\varepsilon$. Using this result, we obtain

$$\|\mu^{(H)} - \mu^*\|^2 = \|\mu^{(H)}\|^2 + \|\mu^*\|^2 - 2\|\mu^{(H)}\|\|\mu^*\|\cos\theta^{(H)}$$

$$\lesssim 2\|\mu^*\|^2 - 2\|\mu^*\|^2\cos\theta^{(H)} + 5B\varepsilon \lesssim 2B^2\left(1 - \frac{1}{\sqrt{1+d^4B^4\varepsilon^2}}\right) + 5B\varepsilon \lesssim d^2B^4\varepsilon\,,$$

where the last inequality follows from the fact that $\sqrt{1+x} \leq 1 + \sqrt{x}$ for any $x > 0$. $\qquad\square$

# E  Learning mixtures of $K$ Gaussians from a warm start

In this section, we provide details about our main result on learning mixtures of $K$ Gaussians. We start by describing our main theorem in this case.

**Theorem E.1** (Formal version of Theorem 16). *Let $q$ be a mixture of Gaussians (in the form of Eq. (6)) with center parameters $\theta^* = \{\mu_1^*, \mu_2^*, \ldots, \mu_K^*\} \in \mathbb{R}^d$ satisfying the separation Assumption 14, and suppose we have estimates $\theta$ for the centers such that the warm initialization Assumption 15 is satisfied. For any $\varepsilon > \varepsilon_0$ and noise scale $t$ where*

$$\varepsilon_0 = 1/\mathrm{poly}(d) \quad and \quad t = \Theta(\varepsilon)\,,$$

*gradient descent on the DDPM objective at noise scale $t'$ (Algorithm 1) outputs $\tilde{\theta} = \{\tilde{\mu}_1, \tilde{\mu}_2, \ldots, \tilde{\mu}_K\}$ such that $\min_i \|\tilde{\mu}_i - \mu_i^*\| \leq \varepsilon$ with high probability. The algorithm runs for $H \geq H'$ iterations and uses $n \geq n'$ number of samples where*

$$H' = \Theta(\log(\varepsilon^{-1}\log d)) \quad and \quad n' = \Theta(K^4 d^5 B^6/\varepsilon^2)\,.$$

We first give an overview of the proof for population gradient descent, and then show that the empirical gradients concentrate well around the population gradients. We start by simplifying the population gradient update for mixtures of $K$ Gaussians using Stein's lemma in Lemma E.2, which yields

$$-\nabla_{\mu_{1,t}} L_t(s_{\theta_t}) = \mathbb{E}[w_{1,t}(X_t)(X_t - \mu_{1,t})] + [\text{extra terms}]\,,$$

recalling the notation of Eq. (9). As discussed in the body of the paper, $\mathbb{E}[w_{1,t}(X_t)(X_t - \mu_{1,t})]$ is precisely the update for the gradient EM algorithm (see Fact 6) and known results for the latter [KC20, SN21] can be used to show that the distance $\|\mu_{1,t} - \mu_{1,t}^*\|$ contracts in each step when the separation Assumption 14 and the warm initialization Assumption 15 are satisfied. Therefore, showing that the "extra terms" do not disturb the progress coming from the gradient EM update is sufficient. We prove that the "extra terms" are $1/\mathrm{poly}(d)$ in Lemma E.4 when the separation Assumption 14 and warm initialization Assumption 15 hold.

The intuition behind Lemma E.4 is as follows: We start with a key observation that each of the "extra terms" either contains $w_{1,t}(X_t)(1 - w_{1,t}(X_t))$ or $w_{1,t}(X_t)w_{j,t}(X_t)$ where $j \neq 1$. Note that the $w_{1,t}(X_t)$ can be interpreted as the conditional probability of the underlying component being $\mathcal{N}(\mu_{1,t}, I)$ given $X_t$. When Assumption 14 and Assumption 15 are satisfied, Proposition 4.1 of [SN21] shows that

$$\mathbb{E}_{X_t \sim \mathcal{N}(\mu_{1,t}^*, I)}[w_{j,t}(X_t)] \lesssim 1/\mathrm{poly}(d) \quad \text{for any } j \neq 1\,.$$

This result can be extended to show both $\mathbb{E}_{X_t}[w_{1,t}(X_t)(1 - w_{1,t}(X_t))] \lesssim 1/\mathrm{poly}(d)$ as well as $\mathbb{E}_{X_t}[w_{1,t}(X_t)w_{j,t}(X_t)] \lesssim 1/\mathrm{poly}(d)$ for any $j \neq 1$ (see Lemma E.5 for the proof). Using these bounds, we conclude that $[\text{"extra terms"}] \lesssim 1/\mathrm{poly}(d)$ in Lemma E.4.

## E.1 EM and population gradient descent on DDPM objective

We begin by writing out the gradient update explicitly:

**Lemma E.2.** *For any noise scale $t > 0$, the gradient of the population DDPM objective $\mathbb{E}[L_t(s_{\theta_t}(X_t))]$ with respect to parameter $\mu_{1,t}$ is given by*

$$\nabla_{\mu_{1,t}} L_t(s_{\theta_t}) = \mathbb{E}\Big[ -w_{1,t}(X_t)(X_t - \mu_{1,t}) + w_{1,t}(X_t)(X_t - \mu_{1,t}) \sum_{i=1}^{K} w_{i,t}(X_t)\mu_{i,t}^{\top}(X_t - \mu_{1,t})$$

$$+ w_{1,t}(X_t)\mu_{1,t} - w_{1,t}(X_t)(X_t - \mu_{1,t})^{\top}\mu_{1,t}(X_t - \mu_{1,t}) - w_{1,t}(X_t) \sum_{i=1}^{K} w_{i,t}(X_t)\mu_{i,t}$$

$$- w_{1,t}(X_t) \sum_{i=1}^{K} \nabla_x w_{i,t}(X_t)^{\top}\mu_{i,t}(X_t - \mu_{1,t})\Big]$$

*where $w_{1,t}(x)$ and $\mu_{1,t}$ are defined in Eq. (9).*

*Proof.* Recall that the score function of mixture of Gaussians is given by

$$s_{\theta_t}(X_t) = \sum_{i} w_{i,t}(X_t)\mu_{i,t} - X_t$$

Finding the gradient $\nabla_{\mu_{1,t}} w_{i,t}(X_t)$, we have

$$\nabla_{\mu_{1,t}} w_{i,t}(X_t) = \begin{cases} w_{1,t}(X_t)(1 - w_{1,t}(X_t))(X_t - \mu_{1,t}) & \text{if } i = 1 \\ -w_{1,t}(X_t)w_{i,t}(X_t)(X_t - \mu_{1,t}) & \text{otherwise.} \end{cases}$$

The gradient of the score function is given by

$$\nabla_{\mu_{1,t}} s_{\theta_t}(X_t) = \nabla_{\mu_{1,t}} \big( w_{1,t}(X_t)\mu_{1,t} \big) + \sum_{i=2}^{K} \nabla_{\mu_{1,t}} \big( w_{i,t}(X_t)\mu_{i,t} \big)$$

$$= w_{1,t}(X_t)(1 - w_{1,t}(X_t))\mu_{1,t}(X_t - \mu_{1,t})^{\top} + w_{1,t}(X_t)I - w_{1,t}(X_t) \sum_{i=2}^{K} w_{i,t}(X_t)\mu_{i,t}(X_t - \mu_{1,t})^{\top}$$

$$= w_{1,t}(X_t)\mu_{1,t}(X_t - \mu_{1,t})^{\top} + w_{1,t}(X_t)I - w_{1,t}(X_t) \sum_{i=1}^{K} w_{i,t}(X_t)\mu_{i,t}(X_t - \mu_{1,t})^{\top}.$$

The gradient of $\frac{1}{2}\|s_{\theta_t}\|^2$ is given by

$$\frac{1}{2}\nabla\big\|s_{\theta_t}(X_t)\big\|^2 = \sum_{j=1}^{d} [s_{\theta_t}(X_t)]_j [\nabla_{\mu_{1,t}} s_{\theta_t}(X_t)]_j = \nabla_{\mu_{1,t}} s_{\theta_t}(X_t)^{\top} s_{\theta_t}(X_t)$$

$$\text{where } [\nabla_{\mu_{1,t}} s_{\theta_t}(X_t)]_j \text{ is } j^{th} \text{ row of } \nabla_{\mu_{1,t}} s_{\theta_t}(X_t).$$

The gradient of this is given by

$$\frac{\nabla_{\mu_{1,t}} s_{\theta_t}(X_t)^{\top} Z_t}{\beta_t} = \frac{1}{\beta_t}\Big( w_{1,t}(X_t)(X_t - \mu_{1,t})\mu_{1,t}^{\top}Z_t + w_{1,t}(X_t)Z_t$$

$$- w_{1,t}(X_t) \sum_{i=1}^{K} w_{i,t}(X_t)(X_t - \mu_{1,t})\mu_{i,t}^{\top}Z_t \Big) \qquad \text{(E.1)}$$

Applying Stein's lemma to the expectation of the first term in Eq. (E.1), we have

$$\mathbb{E}_{X_0,Z_t}[w_{1,t}(X_t)(X_t - \mu_{1,t})\mu_{1,t}^{\top}Z_t] = \sum_{j=1}^{d} \mathbb{E}_{X_0,Z_t}[w_{1,t}(X_t)(X_t - \mu_{1,t})\mu_{1,t,j}Z_{t,j}]$$

$$= \sum_{j=1}^{d} \mathbb{E}_{X_0,Z_t}[w_{1,t}(X_t)\beta_t e_j \mu_{1,t,j} + \beta_t \nabla_x w_{1,t}(X_t)^{\top} e_j (X_t - \mu_{1,t})\mu_{1,t,j}]$$

$$= \mathbb{E}_{X_0,Z_t}[w_{1,t}(X_t)\beta_t \mu_{1,t} + \beta_t \nabla_x w_{1,t}(X_t)^{\top}\mu_{1,t}(X_t - \mu_{1,t})]$$

The expectation of the second term in Eq. (E.1) simplifies to $\beta_t \mathbb{E}_{X_t}[\nabla_x w_{1,t}(X_t)]$ by Stein's Lemma. Each summand in the third term in Eq. (E.1) simplifies as following:

$$
\mathbb{E}_{X_0,Z_t}\left[w_{1,t}(X_t)w_{i,t}(X_t)(X_t - \mu_{1,t})\mu_{i,t}^\top Z_t\right]
$$

$$
= \sum_{j=1}^d \mathbb{E}_{X_0,Z_t}\left[w_{1,t}(X_t)w_{i,t}(X_t)(X_t - \mu_{1,t})\mu_{i,t,j}Z_{t,j}\right]
$$

$$
= \sum_j \mu_{i,t,j}\mathbb{E}_{X_0,Z_t}\left[w_{1,t}(X_t)w_{i,t}(X_t)\beta_t e_j + \beta_t w_{1,t}(X_t)\nabla_x w_{i,t}(X_t)^\top e_j (X_t - \mu_{1,t})\right.
$$

$$
\left. + \beta_t \nabla_x w_{1,t}(X_t)^\top e_j w_{i,t}(X_t)(X_t - \mu_{1,t})\right]
$$

$$
= \beta_t \, \mathbb{E}_{X_0,Z_t}\left[w_{1,t}(X_t)w_{i,t}(X_t)\mu_{i,t} + w_{1,t}(X_t)\nabla_x w_{i,t}(X_t)^\top \mu_{i,t}(X_t - \mu_{1,t})\right.
$$

$$
\left. + \nabla_x w_{1,t}(X_t)^\top \mu_{i,t} w_{i,t}(X_t)(X_t - \mu_{1,t})\right] \tag{E.2}
$$

Combining the gradients of all the terms of Eq. (E.2), we have

$$
\nabla_{\mu_{1,t}} L_t(s_{\theta_t})
$$

$$
= \mathbb{E}\Big[w_{1,t}(X_t)(X_t - \mu_{1,t})\mu_{1,t}^\top s_{\theta_t}(X_t) + w_{1,t}(X_t)s_{\theta_t}(X_t) - w_{1,t}(X_t)(X_t - \mu_{1,t})\sum_i w_{i,t}(X_t)\mu_{i,t}^\top s_{\theta_t}(X_t)
$$

$$
+ \nabla_x w_{1,t}(X_t) + w_{1,t}(X_t)\mu_{1,t} + \nabla_x w_{1,t}(X_t)^\top \mu_{1,t}(X_t - \mu_{1,t}) - w_{1,t}(X_t)\sum_i w_{i,t}(X_t)\mu_{i,t}
$$

$$
- w_{1,t}(X_t)\sum_i \nabla_x w_{i,t}(X_t)^\top \mu_{i,t}(X_t - \mu_{1,t}) - \sum_i \nabla_x w_{1,t}(X_t)^\top \mu_{i,t}w_{i,t}(X_t)(X_t - \mu_{1,t})\Big]
$$

$$
= \mathbb{E}\Big[-w_{1,t}(X_t)(X_t - \mu_{1,t}) + w_{1,t}(X_t)(X_t - \mu_{1,t})\sum_i w_{i,t}(X_t)\mu_{i,t}^\top(X_t - \mu_{1,t})
$$

$$
+ w_{1,t}(X_t)\mu_{1,t} - w_{1,t}(X_t)(X_t - \mu_{1,t})^\top \mu_{1,t}(X_t - \mu_{1,t}) - w_{1,t}(X_t)\sum_i w_{i,t}(X_t)\mu_{i,t}
$$

$$
- w_{1,t}(X_t)\sum_i \nabla_x w_{i,t}(X_t)^\top \mu_{i,t}(X_t - \mu_{1,t})\Big],
$$

where the last equality uses Lemma E.3. Specifically, it uses

$$
\nabla_x w_{1,t}(X_t) + w_{1,t}(X_t)s_{\theta_t}(X_t) = -w_{1,t}(X_t)(X_t - \mu_{1,t})
$$

$$
(\nabla_x w_{1,t}(X_t) + w_{1,t}(X_t)s_{\theta_t}(X_t))^\top \mu_{1,t}(X_t - \mu_{1,t}) = -w_{1,t}(X_t)(X_t - \mu_{1,t})^\top \mu_{1,t}(X_t - \mu_{1,t}). \;\square
$$

We will also need the following intermediate calculation:

**Lemma E.3.** *For any $i \in [K]$, the gradient of $w_{i,t}(X_t)$ with respect to $X_t$ is given by*

$$
\nabla_x w_{i,t}(X_t) = -w_{i,t}(X_t)(X_t - \mu_{i,t}) - w_{i,t}(X_t)s_{\theta_t}(X_t)
$$

$$
= -w_{i,t}(X_t)(1 - w_{i,t}(X_t))(X_t - \mu_{i,t}) + w_{i,t}(X_t) \cdot \sum_{j \in [K]: j \neq i} w_{j,t}(X_t)(X_t - \mu_{j,t}).
$$

*Proof.* By taking the gradient of $w_{i,t}(X_t)$ and simplifying it, we get the result:

$$\nabla_x w_{i,t}(X_t) = -\frac{\exp\left(-\frac{\|X_t - \mu_{i,t}\|^2}{2}\right)(X_t - \mu_{i,t})}{\sum_{j=1}^K \exp\left(-\frac{\|X_t - \mu_{j,t}\|^2}{2\sigma^2}\right)}$$

$$+\frac{\exp\left(-\frac{\|X_t - \mu_{i,t}\|^2}{2}\right) \cdot \sum_{j=1}^K \exp\left(-\frac{\|X_t - \mu_{j,t}\|^2}{2}\right)(X_t - \mu_{j,t})}{\left(\sum_{j=1}^K \exp\left(-\frac{\|X_t - \mu_{j,t}\|^2}{2}\right)\right)^2}$$

$$= -w_{i,t}(X_t)(X_t - \mu_{i,t}) + w_{i,t}(X_t)\left(\sum_{j=1}^K w_{j,t}(X_t)(X_t - \mu_{j,t})\right)$$

$$= -w_{i,t}(X_t)(1 - w_{i,t}(X_t))(X_t - \mu_{i,t}) + w_{i,t}(X_t)\left(\sum_{j=1,j\neq i}^K w_{j,t}(X_t)(X_t - \mu_{j,t})\right). \quad \square$$

We are now ready to establish the connection between gradient descent on the DDPM objective and the gradient EM update, for mixtures of $K$ Gaussians:

**Lemma E.4.** *Suppose the centers of the mixture of $K$ Gaussians are well-separated according to Assumption 14, and the parameters $\theta = \{\mu_1, \mu_2, \ldots, \mu_K\}$ that the student network is initialized to satisfy the warm start Assumption 15. Then, for noise scale $t = O(1)$, gradient descent on the DDPM objective is close to the gradient EM update:*

$$\left\|\nabla_{\mu_{1,t}} L_t(s_{\theta_t}) + \mathbb{E}[w_{1,t}(X_t)(X_t - \mu_{1,t})]\right\| \lesssim \frac{K^2 B^2}{d^{c_r^2/4000}} = \frac{1}{\text{poly}(d)},$$

*where $c_r$ is a large constant.*

*Proof.* Observe that the first term in the expression for the population gradient of the DDPM objective in Lemma E.2 is exactly the gradient EM update for the mixture of $K$ Gaussian in Fact 6. To prove the closeness between the GD update and the gradient EM update, we will show that the additional terms in Lemma E.2 are small.

Note that when the ground truth parameters $\theta^* = \{\mu_1^*, \mu_2^*, \ldots, \mu_K^*\}$ satisfy Assumption 14, $\theta_t^*$ also satisfies Assumption 14 for $t = O(1)$. Similarly, it is straightforward to show that when the parameters $\theta$ satisfy Assumption 15, $\theta_t = \{\mu_{1,t}, \mu_{2,t}, \ldots, \mu_{K,t}\}$ also satisfies the assumption.

We focus on the $d \leq K$ case for this proof. A similar calculation with projection onto $O(K)$ dimensional subspace of $\mu_{i,t}^*$ will give the result for $d \geq K$ case [VW04, YYS17].

Using Lemma E.6 below, we have

$$\left\|\mathbb{E}\big[w_{1,t}(X_t)(1 - w_{1,t}(X_t))(X_t - \mu_{1,t})(X_t - \mu_{1,t})^\top\big]\mu_{1,t}\right\| \leq \frac{d^2 c_r^2 B}{d^{c_r^2/1000}},$$

for any $i \in [K]$. We can simplify additional terms as

$$\left\|\sum_{i=2}^K \mathbb{E}[w_{1,t}(X_t)w_{i,t}(X_t)(X_t - \mu_{1,t})(X_t - \mu_{1,t})^\top \mu_{i,t}]\right\|$$

$$\leq \sum_{i=2}^K \mathbb{E}[\|w_{1,t}(X_t)w_{i,t}(X_t)(X_t - \mu_{1,t})(X_t - \mu_{1,t})^\top \mu_{i,t}\|]$$

$$\leq \sum_{i=2}^K \sqrt{\mathbb{E}\big[|w_{1,t}(X_t)w_{i,t}(X_t)|^2\big] \cdot \mathbb{E}\big[\|(X_t - \mu_{1,t})(X_t - \mu_{1,t})^\top \mu_{i,t}\|^2\big]}$$

$$\leq \frac{KB^2}{d^{c_r^2/2000}},$$

where in the last step we used the second part of Lemma E.5. This will allow us to prove that
$\|\mathbb{E}[w_{1,t}(X_t)(X_t - \mu_{1,t}) \sum_{i=1}^{K} w_{i,t}(X_t)\mu_{i,t}^\top(X_t - \mu_{1,t}) - w_{1,t}(X_t)(X_t - \mu_{1,t})^\top \mu_{1,t}(X_t - \mu_{1,t})]\|$
is small.

Using the expression for $\nabla_x w_{i,t}(X_t)$ from Lemma E.3, we have

$$\sum_{i=1}^{K} w_{1,t}(X_t)\nabla_x w_{i,t}(X_t)^\top \mu_{i,t}(X_t - \mu_{1,t})$$

$$= -\sum_{i=1}^{K} w_{1,t}(X_t)w_{i,t}(X_t)(1 - w_{i,t}(X_t))(X_t - \mu_{1,t})(X_t - \mu_{i,t})^\top \mu_{i,t}$$

$$+ \sum_{i=1}^{K} \sum_{j=1, j \neq i}^{K} w_{1,t}(X_t)w_{i,t}(X_t)w_{j,t}(X_t)(X_t - \mu_{1,t})(X_t - \mu_{j,t})^\top \mu_{i,t}.$$

The first term can be simplified as follows:

$$\left\| \sum_{i=1}^{K} \mathbb{E}\Big[ w_{1,t}(X_t)w_{i,t}(X_t)(1 - w_{i,t}(X_t))(X_t - \mu_{1,t})(X_t - \mu_{i,t})^\top \mu_{i,t} \Big] \right\|$$

$$\leq \sum_{i=1}^{K} \mathbb{E}\big[ \| w_{1,t}(X_t)w_{i,t}(X_t)(1 - w_{i,t}(X_t))(X_t - \mu_{1,t})(X_t - \mu_{i,t})^\top \mu_{i,t} \| \big]$$

$$\leq \sum_{i=2}^{K} \sqrt{ \mathbb{E}[w_{1,t}(X_t)^2 w_{i,t}(X_t)^2] \cdot \mathbb{E}\big[ (1 - w_{i,t}(X_t))^2 \cdot \| X_t - \mu_{1,t} \|^2 \cdot \| X_t - \mu_{i,t} \|^2 \cdot \| \mu_{i,t} \|^2 \big] }$$

$$\lesssim \frac{KB^2}{d^{c_r^2/4000}},$$

where the last inequality follows from

$$\mathbb{E}\big[ \| X_t - \mu_{1,t} \|^2 \| X_t - \mu_{i,t} \|^2 \big] \leq \sqrt{ \mathbb{E}\big[ \| X_t - \mu_{1,t} \|^4 \big] \mathbb{E}\big[ \| X_t - \mu_{i,t} \|^4 \big] } \lesssim B^2.$$

Similarly, by simplifying the second term, we get

$$\sum_{i=1}^{K} \sum_{j=1, j \neq i}^{K} \mathbb{E}\big[ \| w_{1,t}(X_t)w_{i,t}(X_t)w_{j,t}(X_t)(X_t - \mu_{1,t})(X_t - \mu_{j,t})^\top \mu_{i,t} \| \big]$$

$$\leq \sum_{i=1}^{K} \sum_{j=1, j \neq i}^{K} \sqrt{ \mathbb{E}\big[ w_{i,t}^2(X_t)w_{j,t}^2(X_t) \big] \mathbb{E}\big[ w_{1,t}^2(X_t)\|(X_t - \mu_{1,t})(X_t - \mu_{j,t})\mu_{i,t}\|^2 \big] } \lesssim \frac{K^2 B^2}{d^{c_r^2/4000}},$$

where the last inequality uses Lemma E.5. Simplifying the following term using Lemma E.5, we have

$$\left\| \mathbb{E}[w_{1,t}(X_t)\mu_{1,t} - w_{1,t}(X_t) \sum_{i=1}^{K} w_{i,t}(X_t)\mu_{i,t}] \right\|$$

$$\leq \sum_{i=2}^{K} \mathbb{E}\big[ \| w_{1,t}(X_t)w_{i,t}(X_t)\mu_{i,t} \| \big] + \sum_{i=2}^{K} \mathbb{E}\big[ \| w_{1,t}(X_t)w_{i,t}(X_t)\mu_{1,t} \| \big] \leq \frac{2KB}{d^{c_r^2/200}}.$$

Combining all the results, we obtain the theorem statement. $\qquad \square$

The above proof made use of the following two helper lemmas which follow from prior work analyzing EM for learning mixtures of Gaussians:

**Lemma E.5.** *There is some absolute constant $c_r > 0$ for which the following holds. For any $\theta = \{\mu_1, \mu_2, \ldots, \mu_K\}$ such that $\| \mu_i - \mu_i^* \| \leq \frac{c_r}{4}\sqrt{\log d}$ for all $i \in [K]$ and any $j$ such that $j \neq i$, we have*

$$\mathbb{E}_{X_t \sim \mathcal{N}(\mu_{i,t}^*, I)}[w_{j,t}(X_t)] \leq \frac{1}{d^{c_r^2/100}}.$$

*Additionally, for any $j \neq k$ such that $j \in [K]$ and $k \in [K]$, we have*

$$\mathbb{E}_{X_t}[w_{j,t}(X_t)w_{k,t}(X_t)] \leq \frac{1}{d^{c_r^2/200}}.$$

*Proof.* Using Proposition 4.1 from [SN21], for any $\theta = \{\mu_1, \mu_2, \ldots, \mu_K\}$ such that $\|\mu_i - \mu_i^*\| \leq \frac{c_r}{4}\sqrt{\log d}$ for all $i \in [K]$ and $j \neq i$, we have

$$\mathbb{E}_{X_t \sim \mathcal{N}(\mu_{i,t}^*, I)}[w_{j,t}(X_t)] \leq \frac{1}{d^{c_r^2/100}}.$$

Computing the expectation of the product of the weights $w_{j,t}$ and $w_{k,t}$ for any distinct $j, k$, we have

$$\mathbb{E}_{X_t}[w_{j,t}(X_t)w_{k,t}(X_t)] = \sum_{i=1}^{K} \frac{1}{K} \mathbb{E}_{x \sim \mathcal{N}(\mu_i^*, I)}[w_{j,t}(x)w_{k,t}(x)]$$

$$\leq \frac{1}{K} \sum_{i=1}^{K} \sqrt{\mathbb{E}_{x \sim \mathcal{N}(\mu_i^*, I)}[w_{j,t}(x)^2]\mathbb{E}_{x \sim \mathcal{N}(\mu_i^*, I)}[w_{k,t}(x)^2]}$$

$$\leq \frac{1}{d^{c_r^2/200}}$$

where the last inequality uses the fact that either $i \neq j$ or $i \neq k$ and $w_{j,t}(x)^2 \leq w_{j,t}(x) \leq 1$. $\square$

**Lemma E.6** (Lemma 4.3 of [SN21]). *Suppose $X$ is distributed according to a mixture of $K$ Gaussians with centers $\theta^* = \{\mu_1^*, \ldots, \mu_K^*\}$ as in Eq. (6). For any $\theta = \{\mu_1, \mu_2, \ldots, \mu_K\}$ such that $\|\mu_i - \mu_i^*\| \leq \frac{c_r}{4}\sqrt{\log d}$ for all $i \in [K]$, then for any distinct $i, j \in [K]$, we have*

$$\left\| \mathbb{E}_X[w_i(X, \mu)(1 - w_i(X, \mu))(X - \mu_i)(X - \mu_i)^\top] \right\|_{\mathsf{op}} \leq \frac{d^2 c_r^2}{d^{c_r^2/1000}}$$

$$\left\| \mathbb{E}_X[w_i(X, \theta)w_j(x, \theta)(X - \mu_i)(X - \mu_j)^\top] \right\|_{\mathsf{op}} \leq \frac{d^2 c_r^2}{d^{c_r^2/1000}}$$

### E.2  Closeness between population gradient descent and empirical gradient descent

In this section, we show that the population gradient descent on the DDPM objective is close to the empirical gradient descent for mixtures of $K$ Gaussians.

**Lemma E.7.** *For any $\varepsilon$ that is $\Theta(\frac{1}{\text{poly}(d)})$ and noise scale $t > t'$ where $t' \lesssim 1$, the empirical estimate of gradient descent update on the DDPM objective with the number of samples $n > n'$ concentrates well to the population gradient descent update where $n' = O(\frac{K^4 d^5 B^6}{\varepsilon^2})$. More specifically, the following inequality holds with probability at least $1 - \exp(-d^{0.99})$:*

$$\left\| \nabla_{\mu_{1,t}} \left( \frac{1}{n} \sum_{i=1}^{n} L_t(s_{\theta_t}(x_{i,0}, z_{i,t})) \right) - \nabla_{\mu_{1,t}} L_t(s_{\theta_t}) \right\| \leq \varepsilon.$$

*Proof.* Recall that the population gradient is given by

$$\nabla_{\mu_{1,t}} L_t(s_{\theta_t}) = \mathbb{E}\left[ \frac{1}{2} \nabla_{\mu_{1,t}} \left\| s_{\theta_t}(X_t) \right\|^2 + \frac{\nabla_{\mu_{1,t}} s_{\theta_t}(X_t)^\top Z_t}{\beta_t} \right],$$

where

$$\mathbb{E}\left[ \frac{1}{2} \nabla_{\mu_{1,t}} \left\| s_{\theta_t}(X_t) \right\|^2 \right] = \mathbb{E}\left[ \left( w_{1,t}(X_t)(X_t - \mu_{1,t})\mu_{1,t}^\top + w_{1,t}(X_t) \cdot \mathrm{Id} \right. \right.$$

$$\left. \left. - w_{1,t}(X_t) \sum_{i=1}^{K} w_{i,t}(X_t)(X_t - \mu_{1,t})\mu_{i,t}^\top \right) \cdot \sum_{i=1}^{K} \left( w_{i,t}(X_t)\mu_{i,t} - X_t \right) \right],$$

and

$$\mathbb{E}\left[ \nabla_{\mu_{1,t}} s_{\theta_t}(X_t)^\top Z_t \right] = \mathbb{E}\left[ \left( w_{1,t}(X_t)(X_t - \mu_{1,t})\mu_{1,t}^\top Z_t \right. \right.$$

$$\left. \left. + w_{1,t}(X_t)Z_t - w_{1,t}(X_t) \sum_{i=1}^{K} w_{i,t}(X_t)(X_t - \mu_{1,t})\mu_{i,t}^\top Z_t \right) \right] \tag{E.3}$$

We will prove that the sample estimate of each coordinate in Eq. (E.3) concentrates well around the expectation. We will prove the concentration of the first coordinate and a similar analysis holds for other coordinates. For the rest of the proof, we use $\tilde{x}_t$ to denote the first coordinate of $X_t$ and $\tilde{\mu}_{i,t}$ to indicate the first coordinate $\mu_{i,t}$. For any random variable $Y \in \mathbb{R}$, we use $\|Y\|_{\psi_1}$ to denote the sub-exponential norm of $Y$ and $\|Y\|_{\psi_2}$ to denote the sub-gaussian norm of $Y$ (See lemma B.1 for details). Using properties of a sub-Gaussian random variable from Lemma B.1, we get

$$\Big\| \sum_{j=1}^{K} w_{1,t}(X_t) w_{j,t}(X_t)(\tilde{x}_t - \tilde{\mu}_{1,t}) \mu_{1,t}^\top \mu_{j,t} \Big\|_{\psi_2}$$

$$\lesssim \sum_{j=1}^{K} \Big\| w_{1,t}(X_t) w_{j,t}(X_t)(\tilde{x}_t - \tilde{\mu}_{1,t}) \mu_{1,t}^\top \mu_{j,t} \Big\|_{\psi_2}$$

(Using sum of sub-Gaussian random variables property in Lemma B.1)

$$\lesssim \sum_{j=1}^{K} \Big\| w_{1,t}(X_t) w_{j,t}(X_t) \mu_{1,t}^\top \mu_{j,t} z \Big\|_{\psi_2} + \Big\| w_{1,t}(X_t) w_{j,t}(X_t) \mu_{1,t}^\top \mu_{j,t} (\tau - \tilde{\mu}_{1,t}) \Big\|_{\psi_2}$$

$$\lesssim KB^2 + KB^3 \lesssim KB^3, \tag{E.4}$$

where the third inequality follows by writing $\tilde{x}_t = z + \tau$ where $z \sim \mathcal{N}(0,1)$ and $\tau$ is a random variable that takes $\tilde{\mu}_{i,t}^*$ for every $i \in [K]$ with probability $\frac{1}{K}$. The fourth inequality follows from the sub-Gaussian property of a bounded random variable and the product of a sub-Gaussian random variable with bounded random variable property in Lemma B.1. Using the sum of sub-Gaussian random variable property in Lemma B.1, we have

$$\Big\| \sum_{i=1}^{K} w_{1,t}(X_t) w_{i,t}(X) \tilde{\mu}_{i,t} \Big\|_{\psi_2} \lesssim \sum_{i=1}^{K} \| w_{1,t}(X_t) w_{i,t}(X) \tilde{\mu}_{i,t} \|_{\psi_2} \lesssim KB. \tag{E.5}$$

Using properties of the sub-Gaussian random variable from Lemma B.1 in a similar way of Eq. (E.4), we have

$$\Big\| \sum_{i=1}^{K} \sum_{j=1}^{K} w_{1,t}(X_t) w_{i,t}(X_t) w_{j,t}(X_t) \mu_{i,t}^\top \mu_{j,t} (\tilde{x}_t - \tilde{\mu}_{1,t}) \Big\|_{\psi_2}$$

$$\leq \sum_{i=1}^{K} \sum_{j=1}^{K} \Big\| w_{1,t}(X_t) w_{i,t}(X_t) w_{j,t}(X_t) \mu_{i,t}^\top \mu_{j,t} (\tilde{x}_t - \tilde{\mu}_{1,t}) \Big\|_{\psi_2}$$

$$\leq \sum_{i=1}^{K} \sum_{j=1}^{K} \Big\| w_{1,t}(X_t) w_{i,t}(X_t) w_{j,t}(X_t) \mu_{i,t}^\top \mu_{j,t} z \Big\|_{\psi_2} + \Big\| w_{1,t}(X_t) w_{i,t}(X_t) w_{j,t}(X_t) \mu_{i,t}^\top \mu_{j,t} (\tau - \tilde{\mu}_{i,t}) \Big\|_{\psi_2}$$

$$\leq K^2 B^2 + K^2 B^3 \lesssim K^2 B^3 \tag{E.6}$$

We know that $\| w_{1,t}(X_t) \mu_{1,t}^\top X_t \|_{\psi_2} \leq \| \sum_{i=1}^{d} \mu_{1,t}(i) X_t(i) \|_{\psi_2} \lesssim dB^2$ and $\| \tilde{x}_t - \tilde{\mu}_{1,t} \|_{\psi_2} \lesssim B$. Using the fact that the product of two sub-Gaussian random variables is a sub-exponential random variable, we have

$$\| w_{1,t}(X_t) \mu_{1,t}^\top X_t (\tilde{x}_t - \tilde{\mu}_{1,t}) \|_{\psi_1} \leq \| \tilde{x}_t - \tilde{\mu}_{1,t} \|_{\psi_2} \| w_{1,t}(X_t) \mu_{1,t}^\top X_t \|_{\psi_2} \lesssim dB^3 \tag{E.7}$$

The sub-gaussian norm of $w_{1,t}(X_t) \tilde{x}_t$ term in the gradient is given by

$$\| w_{1,t}(X_t) \tilde{x}_t \|_{\psi_2} \leq \| X_t \|_{\psi_2} \lesssim \| Z \|_{\psi_2} + \| \tau \|_{\psi_2} \lesssim B \tag{E.8}$$

Using the property that the product of two sub-Gaussian random variables is a sub-exponential random variable, we obtain

$$\Big\| w_{1,t}(X_t)(\tilde{x}_t - \tilde{\mu}_{1,t}) \Big( \sum_{i=1}^{K} w_{i,t}(X_t) \mu_{i,t}^\top X_t \Big) \Big\|_{\psi_1}$$

$$\lesssim \| w_{1,t}(X_t)(\tilde{x}_t - \tilde{\mu}_{1,t}) \|_{\psi_2} \Big\| \Big( \sum_{i=1}^{K} w_{i,t}(X_t) \mu_{i,t}^\top X_t \Big) \Big\|_{\psi_2}$$

$$\lesssim KdB^3 \tag{E.9}$$

For any random variable $Y$, we know that $\|X\|_{\psi_1} \leq \|X\|_{\psi_2}$. Therefore, combining Eq. (E.4), (E.5), (E.6), (E.7), (E.8) and (E.9), we have

$$\|[\nabla_{\mu_{1,t}} s_{\theta_t}(X_t)^\top s_{\theta_t}(X_t)]_1 - \mathbb{E}[\nabla_{\mu_{1,t}} s_{\theta_t}(X_t)^\top s_{\theta_t}(X_t)]_1\|_{\psi_1} \lesssim \|[\nabla_{\mu_{1,t}} s_{\theta_t}(X_t)^\top s_{\theta_t}(X_t)]_1\|_{\psi_1}$$
$$\lesssim K^2 dB^3 \qquad \text{(E.10)}$$

Now, we shift our focus on obtaining the sub-exponential norm of $\nabla_{\mu_{1,t}} s_{\theta_t}(X_t)^\top Z_t$. Using $\|w_{1,t}(X_t)(\tilde{x}_t - \tilde{\mu}_{1,t})\|_{\psi_2} \lesssim B$ and $\|\mu_{1,t}^\top Z_t\|_{\psi_2} \lesssim dB$, we obtain

$$\|w_{1,t}(X_t)(\tilde{x}_t - \tilde{\mu}_{1,t})\mu_{1,t}^\top Z_t\|_{\psi_1} \leq \|w_{1,t}(X_t)(\tilde{x}_t - \tilde{\mu}_{1,t})\|_{\psi_2}\|\mu_{1,t}^\top Z_t\|_{\psi_2} \lesssim dB^2 \qquad \text{(E.11)}$$

Using Lemma B.1, we have $\|w_{1,t}(X_t)z_t\|_{\psi_2} \leq \|z_t\|_{\psi_2} \lesssim 1$. For the last term, we have

$$\left\|w_{1,t}(X_t)(\tilde{x}_t - \tilde{\mu}_{1,t})\sum_{i=1}^{K} w_{i,t}(X_t)\mu_{i,t}^\top Z_t\right\|_{\psi_1} \leq \|w_{1,t}(X_t)(\tilde{x}_t - \tilde{\mu}_{1,t})\|_{\psi_2}\left\|\sum_{i=1}^{K} w_{i,t}(X_t)\mu_{i,t}^\top Z_t\right\|_{\psi_2}$$

$$\lesssim KdB^2 \qquad \text{(E.12)}$$

Combining Eq. (E.11), (E.12), we have

$$\left\|\frac{[\nabla_{\mu_{1,t}} s_{\theta_t}(X_t)^\top Z_t]_1}{\beta_t} - \frac{\mathbb{E}[\nabla_{\mu_{1,t}} s_{\theta_t}(X_t)^\top Z_t]_1}{\beta_t}\right\|_{\psi_1} \lesssim \left\|\frac{[\nabla_{\mu_{1,t}} s_{\theta_t}(X_t)^\top Z_t]_1}{\beta_t}\right\|_{\psi_1} \lesssim \frac{KdB^2}{\beta_t} \text{(E.13)}$$

where $[\nabla_{\mu_{1,t}} s_{\theta_t}(X_t)^\top Z_t]_1$ denotes the first coordinate of $\nabla_{\mu_{1,t}} s_{\theta_t}(X_t)^\top Z_t$. Combining Eq. (E.10) and Eq. (E.13), we have

$$\left\|[\nabla_{\mu_{1,t}} L_t(s_{\theta_t}(X_t))]_1 - [\nabla_{\mu_{1,t}} L_t(s_{\theta_t})]_1\right\|_{\psi_1} \lesssim \frac{K^2 dB^3}{\beta_t}$$

For each i.i.d. sample $x_{i,t}$, the term $[\nabla_{\mu_{1,t}} L_t(s_{\theta_t}(x_{i,t}))]_1 - [\nabla_{\mu_{1,t}} L_t(s_{\theta_t})]_1$ is also independent and identically distributed. Therefore, using Lemma B.3, for any $\varepsilon$ that is $\Theta(\frac{1}{\text{poly}(d)})$, we have

$$\Pr\left[\left|\frac{1}{n}\sum_{i=1}^{n}[\nabla_{\mu_{1,t}} L_t(s_{\theta_t}(x_{i,t}))]_1 - [\nabla_{\mu_{1,t}} L_t(s_{\theta_t})]_1\right| \geq \varepsilon\right] \leq 2\exp\left(-\frac{n\varepsilon^2\beta_t^2}{K^4 d^2 B^6}\right).$$

A similar analysis will give the concentration for each coordinate. Using the union bound and rescaling $\varepsilon$ as $\frac{\varepsilon}{d}$, with probability at least $1 - 2d\exp\left(-\frac{n\varepsilon^2\beta_t^2}{K^4 d^4 B^6}\right)$, we have

$$\left\|\nabla_{\mu_{1,t}}\left(\frac{1}{n}\sum_{i=1}^{n} L_t(s_{\theta_t}(x_{i,t}))\right) - \nabla_{\mu_{1,t}} L_t(s_{\theta_t})\right\| \leq \varepsilon$$

Note that for any $t = \Omega(1)$, $\beta_t \geq c$ for some constant $c$. Therefore, choosing $n$ provided in the Lemma E.7 statement, we obtain the result. $\qquad \square$

### E.3 Proof of Theorem E.1

*Proof of Theorem E.1.* For any training iteration $h$, assume that parameters $\theta_t^{(h)}$ are such that $\left\|\mu_{i,t}^{(h)} - \mu_{i,t}^*\right\| \leq \frac{c_r}{4}\sqrt{\log d}$ we can write the update on the DDPM objective as follows:

$$\|\mu_{1,t}^{(h+1)} - \mu_{1,t}^*\| = \left\|\mu_{1,t}^{(h)} - \eta\nabla\left(\frac{1}{n}\sum_{i=1}^{n} L_t(s_{\theta_t^{(h)}}(x_{i,0}, z_{i,t}))\right) - \mu_{1,t}^*\right\|$$

$$\leq \left\|\mu_{1,t}^{(h)} + \eta\,\mathbb{E}[w_{1,t}(X_t)(X_t - \mu_{1,t}^{(h)})] - \mu_{1,t}^*\right\|$$

$$+ \eta\left\|\left(-\nabla_{\mu_{1,t}} L_t(s_{\theta_t})\right) - \mathbb{E}[w_{1,t}(X_t)(X_t - \mu_{1,t}^{(h)})]\right\|$$

$$+ \eta\left\|\left(\nabla_{\mu_{1,t}} L_t(s_{\theta_t})\right) - \nabla_{\mu_{1,t}}\left(\frac{1}{n}\sum_{i=1}^{n} L_t(s_{\theta_t^{(h)}}(x_{i,0}, z_{i,t}))\right)\right\|.$$

Using Lemma E.4, Lemma E.7 and Theorem 3.2 from [SN21], for any $\eta \in (0, K)$, we have

$$\|\mu_{1,t}^{(h+1)} - \mu_{1,t}^*\| \le \left(1 - \frac{3\eta}{8K}\right)\|\mu_{1,t}^{(h)} - \mu_{1,t}^*\| + \frac{\eta K^2 B^2}{d^{\frac{c_r^2}{4000}}} + \eta\varepsilon.$$

Choosing $\eta = \frac{2K}{3}$, $c_r$ to be sufficiently large constant and $\varepsilon$ to be $\Theta(\frac{1}{\text{poly}(d)})$, we have

$$\|\mu_{1,t}^{(h+1)} - \mu_{1,t}^*\| \le \frac{3}{4}\|\mu_{1,t}^{(h)} - \mu_{1,t}^*\| + \varepsilon$$

By assumption 15, $\|\mu_{1,t}^{(0)} - \mu_{1,t}^*\| \le O(\sqrt{\log d})$ and therefore, choosing $H$ to be $\Omega(\log(\frac{\log d}{\varepsilon}))$, we obtain the result. $\qquad\square$

# F   Additional proofs

## F.1   Proof of Lemma C.2

*Proof of Lemma C.2.* By calculating the negative gradient of the DDPM objective in Eq. (5), we obtain

$$-\nabla_{\mu_t} L_t(s_{\mu_t}) = -\mathbb{E}_{X_0,Z_t}[(\tanh(\mu_t^\top X_t)I + \tanh'(\mu_t^\top X_t)X_t\mu_t^\top)(s_{\mu_t}(X_t) + \frac{Z_t}{\beta_t})]$$

$$= -\mathbb{E}[(\tanh(\mu_t^\top X_t)I + \tanh'(\mu_t^\top X_t)X_t\mu_t^\top)(\tanh(\mu_t^\top X_t)\mu_t - X_t + \frac{Z_t}{\beta_t})]$$

$$= \mathbb{E}[-\tanh^2(\mu_t^\top X_t)\mu_t - \tanh(\mu_t^\top X_t)\tanh'(\mu_t^\top X_t)X_t\|\mu_t\|^2 + \tanh(\mu_t^\top X_t)X_t$$

$$+ \tanh'(\mu_t^\top X_t)\mu_t^\top X_t X_t - \tanh(\mu_t^\top X_t)\frac{Z_t}{\beta_t} - \tanh'(\mu_t^\top X_t)X_t\mu_t^\top\frac{Z_t}{\beta_t}]$$

By simplifying the gradient terms involving $Z_t$ by the Stein's identity as in Lemma F.1 and plugging it back in the gradient, we obtain

$$-\nabla_{\mu_t} L_t(s_{\mu_t}) = \mathbb{E}\Big[\Big(\tanh(\mu_t^\top X_t) - \tanh(\mu_t^\top X_t)\tanh'(\mu_t^\top X_t)\|\mu_t\|^2 + \tanh'(\mu_t^\top X_t)\mu_t^\top X_t\Big)X_t\Big]$$

$$- \mu_t - \mathbb{E}\Big[\tanh''(\mu_t^\top X_t)\|\mu_t\|^2 X_t\Big] - \mathbb{E}\Big[\tanh'(\mu_t^\top X_t)\mu_t\Big]$$

$$= \mathbb{E}\Big[\Big(\tanh(\mu_t^\top X_t) - 0.5\tanh''(\mu_t^\top X_t)\|\mu_t\|^2 + \tanh'(\mu_t^\top X_t)\mu_t^\top X_t\Big)X_t\Big]$$

$$- \mu_t - \mathbb{E}\Big[\tanh'(\mu_t^\top X_t)\mu_t\Big]$$

Observe that $\left(\tanh(\mu^\top x) - \frac{1}{2}\tanh''(\mu^\top x)\|\mu\|^2 + \tanh'(\mu^\top x)\mu^\top x\right)x$ and $\tanh'(\mu^\top x)$ are even functions and $X_t$ is a symmetric distribution, therefore, for any even function $f$, we can write $\mathbb{E}_{X_t}[f(X_t)] = \frac{1}{2}\mathbb{E}_{X_t\sim\mathcal{N}(\mu_t^*,\text{Id})}[f(X_t)] + \frac{1}{2}\mathbb{E}_{X_t\sim\mathcal{N}(-\mu_t^*,I)}[f(X_t)] = \mathbb{E}_{X_t\sim\mathcal{N}(\mu_t^*,\text{Id})}[f(X_t)]$. Applying this property of the even function on the gradient update, we obtain the result. $\qquad\square$

**Lemma F.1.** *When random variable $X_t = \alpha_t X_0 + \beta_t Z_t$ where $Z_t \sim \mathcal{N}(0, I), \alpha_t = \exp(-t)$ and $\beta_t = \sqrt{1 - \exp(-2t)}$, then for any $t > 0$, the following two equations hold.*

$$\mathbb{E}_{X_0,Z_t}\Big[\tanh(\mu_t^\top X_t)\frac{Z_t}{\beta_t} + \tanh^2(\mu_t^\top X_t)\mu_t\Big] = \mu_t$$

$$\mathbb{E}_{X_0,Z_t}\Big[\tanh'(\mu_t^\top X_t)\frac{\mu_t^\top Z_t}{\beta_t}X_t\Big] = \mathbb{E}_{X_0,Z_t}\Big[\tanh''(\mu_t^\top X_t)\|\mu_t\|^2 X_t + \tanh'(\mu_t^\top X_t)\mu_t\Big]$$

*Proof.* Applying Stein's lemma on the first term, we get the first equation of the statement in the Lemma.

$$\mathbb{E}_{X_0,Z_t}\Big[\tanh(\mu_t^\top X_t)\frac{Z_t}{\beta_t}\Big] = \mathbb{E}_{X_0,Z_t}\Big[\tanh(\mu_t^\top(\alpha_t X_0 + \beta_t Z_t))\frac{Z_t}{\beta_t}\Big] = \mathbb{E}_{X_0,Z_t}\Big[\tanh'(\mu_t^\top X_t)\mu_t\Big]$$

$$= \mathbb{E}_{X_0,Z_t}\Big[\Big(1 - \tanh^2(\mu_t^\top X_t)\Big)\mu_t\Big]$$

For the second term, we have

$$\mathbb{E}\left[\tanh'(\mu_t^\top X_t)\frac{\mu_t^\top Z_t}{\beta_t}X_t\right] = \mathbb{E}\left[\tanh'(\mu_t^\top X_t)\frac{\mu_t^\top Z_t}{\beta_t}\alpha_t X_0\right] + \mathbb{E}\left[\tanh'(\mu_t^\top X_t)\mu_t^\top Z_t Z_t\right]$$

$$= \sum_{i=1}^{d}\mathbb{E}\left[\alpha_t X_0 \tanh'(\mu_t^\top X_t)\frac{\mu_t(i)Z_t(i)}{\beta_t}\right] + \mathbb{E}\left[\tanh'(\mu_t^\top X_t)\mu_t\right] + \mathbb{E}\left[\tanh''(\mu_t^\top X_t)\mu_t^\top Z_t\beta_t\mu_t\right]$$

$$= \sum_{i=1}^{d}\mathbb{E}\left[\alpha_t X_0 \tanh''(\mu_t^\top X_t)\mu_t(i)\mu_t(i)\right] + \mathbb{E}\left[\tanh'(\mu_t^\top X_t)\mu_t\right] + \mathbb{E}\left[\tanh''(\mu_t^\top X_t)\mu_t^\top Z_t\beta_t\mu_t\right]$$

where the second equality follows from the Stein's lemma on the $\mathbb{E}[\tanh'(\mu_t^\top X_t)\mu_t^\top Z_t Z_t]$ and the last equality follows from the Stein's lemma on $\mathbb{E}[\alpha_t X_0 \tanh''(\mu_t^\top X_t)\mu_t(i)Z_t(i)]$. Applying Stein's inequality on the $\mathbb{E}\left[\tanh''(\mu_t^\top X_t)\mu_t^\top Z_t\beta_t\mu_t\right]$, we obtain

$$= \mathbb{E}\left[\alpha_t X_0 \tanh''(\mu_t^\top X_t)\|\mu_t\|^2\right] + \mathbb{E}\left[\tanh'(\mu_t^\top X_t)\mu_t\right] + \sum_{i=1}^{d}\beta_t\mu_t\mathbb{E}\left[\tanh'''(\mu_t^\top X_t)\mu_t(i)\beta_t\mu_t(i)\right]$$

$$= \mathbb{E}\left[X_t \tanh''(\mu_t^\top X_t)\|\mu_t\|^2\right] - \mathbb{E}\left[\beta_t Z_t \tanh''(\mu_t^\top X_t)\|\mu_t\|^2\right] + \mathbb{E}\left[\tanh'(\mu_t^\top X_t)\mu_t\right]$$

$$+ \beta_t^2\|\mu_t\|^2\,\mu_t\mathbb{E}\left[\tanh'''(\mu_t^\top X_t)\right]$$

$$= \mathbb{E}\left[X_t \tanh''(\mu_t^\top X_t)\|\mu_t\|^2\right] + \mathbb{E}\left[\tanh'(\mu_t^\top X_t)\mu_t\right].$$

$\square$

## F.2 Proof of Lemma C.8

*Proof of Lemma C.8.* Recall that the gradient update for any $\mu_t^*$ is given by

$$-\nabla_{\mu_t^*}L_t(s_{\mu_t^*}) = G(\mu_t^*, \mu_t^*) + \eta\mathbb{E}_{x\sim\mathcal{N}(\mu_t^*,\mathrm{Id})}[\tanh(\mu_t^{*\top}x)x] - \eta\mu_t^* \tag{F.1}$$

We know that $\mathbb{E}_{x\sim\mathcal{N}(\mu_t^*,\mathrm{Id})}[\tanh(\mu_t^{*\top}x)x] = \mu^*$ (Eq.(2.1) of [DTZ17]) and $\nabla_{\mu_t^*}L_t(s_{\mu_t^*}) = 0$ because $\mu_t^*$ is a stationary point of the regression objective of diffusion model. This implies that $G(\mu_t^*, \mu_t^*) = 0$ for any $\mu_t^*$.

Note that this proof only talks about 1D case therefore, for the purpose of this proof, we use $a$ to denote $\mu$ and $b$ to denote $\mu^*$. In 1D, using Mean value theorem, we have

$$\frac{G(a,b) - G(a,a)}{b - a} = \frac{dG(a,\xi)}{d\xi} \text{ for some } \xi \in [a,b] \text{ (if } a < b) \tag{F.2}$$

Using the fact that $G(a,a) = 0$ in Eq. (F.2), we have

$$|G(a,b)| = \left|\frac{dG(a,\xi)}{d\xi}\right||b - a|$$

Observe that it suffices to prove $\left|\frac{dG(a,\xi)}{d\xi}\right| \le 0.01$ to obtain the lemma. By computing the gradient of $G$, we obtain

$$\frac{dG(a,\xi)}{d\xi} = \eta\mathbb{E}_{x\sim\mathcal{N}(\xi,1)}\left[2\tanh'(ax)ax + \tanh''(ax)\left(\frac{-3a^2}{2} + a^2x^2\right) - \frac{1}{2}a^3x\tanh'''(ax)\right]$$

For the first term, we have

$$\mathbb{E}_{x\sim\mathcal{N}(\xi,I)}[\tanh'(ax)ax] = \frac{1}{\sqrt{2\pi}}\int_{-\infty}^{\infty}\tanh'(ax)axe^{-\frac{(x-\xi)^2}{2}}dx$$

$$= \frac{1}{\sqrt{2\pi}}\int_{0}^{\infty}\tanh'(ax)ax\left(e^{-\frac{(x-\xi)^2}{2}}-e^{-\frac{(x+\xi)^2}{2}}\right)dx$$

$$\leq \frac{1}{\sqrt{2\pi}}\int_{0}^{\infty}e^{-ax}axe^{-\frac{(x-\xi)^2}{2}}dx$$

$$\leq \frac{ae^{\frac{a^2-2a\xi}{2}}}{\sqrt{2\pi}}\int_{0}^{\infty}xe^{-\frac{(x-\xi+a)^2}{2}}dx$$

$$\leq ae^{\frac{a^2-2a\xi}{2}}(\sqrt{\frac{2}{\pi}}e^{-\frac{(\xi-a)^2}{2}}+(\xi-a)\mathrm{erf}\left(\frac{\xi-a}{\sqrt{2}}\right))$$

$$\leq ae^{-\frac{\xi^2}{2}}+a|\xi-a|\,e^{\frac{-2a(\xi-a)-a^2}{2}}$$

Using Lemma 1 of [DTZ17], we know that $\mathbb{E}_{x\sim\mathcal{N}(\xi,I)}[\tanh'(ax)ax] > 0$. Therefore, we have

$$\left|\mathbb{E}_{x\sim\mathcal{N}(\xi,I)}[\tanh'(ax)ax]\right| \leq ae^{-\frac{\xi^2}{2}}+a|\xi-a|\,e^{\frac{-2a(\xi-a)-a^2}{2}}$$

For the second term, we have

$$\mathbb{E}_{x\sim\mathcal{N}(\xi,1)}[\tanh''(ax)(-\frac{3a^2}{2}+a^2x^2)]$$

$$= \frac{1}{\sqrt{2\pi}}\int_{0}^{\infty}a^2\tanh''(ax)(-\frac{3}{2}+x^2)\left(\exp(-\frac{(x-\xi)^2}{2})-\exp(-\frac{(x+\xi)^2}{2})\right)dx$$

$$\leq \frac{1}{\sqrt{2\pi}}\int_{0}^{\sqrt{\frac{3}{2}}}a^2e^{-2ax}(\frac{3}{2}-x^2)\exp(-\frac{(x-\xi)^2}{2})dx$$

$$\leq \frac{3}{\sqrt{2\pi}}a^2\exp(-\frac{a^2}{16})$$

Assuming $a\geq\sqrt{6}$, then when $\xi\geq a\geq\sqrt{6}$, we have $\exp(-\frac{(x-\xi)^2}{2})\leq\exp(-\frac{a^2}{4})$ and when $\xi\leq a$, using $\xi\geq\frac{3a}{4}$, we have $\exp(-\frac{(x-\xi)^2}{2})\leq\exp(-\frac{a^2}{16})$. For the lower bound, we have

$$\mathbb{E}_{x\sim\mathcal{N}(\xi,1)}[\tanh''(ax)(-\frac{3a^2}{2}+a^2x^2)]$$

$$= \frac{1}{\sqrt{2\pi}}\int_{0}^{\infty}\tanh''(ax)(-\frac{3a^2}{2}+a^2x^2)\left(\exp(-\frac{(x-\xi)^2}{2})-\exp(-\frac{(x+\xi)^2}{2})\right)dx$$

$$\geq \frac{1}{\sqrt{2\pi}}\int_{\sqrt{\frac{3}{2}}}^{\infty}\tanh''(ax)(-\frac{3a^2}{2}+a^2x^2)\left(\exp(-\frac{(x-\xi)^2}{2})-\exp(-\frac{(x+\xi)^2}{2})\right)dx$$

$$\geq \frac{1}{\sqrt{2\pi}}\int_{\sqrt{\frac{3}{2}}}^{\infty}\tanh''(ax)a^2x^2\left(\exp(-\frac{(x-\xi)^2}{2})-\exp(-\frac{(x+\xi)^2}{2})\right)dx$$

$$\geq -\frac{8a^2}{\sqrt{2\pi}}\int_{\sqrt{\frac{3}{2}}}^{\infty}e^{-2ax}x^2\left(\exp(-\frac{(x-\xi)^2}{2})-\exp(-\frac{(x+\xi)^2}{2})\right)dx$$

$$\geq -\frac{8a^2e^{-\sqrt{6}a}}{\sqrt{2\pi}}\int_{\sqrt{\frac{3}{2}}}^{\infty}x^2\exp(-\frac{(x-\xi)^2}{2})dx \geq -8a^2e^{-\sqrt{6}a}$$

Using upper bound and lower bound, we have

$$\left|\mathbb{E}_{x\sim\mathcal{N}(\xi,1)}[\tanh''(ax)a^2(-\frac{3}{2}+x^2)]\right| \leq 8a^2e^{-\sqrt{6}a}$$

For the third term, we have

$$
\left| \mathbb{E}_{x \sim \mathcal{N}(\xi,1)} \left[ \frac{a^3 x}{2} \tanh'''(ax) \right] \right|
$$

$$
= \left| \frac{1}{32\sqrt{2\pi}} \int_0^\infty a^3 x \sigma(2ax)(1 - \sigma(2ax)) \left( 1 - 6\sigma(2ax)(1 - \sigma(2ax)) \right) \left( \exp\left( -\frac{(x-\xi)^2}{2} \right) \right. \right.
$$

$$
\left. \left. - \exp\left( -\frac{(x+\xi)^2}{2} \right) \right) dx \right|
$$

$$
\leq \left| \frac{3a^3}{16\sqrt{2\pi}} \int_0^\infty x \sigma^2(2ax)(1 - \sigma(2ax))^2 \left( \exp\left( -\frac{(x-\xi)^2}{2} \right) - \exp\left( -\frac{(x+\xi)^2}{2} \right) \right) dx \right|
$$

$$
\leq \frac{3a^3}{16\sqrt{2\pi}} \int_0^\infty x e^{-ax} \exp\left( -\frac{(x-\xi)^2}{2} \right) dx
$$

$$
\leq \frac{a^3}{10} e^{-\frac{\xi^2}{2}} + \frac{a^3}{10} |\xi - a| \, e^{\frac{-2a(\xi-a)-a^2}{2}}.
$$

We can lower bound the third term as follows:

$$
\mathbb{E}_{x \sim \mathcal{N}(\xi,1)} \left[ \frac{a^3 x}{2} \tanh'''(ax) \right]
$$

$$
\geq \frac{1}{2\sqrt{2\pi}} \int_0^c a^3 x \tanh'''(ax) \left( \exp\left( -\frac{(x+\xi)^2}{2} \right) - \exp\left( -\frac{(x-\xi)^2}{2} \right) \right) dx
$$

$$
\geq \frac{a^3}{2\sqrt{2\pi}} \int_0^c x \exp\left( -\frac{(x-\xi)^2}{2} \right) \left( \exp(-2\xi x) - 1 \right) dx
$$

$$
\geq -\frac{a^3 \xi}{\sqrt{2\pi}} \int_0^c x^2 \exp\left( -\frac{(x-\xi)^2}{2} \right) dx \geq -\frac{\xi \exp(-\frac{\xi^2}{4})}{\sqrt{2\pi}}
$$

Using all the bounds, we have

$$
\left| \frac{dG(a,\xi)}{d\xi} \right| \leq \frac{a^3}{10} e^{-\frac{\xi^2}{2}} + \frac{a^3}{10} |\xi - a| \, e^{\frac{-2a(\xi-a)-a^2}{2}} + 8a^2 e^{-\sqrt{6}a} + a e^{-\frac{\xi^2}{2}} + a|\xi - a| \, e^{\frac{-2a(\xi-a)-a^2}{2}}
$$

When $\xi \geq a$ and $a \geq c$ for some sufficiently large constant $c$ (for example, $c = 25$), then, we have

$$
\left| \frac{dG(a,\xi)}{d\xi} \right| \leq \frac{a^3}{10} e^{-\frac{a^2}{2}} + \frac{a^3}{10} |\xi - a| \, e^{\frac{-a^2}{2}} + 8a^2 e^{-\sqrt{6}a} + a e^{-\frac{a^2}{2}} + a|\xi - a| \, e^{\frac{-a^2}{2}} \leq 0.01
$$

When $\frac{3a}{4} \leq \xi \leq a$ and $a > c$ for sufficiently large constant $c$ (for example, $c = 25$), we have

$$
\left| \frac{dG(a,\xi)}{d\xi} \right| \leq \frac{a^3}{10} e^{-\frac{9a^2}{32}} + \frac{a^4}{40} e^{\frac{-a^2}{4}} + 8a^2 e^{-\sqrt{6}a} + a e^{-\frac{a^2}{2}} + \frac{a^2}{4} e^{\frac{-a^2}{4}} \leq 0.01
$$

Pluggint the bound on $|\frac{dG(a,\xi)}{d\xi}|$ in Eq. (F.1), we obtain the final result. □

### F.3 Proof of Lemma C.10

*Proof of Lemma C.10.* We will prove this by induction. For $h = 0$, this is true because the algorithm initializes the gradient descent on the low noise regime with the output of gradient descent on the high noise regime, and the output is guaranteed to have $\langle \hat{\mu}_t^{(0)}, \hat{\mu}_t^* \rangle$ to be $\Omega(1)$ and by assumption $\|\mu_t^*\| > c'$, therefore $\|\mu_t^{(0)}\| \in [c, \frac{4\langle \hat{\mu}_t^{(0)}, \mu_t^* \rangle}{3}]$.

Suppose $\|\mu_t^{(h)}\| \in [c, \frac{4\langle \hat{\mu}_t^{(h)}, \mu_t^* \rangle}{3}]$, then we know that $\|\mu_t^{(h+1)} - \mu_t^*\| < \|\mu_t^{(h)} - \mu_t^*\|$. To prove $\|\mu_t^{(h+1)}\| \in [c, \frac{4\langle \hat{\mu}_t^{(h+1)}, \mu_t^* \rangle}{3}]$, first we will prove that $\langle \hat{\mu}_t^{(h)}, \mu_t^{(r+1)} \rangle \in [c, \frac{6\langle \hat{\mu}_t^{(h)}, \mu_t^* \rangle}{5}]$. Note that the update in the direction of $\langle \hat{\mu}_t, \mu_t \rangle$ works like 1D. Therefore, we have a contraction for it as follows.

$$
\left| \langle \hat{\mu}_t^{(h)}, \mu_t^{(h+1)} \rangle - \langle \hat{\mu}_t^{(h)}, \mu_t^* \rangle \right| < \left| \langle \hat{\mu}_t^{(h)}, \mu_t^{(h)} \rangle - \langle \hat{\mu}_t, \mu_t^* \rangle \right|
$$

If $\|\mu_t^{(h)}\| \leq \langle \hat{\mu}_t^{(h)}, \mu_t^* \rangle$, then using Lemma F.4, we know $\langle \hat{\mu}_t^{(h)}, \mu_t^{(h+1)} \rangle \leq \frac{6\langle \hat{\mu}_t^{(h)}, \mu_t^* \rangle}{5}$ and $\langle \hat{\mu}_t^{(h)}, \mu_t^{(h+1)} \rangle \geq \|\mu_t^{(h)}\| \geq c$ because of the contraction. If $\|\mu_t^{(h)}\| \geq \langle \hat{\mu}_t^{(h)}, \mu_t^* \rangle$ and $\langle \hat{\mu}_t^{(h)}, \mu_t^{(h+1)} \rangle \geq \langle \hat{\mu}_t^{(h)}, \mu_t^* \rangle$, then $\langle \hat{\mu}_t^{(h)}, \mu_t^{(h+1)} \rangle \leq \|\mu_t^{(h)}\|$ because of the contraction. If $\|\mu_t^{(h)}\| \geq \langle \hat{\mu}_t^{(h)}, \mu_t^* \rangle$ and $\langle \hat{\mu}_t^{(h+1)}, \mu_t^{(h)} \rangle \leq \langle \hat{\mu}_t^{(h)}, \mu_t^* \rangle$, then using $\langle \hat{\mu}_t^{(h+1)}, \mu_t^{(h)} \rangle \geq \|\mu_t^{(h)}\| - \left| U(\langle \hat{\mu}_t^{(h)}, \mu_t^{(h)} \rangle, \langle \hat{\mu}_t^{(h)}, \mu_t^* \rangle) \right| \geq \frac{4\langle \hat{\mu}_t^{(h)}, \mu_t^* \rangle}{5} \geq \frac{4\langle \hat{\mu}_t^{(0)}, \mu_t^* \rangle}{5} \geq c$ from Lemma F.2, we get the result that $\langle \hat{\mu}_t^{(h)}, \mu_t^{(h+1)} \rangle \in [c, \frac{6\langle \hat{\mu}_t^{(h)}, \mu_t^* \rangle}{5}]$. Now, using Lemma F.2, we get

$$\langle \hat{\mu}_t^{(h)}, \mu_t^{(h+1)} \rangle \in [c, \frac{6\langle \hat{\mu}_t^{(h)}, \mu_t^* \rangle}{5}] \implies \|\mu_t^{(h+1)}\| \in [\frac{c}{\cos \alpha_h}, \frac{6\|\mu_t^*\| \cos \beta_h}{5\cos \alpha_h}]$$

$$\implies \|\mu_t^{(h+1)}\| \in [c, \frac{4\|\mu_t^*\| \cos \beta_{h+1}}{3}]$$

$$\implies \|\mu_t^{(h+1)}\| \in [c, \frac{4\langle \hat{\mu}_t^{(h+1)}, \mu_t^* \rangle}{3}] \qquad \square$$

**Lemma F.2.** *Suppose the angle between $\mu^{(r)}$ and $\mu^*$ is $\beta_r$ and $\alpha_r$ is the angle between $\mu^{(r)}$ and $\mu^{(r+1)}$ and assume the contraction is true at time $r$. Assume that $\beta_0 \in (0, \frac{\pi}{2})$. Then:*

$$\alpha_r \in (0, \pi/2) \ \forall r \qquad and \qquad \cos \beta_r \leq \cos \beta_{r+1}$$

*which implies that*

$$\cos \beta_r \leq \cos \beta_{r+1} \ \forall r \implies \langle \hat{\mu}^{(r)}, \mu^* \rangle \geq \langle \hat{\mu}^{(0)}, \mu^* \rangle$$

*Proof.* First, we will prove that if $\beta_r \in (0, \frac{\pi}{2})$ and $\|\mu^{(r)}\| \in [c, \frac{4\langle \hat{\mu}_t^{(r)}, \mu_t^* \rangle}{3}]$, then $\alpha_r \in (0, \beta_r)$ for any $r$. We denote $\alpha_r > 0$ if $\mu^{(r)}$ moves towards $\mu^{(r)\perp}$ and hence towards $\mu^*$. The following simple observation of $\langle \hat{\mu}^{(r)\perp}, \mu^{(r+1)} \rangle \geq 0$ proves that $\alpha_r > 0$.

$\langle \hat{\mu}^{(r)\perp}, \mu^{(r+1)} \rangle$

$= \mathbb{E}_{x \sim \mathcal{N}(\mu^*, 1)} \left[ \eta \left( \tanh(\mu^{(r)\top} x) - \frac{1}{2} \tanh''(\mu^{(r)\top} x)\|\mu^{(r)}\|^2 + \tanh'(\mu^{(r)\top} x)\mu^{(r)\top} x \right) \cdot \langle \hat{\mu}^{(r)\perp}, x \rangle \right]$

$= \mathbb{E}_{x \sim \mathcal{N}(0,1)} \Big[ \eta \Big( \tanh(\mu^{(r)\top}(x + \mu^*)) - \frac{1}{2} \tanh''(\mu^{(r)\top}(x + \mu^*))\|\mu^{(r)}\|^2$
$\qquad + \tanh'(\mu^{(r)\top}(x + \mu^*))\mu^{(r)\top}(x + \mu^*) \Big) \cdot \langle \hat{\mu}^{(r)\perp}, (x + \mu^*) \rangle \Big]$

$= \mathbb{E}_{\alpha_1, \alpha_2 \sim \mathcal{N}(\langle \hat{\mu}^{(r)}, \mu^* \rangle, 1)} \Big[ \eta \Big( \tanh(\|\mu^{(r)}\|\alpha_1) - \frac{1}{2} \tanh''(\|\mu^{(r)}\|\alpha_1)\|\mu^{(r)}\|^2$
$\qquad + \tanh'(\|\mu^{(r)}\|\alpha_1)\|\mu^{(r)}\|\alpha_1 \Big) (\alpha_2 + \langle \hat{\mu}^{(r)\perp}, \mu^* \rangle) \Big]$

$= \mathbb{E}_{\alpha_1, \alpha_2 \sim \mathcal{N}(\langle \hat{\mu}^{(r)}, \mu^* \rangle, 1)} \Big[ \eta \Big( \tanh(\|\mu^{(r)}\|\alpha_1) - \frac{1}{2} \tanh''(\|\mu^{(r)}\|\alpha_1)\|\mu^{(r)}\|^2$
$\qquad + \tanh'(\|\mu^{(r)}\|\alpha_1)\|\mu^{(r)}\|\alpha_1 \Big) \cdot \langle \hat{\mu}^{(r)\perp}, \mu^* \rangle \Big] > 0 \,,$

where in the last step we used the fact that $\langle \hat{\mu}^{(r)}, \mu^* \rangle > 0$ and $\langle \hat{\mu}^{(r)\perp}, \mu^* \rangle > 0$.

Now, we will prove that $\cot \alpha_r > \cot \beta_r$ which will prove that $\alpha_r \in (0, \beta_r)$. Note that

$$\cot \alpha_r = \frac{\langle \hat{\mu}^{(r)}, \mu^{(r+1)} \rangle}{\langle \hat{\mu}^{(r)\perp}, \mu^{(r+1)} \rangle} \qquad \text{where}$$

$$\langle \hat{\mu}^{(r)}, \mu^{(r+1)} \rangle = (1 - \eta)\|\mu^{(r)}\| + \eta \mathbb{E}_{\alpha_1 \sim \mathcal{N}(\hat{\mu}^{(r)\top}\mu^*, 1)}[\tanh(\|\mu^{(r)}\|\alpha_1)\alpha_1]$$

$$+ \eta \mathbb{E}_{\alpha_1 \sim \mathcal{N}(\hat{\mu}^{(r)\top}\mu^*, 1)}[-\frac{1}{2}\tanh''(\|\mu^{(r)}\|\alpha_1)\|\mu^{(r)}\|^2\alpha_1 + \tanh'(\|\mu^{(r)}\|\alpha_1)\|\mu^{(r)}\|\alpha_1^2$$

$$- \tanh'(\|\mu^{(r)}\|\alpha_1)\|\mu^{(r)}\|]$$

$$\langle \hat{\mu}^{(r)\perp}, \mu^{(r+1)} \rangle = \eta \langle \hat{\mu}^{(r)\perp}, \mu^* \rangle \mathbb{E}_{\alpha_1 \sim \mathcal{N}(\hat{\mu}^{(r)\top}\mu^*, 1)}[\tanh(\|\mu^{(r)}\|\alpha_1) - \frac{1}{2}\tanh''( \|\mu^{(r)}\|\alpha_1)\|\mu^{(r)}\|^2$$

$$+ \tanh'( \|\mu^{(r)}\|\alpha_1)\|\mu^{(r)}\|\alpha_1]$$

$$\text{and} \quad \cot \beta_r = \frac{\langle \hat{\mu}^{(r)}, \mu^* \rangle}{\langle \hat{\mu}^{(r)\perp}, \mu^* \rangle}$$

Observe the fact that to prove $\frac{a+c'}{b+c} - \frac{a}{b} > 0$, it is sufficient to prove $c' > \frac{ac}{b}$ for $b, c > 0$. Using this observation, to prove $\cot \alpha_r > \cot \beta_r$, it is sufficient to prove

$$\left(1 - \eta - \eta \mathbb{E}[\tanh'(\|\mu^{(r)}\|x)]\right)\|\mu^{(r)}\| + \eta \mathbb{E}_x\left[-\frac{1}{2}\tanh''(\|\mu^{(r)}\|x)\|\mu^{(r)}\|^2(x - \langle \hat{\mu}^{(r)}, \mu^* \rangle)\right.$$

$$\left. + \tanh'(\|\mu^{(r)}\|x)(x^2 - \langle \hat{\mu}^{(r)}, \mu^* \rangle x) + \tanh(\|\mu^{(r)}\|x)(x - \langle \hat{\mu}^{(r)}, \mu^* \rangle))\right] > 0,$$

where the expectation is wrt $\mathcal{N}(\langle \mu^{(r)}, \mu^* \rangle, 1)$. Lemma F.3 shows that this is indeed true. $\qquad \square$

**Lemma F.3.** *For any $\eta = \frac{1}{20}$, assuming $a \in [30, \frac{4b}{3}]$, we have*

$$(1 - \eta - \eta \mathbb{E}_{x \sim \mathcal{N}(b, 1)}[\tanh'(ax)])a$$

$$+ \eta \, \mathbb{E}_{x \sim \mathcal{N}(b, 1)}\left[-\frac{1}{2}\tanh''(ax)a^2(x - b)\tanh'(ax)(x^2 - bx) + \tanh(ax)(x - b)\right] > 0.$$

*Proof.* First, we will find the upper bound on $\mathbb{E}[\tanh''(ax)(x - b)]$.

$$\mathbb{E}[\tanh''(ax)(x - b)] = \int_{-\infty}^{\infty} \tanh''(ax)(x - b)\exp\left(-\frac{(x-b)^2}{2}\right)dx$$

$$\leq \int_0^b \tanh''(ax)(x - b)\exp\left(-\frac{(x-b)^2}{2}\right)dx$$

$$\leq \int_0^b \tanh''(ax)x\exp\left(-\frac{(x-b)^2}{2}\right)dx$$

$$\leq \int_0^b \exp(-ax)x\exp\left(-\frac{(x-b)^2}{2}\right)dx$$

$$\leq \exp\left(\frac{a^2 - 2ab}{2}\right)\int_0^b x\exp\left(-\frac{(x-b)^2 + 2a(x-b) + a^2}{2}\right)dx$$

$$\leq \exp(\frac{a^2 - 2ab}{2})\int_0^\infty x\left[\exp\left(-\frac{(x-b+a)^2}{2}\right) + \exp\left(-\frac{(x+b-a)^2}{2}\right)\right]dx$$

$$\leq \exp(-b^2/2) + |a - b| \cdot \exp\left(\frac{a^2 - 2ab}{2}\right).$$

Now, for the second term, we have

$$\mathbb{E}_{x\sim\mathcal{N}(b,1)}[\tanh'(ax)(x^2 - bx)]$$

$$= \int_{-\infty}^{\infty} \tanh'(ax)x(x - b)\exp\Big(-\frac{(x - b)^2}{2}\Big)dx$$

$$\geq -b\int_{0}^{b} xe^{-ax}\exp\Big(-\frac{(x - b)^2}{2}\Big)dx$$

$$\geq -b\exp\Big(\frac{a^2 - 2ab}{2}\Big)\int_{0}^{\infty} x\Big[\exp\Big(-\frac{(x - b + a)^2}{2}\Big) + \exp\Big(-\frac{(x + b - a)^2}{2}\Big)\Big]dx$$

$$\geq -b\exp(-b^2/2) - b|a - b| \cdot \exp\Big(\frac{a^2 - 2ab}{2}\Big)$$

We can rewrite the last term as $\mathbb{E}_{x\sim\mathcal{N}(0,1)}[\tanh(a(x + b))x]$. Using the fact that $\tanh(a(x + b)) > \tanh(a(-x + b))$, we get that $\mathbb{E}_{x\sim\mathcal{N}(0,1)}[\tanh(a(x + b))x] > 0$. Finally, using the upper bound on $\mathbb{E}[\tanh'(ax)]$, we get the following lower bound.

$$(1 - \eta - \eta\,\mathbb{E}_{x\sim\mathcal{N}(b,1)}[\tanh'(ax)])\,a + \eta\mathbb{E}_{x\sim\mathcal{N}(b,1)}\Big[-\frac{1}{2}\tanh''(ax)a^2(x - b) + \tanh'(ax)(x^2 - bx)\Big]$$

$$\geq \frac{a}{20}(19 - 4e^{\frac{a^2 - 2ab}{2}}) + \frac{1}{20}\Big(-\frac{a^2}{2}\Big[\exp(-b^2/2) + |a - b|\exp\Big(\frac{a^2 - 2ab}{2}\Big)\Big]$$

$$- b\exp(-b^2/2) - b|a - b|\exp(\frac{a^2 - 2ab}{2})\Big) \geq 1. \qquad \square$$

**Lemma F.4.** *For any $a, b > 0$ and $a \in [30, \frac{4b}{3}]$, the following holds. Define*

$$U(a, b) \triangleq \eta\mathbb{E}_{x\sim\mathcal{N}(b,1)}\Big[\Big(\tanh(ax) - \frac{1}{2}\tanh''(ax)a^2 + \tanh'(ax)ax\Big)x\Big] - \eta\mathbb{E}_{x\sim\mathcal{N}(b,1)}\big[\tanh'(ax)a\big] - \eta a.$$

*When the learning rate $\eta = \frac{1}{20}$, is given by, we have*

$$|U(a, b)| \leq \frac{a + b}{10}$$

*Proof.* We upper bound each term in $U(a, b)$ and they apply triangle inequality to get the result. We start with $|\mathbb{E}_{x\sim\mathcal{N}(b,1)}\big[\tanh''(ax)a^2x\big]|$:

$$-\mathbb{E}_{x\sim\mathcal{N}(b,1)}\Big[\tanh''(ax)a^2x\Big] = \frac{a^2}{8\sqrt{2\pi}}\int_{0}^{\infty} x\sigma(2ax)(1 - \sigma(2ax))(2\sigma(2ax) - 1)\Big(e^{-\frac{(x - b)^2}{2}} + e^{-\frac{(x + b)^2}{2}}\Big)dx$$

$$\leq \frac{a^2}{4\sqrt{2\pi}}\int_{0}^{\infty} xe^{-2ax}e^{-\frac{(x - b)^2}{2}}dx$$

$$\leq \frac{a^2}{4\sqrt{2\pi}}\int_{0}^{\infty} e^{-ax}xe^{-\frac{(x - b)^2}{2}}dx$$

$$\leq \frac{a^2}{2}e^{-\frac{b^2}{2}} + \frac{a^2}{2}|b - a|e^{-\frac{2a(b-a)-a^2}{2}}$$

$$\mathbb{E}_{x\sim\mathcal{N}(b,1)}[\tanh'(ax)ax^2] = \frac{1}{\sqrt{2\pi}}\int_{0}^{\infty} \tanh'(ax)ax^2\Big(e^{-\frac{(x - b)^2}{2}} + e^{-\frac{(x + b)^2}{2}}\Big)dx$$

$$\leq a\int_{0}^{\infty} e^{-ax}x^2e^{-\frac{(x - b)^2}{2}}dx$$

$$\leq ae^{\frac{a^2 - 2ab}{2}}\int_{0}^{\infty} x^2e^{-\frac{(x - b + a)^2}{2}}dx$$

$$\leq 2a(a - b)^2e^{\frac{a^2 - 2ab}{2}}$$

$$-\mathbb{E}_{x\sim\mathcal{N}(b,1)}[a\tanh'(ax)] = -\frac{a}{\sqrt{2\pi}}\int_0^\infty \tanh'(ax)\left(e^{-\frac{(x-b)^2}{2}} + e^{-\frac{(x+b)^2}{2}}\right)dx$$

$$\geq -a\int_0^\infty e^{-ax}e^{-\frac{(x-b)^2}{2}}dx$$

$$\geq -ae^{\frac{a^2-2ab}{2}}\int_0^\infty e^{-\frac{(x-b+a)^2}{2}}dx$$

$$\geq -4ae^{\frac{a^2-2ab}{2}}.$$

Now, using the fact that $\tanh'(x)$ and $-\tanh''(x)x$ are always positive, we have the following upper bound.

$$\left|U(a,b)\right| \leq \eta\left|\mathbb{E}_{x\sim\mathcal{N}(b,1)}\left[\left(\tanh(ax) - \frac{1}{2}\tanh''(ax)a^2 + \tanh'(ax)ax\right)\cdot x\right]\right|$$

$$+ \eta|a| + \eta\left| - \mathbb{E}_{x\sim\mathcal{N}(b,I)}\left[\tanh'(ax)a\right]\right|$$

$$\leq \eta\left(2b + a + \frac{a^2}{2}e^{-\frac{b^2}{2}} + \frac{a^2}{2}|b-a|e^{\frac{-2a(b-a)-a^2}{2}} + 2a(b-a)^2e^{\frac{a^2-2ab}{2}} + 2ae^{\frac{a^2-2ab}{2}}\right)$$

If $b \geq a$ and $a \geq 30$, then we have

$$\left|U(a,b)\right| \leq \eta\left(2b + a + 0.1\right)$$

If $b \leq a \leq \frac{4b}{3}$ and $a \geq 30$, then

$$\left|U(a,b)\right| \leq \eta\left(2b + a + 0.1\right)$$

Using $\eta = 1/20$ and for any $a > 30$, we have

$$\left|U(a,b)\right| \leq \frac{a+b}{10}.$$

$\square$

### F.4 Additional proofs for mixtures of two Gaussians

**Lemma F.5.** *Suppose $a, b > 0$ satisfy $a \in [30, \frac{4b}{3}]$, then the following inequality holds:*

$$|\mathbb{E}_{x\sim\mathcal{N}(b,1)}[-0.5\tanh''(ax)a^2 + \tanh'(ax)ax]| \leq 0.01$$

*Proof.* We first show that $\mathbb{E}_{x\sim\mathcal{N}(b,1)}[-0.5\tanh''(ax)a^2] > 0$ for any $a, b > 0$.

$$\mathbb{E}_{x\sim\mathcal{N}(b,1)}[-0.5\tanh''(ax)a^2] = -0.5a^2\int_{-\infty}^\infty \tanh''(ax)\exp(-0.5(x-b)^2)dx$$

$$= -0.5a^2\int_0^\infty \tanh''(ax)(\exp(-0.5(x-b)^2) - \exp(-0.5(x+b)^2))dx > 0$$

where the last inequality follows from $\exp(-0.5(x-b)^2) > \exp(-0.5(x+b)^2)$ and $\tanh''(ax) < 0$ for $x > 0$. We can upper bound $\mathbb{E}_{x\sim\mathcal{N}(b,1)}[-0.5\tanh''(ax)a^2]$ as follows:

$$\mathbb{E}_{x\sim\mathcal{N}(b,1)}[-\frac{1}{2}\tanh''(ax)a^2] \leq -\frac{1}{2}a^2\int_0^\infty \tanh''(ax)\exp(-\frac{1}{2}(x-b)^2)dx$$

$$\leq a^2\int_0^\infty \exp(-ax)\exp(-\frac{1}{2}(x-b)^2)dx$$

$$\leq a^2\exp(\frac{1}{2}(a^2-2ab))\int_0^\infty \exp(-\frac{1}{2}(x-b+a)^2)dx$$

$$\leq a^2\exp(\frac{1}{2}(a^2-2ab))$$

When $a \leq b$, by writing $a^2 - 2ab = -2a(b-a) - a^2 \leq -a^2$, we have $\mathbb{E}[-\frac{1}{2}\tanh''(ax)a^2] \leq 0.005$ for $a \geq 30$. When $a \in [b, \frac{4b}{3}]$, $a^2 - 2ab = \leq -\frac{2b^2}{9}$, we have $|\mathbb{E}[-\frac{1}{2}\tanh''(ax)a^2]| \leq 0.005$. Similar to the $\mathbb{E}_{x\sim\mathcal{N}(b,1)}[-\frac{1}{2}\tanh''(ax)a^2]$, we prove $\mathbb{E}_{x\sim\mathcal{N}(b,1)}[\tanh'(ax)ax] > 0$ and $\mathbb{E}_{x\sim\mathcal{N}(b,1)}[\tanh'(ax)ax] < 0.005$. Combining bounds for $|\mathbb{E}[\tanh'(ax)ax]|$ and $|\mathbb{E}[-\frac{1}{2}\tanh''(ax)a^2]|$ using triangle inequality, we obtain the result. $\square$

# G  Experiments

In this section, we perform two sets of experiments to understand the role of large and small noise regimes in the training of mixtures of two Gaussians. Mainly, we want to answer the following questions:

1. Does the large noise regime helps in achieving the warm start required for the small noise regime (as predicted by theory)?   *Answer:* Yes

2. Does the large noise scale regime learn the direction of the true mean vector despite having a high amount of noise?   *Answer:* Yes

**Setup.**   The task in both experiments is to learn the true parameters of zero-centered mixtures of two Gaussians in 100 dimensions. We use $\mu^*$ and $-\mu^*$ to denote the mean vectors of two mixtures. Each element of the $\mu^*$ vector is sampled uniformly from $[0, 1]$. We use stochastic gradient descent (SGD) with batch size 128 and learning rate 0.001 for the training. We use $t = 0.01$ for the small noise scale training and $t \in \{1, 1.1, 1.2\}$ for the large noise scale. All results are averaged over 5 independent runs.

**Results.**   To answer the first question, we plot the angle and $L_2$ distance between the iterate and the ground truth in Figure 1. From the figure, it is evident that the large noise scale training brings the iterate near the ground truth $\mu^*$ and then, training with smaller noise scale reduces the $L_2$ distance quickly. In contrast, only small noise scale training does not make any progress. For the second question, even for large noise scale $t$, we show that the angle between the learned mean and true mean is decreasing.

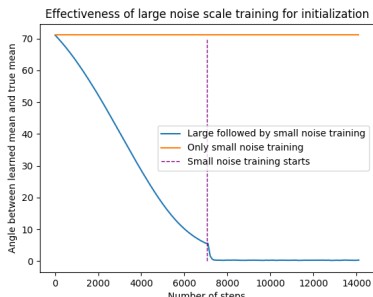
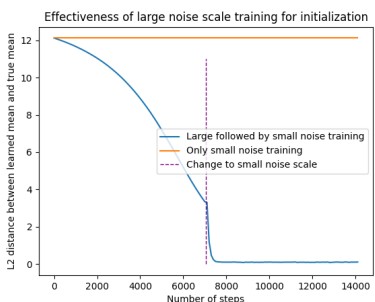

(a) Angle between the iterate and the ground truth.   (b) $L_2$ distance between the iterate and the ground truth.

Figure 1: For the blue curve, we initialize randomly, first train in the large $t$ regime for 7000 steps, and then train in the small $t$ regime for 7000 steps. For the orange curve, we initialize randomly and only train in the small $t$ regime for 14000 steps. We see that large $t$ training helps get in a neighborhood of the ground truth, at which point small $t$ training decreases $L_2$ distance much more quickly, as our theory predicts. In contrast, if we only train with small $t$, we do not make any noticeable progress.

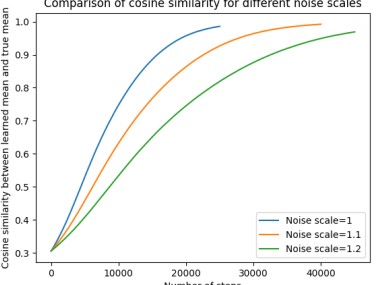
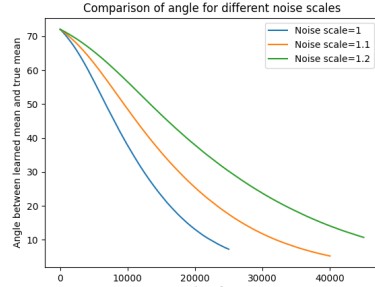

(a) Cosine similarity between the iterate and the ground    (b) Angle between the iterate and the ground truth.
truth.

Figure 2: We show that for some large noise scale, the cosine similarity of learned mean and true mean is increasing (or equivalently, angle is decreasing) as we run for more steps.