# OpenReview forum: "Learning Mixtures of Gaussians Using the DDPM Objective"
_NeurIPS.cc/2023/Conference — NeurIPS 2023 poster_

### Official Review · Reviewer_6LCQ · 2023-06-26

**Soundness:** 1 poor
**Presentation:** 3 good
**Contribution:** 2 fair
**Rating:** 3
**Confidence:** 2

**Summary:**

The authors propose to leverage the DDPM objective to learn Mixtures of Gaussians and prove that gradient descent on the DDPM objective can efficiently recover the ground truth parameters of the mixture model under certain assumptions.

**Strengths:**

Several interesting insights are revealed, such as those associated with Eq. (5) and those related to large/low noise levels.

**Weaknesses:**

(1) The novelties of the proposed method over existing works are not clearly stated. For example, in Line 52, it's stated that ``EM achieves the quantitative guarantees of Theorems 1 and 2 Daskalakis et al. (2017); Xu et al. (2016);...'' So what are the advantages of using the DDPM objective?

(2) It's not clear how practical the proposed method is for general Mixtures of Gaussians with $K\ge 3$ components, where a global optimum is generally impossible for maximizing the loglikelihood [1]. Note that a contradiction between the presented paper and [1] might emerge here, because minimizing the DDPM objective is equivalent to maximizing a hierarchically constructed lower bound of the loglikelihood [2].

(3) No empirical experiments are given.

[1] Chi Jin, Yuchen Zhang, Sivaraman Balakrishnan, Martin J Wainwright, and Michael I Jordan. Local maxima in the likelihood of Gaussian mixture models: Structural results and algorithmic consequences. Advances in neural information processing systems, 29, 2016.

[2] Calvin Luo. Understanding Diffusion Models: A Unified Perspective. Arxiv, 2022.

**Questions:**

Please address the questions in the above ``Weaknesses.''

Minor.

(1) "poweri" in Line 156 is a typo.

(2) "Gaussian"  in Line 214 is a typo.

---

> ### Author Rebuttal · Authors · 2023-08-10
>
> We thank the reviewer for their comments and address them point-by-point. We believe that the criticisms stem from misunderstandings about the scope of what is known in this literature and its relation to our work. We have also provided numerical experiments in the rebuttal, though we also clarify below a broader point about the role of experiments in theoretical works. Given that the stated concerns can all be addressed, we strongly urge the reviewer to reconsider their score.
>
> **EM vs DDPM:**
> - Our work already points to one sense in which DDPMs outperforms EM. The latter is known to suffer from poor global convergence, and provable guarantees for $K > 2$ require a warm start. In contrast, our theory suggests that the different noise regimes of DDPM inherently allow for a two-stage algorithm in which the high-noise regime leads to a spectral initialization close to the true parameters, and the low-noise regime mimics EM from this warm start. **Furthermore, we see this advantage borne out in the numerical experiments performed for the rebuttal.**
> - That said, note that the question of what advantages DDPM affords over EM is very much outside of the scope of our work. We re-emphasize that prior to our work, there were *no known end-to-end* provable guarantees for using the DDPM objective for distribution learning, and one of the key conceptual contributions in our paper is to establish a formal connection between minimizing the DDPM objective and running EM. This connection is highly nontrivial, and given the significant attention that EM has received in the last five decades and the significant recent attention that DDPMs have received, we believe this will be of strong interest to the statistics and generative modeling theory communities moving forward.
>
> **Maximizing log likelihood:**
> - We believe the reviewer may be somewhat confused. The result of Jin et al. says that 1) there are bad local maxima for the log likelihood, and 2) EM with *random* initialization will converge to bad critical points with high probability for $K > 2$ components. 1) is irrelevant to our paper, and 2) does not contradict anything in our paper, because in the $K > 2$ case, we only consider EM initialized in a neighborhood of the true parameters. In fact, as discussed above, we expect that a more sophisticated analysis of DDPMs in which we combine both high- and low-noise regimes will allow us to obtain a (non-EM-based) initialization close to the true parameters via the connection to power method in the high-noise regime, and then use EM-like updates in the low-noise regime to refine our estimate.
>
> - In any case, whether or not DDPMs exhibit global convergence for $K > 2$ is well outside the scope of our paper: we are giving the *first ever* provable guarantees for DDPMs for Gaussian mixtures. In contrast, it is a major open problem in the distribution learning literature to show that *any* algorithm not based on method of moments can learn for $K > 2$ without a warm start.
>
> **Experiments:**
>
> - We have performed a set of numerical experiments to validate our theory, as part of the rebuttal. Specifically, it shows that the constant factors in our analysis are benign so that our convergence guarantees are practical, and it demonstrates the utility of training using the DDPM objective in both high- and low-noise regimes.
> - That being said, we would like to make a broader point. Please note that this is a theory paper. The main contributions are a set of mathematical theorems.  Experiments are orthogonal to the thrust of our work, and indeed orthogonal to the thrust of recent *theoretical* developments for diffusion models. The field of diffusion generative modeling is inundated with experimental works, and it is well-known that the algorithms used in this field are empirically successful. In the context of our work, there is not much to validate experimentally that would add to the empirical literature.
> - In contrast, our theoretical understanding of diffusion models is seriously lacking, apart from a handful of works written over the last year which we have cited in Section 1.1. The whole point of our work is to give a rigorous theoretical justification for why these existing algorithms are effective.

---

### Official Review · Reviewer_NKbJ · 2023-06-28

**Soundness:** 3 good
**Presentation:** 3 good
**Contribution:** 3 good
**Rating:** 7
**Confidence:** 4

**Summary:**

This paper shows that the diffusion model can be used to learn mixtures of Gaussians. In particular, they show that GD on the DDPM objective can efficiently recover the ground truth parameters for both mixtures of two spherical Gaussians and mixtures of $K$ spherical Gaussians under different assumptions. The key ingredient in their analysis is the observation that there is an inherent connection between score-based models and EM, spectral methods.

**Strengths:**

This is a very interesting paper and can be served as a starting point to further explore the power of DDPM from the optimization perspective. Firstly, mixtures of Gaussians are one of the most important and fundamental distribution families of interest, and showing DDPM works for mixtures of Gaussians can be very strong support for its solidness. On the other hand, different from prior work, this work does not require the existence of an oracle for score estimation. Instead, they show that empirical estimation suffices. Lastly, the connection between DDPM and the classic algorithms in this field, i.e., EM and spectral methods can be of independent interest. This might be used to show the effectiveness of DDPM.

**Weaknesses:**

My main concern is that the model and algorithm are a little bit artificial.

- First, the model used for mixtures of two Gaussians is $s_{\theta_t}(x)=\tanh \left(\mu_t^{\top} x\right) \mu_t-x$. Here $-x$ can be understood as the residual connection and $\tanh(\cdot)$ is also a reasonable activation. However, using the same weight $\mu_t$ for both layers is a little bit restrictive. A far more general model is $s_{\theta_t}(x)=\tanh \left(u_t^{\top} x\right) v_t-x$. My conjecture is that it is not hard to extend the current result to this "asymmetric" version. The reason is that there might exist some kind of balanceness between $u_t$ and $v_t$ so that they are roughly the same.

- Second, the authors consider two noise regimes: first they use a large noise and run several iterations of GD, then they switch to the small noise regime. This mimics the practical noise regime. However, in practice,  people decrease the noise level in a continuous manner, like exponentially decaying. Is it possible to adapt the current analysis to a more realistic noise-decaying regime?

Some technical issues:
- When claiming high noise regime, only the upper bound is provided, see line 252. I think there should be also a lower bound for $t_1$.

Last question: The result in this paper only show that DDPM is as good as EM and spectral methods. A far more important question is why empirically DDPM outperforms these methods. Can DDPM outperforms the existing methods for the mixtures of Gaussians in some aspects like computational time?

**Questions:**

Please see the weakness part.

**Limitations:**

Please see the weakness part.

---

> ### Author Rebuttal · Authors · 2023-08-10
>
> We thank the reviewer for the encouraging words and for finding our paper very interesting. Here we address the questions raised, one of our main points being that while the points raised are indeed great directions for future work, we would like to underscore that prior to our work, there were no provable score estimation results in the theoretical literature on distribution learning. We thus encourage the reviewer to reconsider their score and the significance of our contributions in this context.
>
> **Decoupling the two layers' weight in the score function**
>
> It’s an interesting question. It should not be hard to show some kind of layerwise training guarantee at least: if the output layer weights are fixed at random initialization and the input layer weights are trained, one should be able to show that the input layer weights converge to the ground truth because of similar power iteration / EM - like considerations, after which if we train the output layer weights, then this is simply a linear problem and we will likewise converge to the ground truth. Extending this beyond layerwise training will require more work. Nevertheless, we find that the novelty of showing that *any* gradient-based algorithm can be used for provable score estimation in a nontrivial high-dimensional distribution learning context perhaps outweighs the strengths of specific architectural choices, unless if the architecture is chosen to be much more generic, e.g. a general feedforward O(1)-layer network. This would, however, be outside the scope of our work.
>
> **Continuously decaying the noise**
>
> The reviewer is correct that our two-stage algorithm is a little different from how DDPMs are trained in practice, but the objective used in practice is actually a little different from what the reviewer describes, and we feel that understanding the former, while of great interest, is also well out of the scope of what is currently possible in the literature on provable score estimation.
>
> For context, in diffusion models, one trains a score network $s(x_t,t)$ which accepts as input (an embedding of) a noise level parameter $t$ and an image $x_t$ at noise level $t$, and outputs a denoised image. To sample an image, one then starts with a fully noisy, e.g. Gaussian, seed and runs a discretization of a suitable reverse SDE/ODE that iteratively applies the score network along some noise schedule as the reviewer suggests.
>
> We caution that this should *not* be confused with the following: train the score network by running GD with respect to the DDPM objective at noise level t_1, then with respect to the DPPM objective at noise level t_2, etc.
>
> Instead, how one trains $s(x_t,t)$ is by running GD on a DDPM objective which is averaged across DDPM objectives at different noise levels. See e.g. Eq. (8) in https://arxiv.org/pdf/2206.00364.pdf. There is flexibility in how to choose the distribution D over noise levels. Perhaps what the reviewer had in mind is that D in practice is a continuous distribution over noise levels, whereas our analysis, as written, pertains to D which is discrete. We agree that the former is a very interesting and practically relevant direction for study. It would however require first defining a suitable architecture which incorporates a time embedding. This already takes us quite far from the domain of settings where we can hope for an end-to-end analysis of gradient-based methods, and given that nothing was known about provable score estimation for distribution learning prior to our work, we think it is fair to regard this as a topic much further down the line in terms of things that we can currently analyze.
>
> **Lower bound for $t_1$**
>
> Yes, this is a typo that we will fix in the revision. The noise scale t_1 should be $\Theta (log d)$. The lower bound on noise scale $t_1$ is used for obtaining equivalence between the power method and Gradient Descent on the diffusion model (See Lemma B.3) because the approximation error between gradient descent on the DDPM objective and power method depends polynomially on \norm{\mu_t} and \norm{\mu_t^*}. Additionally, the parameters at a noise scale t is $\mu_t = \mu \exp(-t)$ and when t is $\Omega (\log d)$, then $\| \mu_t \|$ is $O(1 / \mathrm{poly}(d))$. Similarly, we also obtain $\|\mu_t^*\|$ to be $O(1 / \mathrm{poly}(d))$ for $t = \Omega (\log d)$. Therefore, the approximation error between the power method and Gradient Descent on the DDPM objective becomes small $O(1/poly(d))$.
>
> **Outperforming other methods:**
>
> While we think it is already a compelling contribution to even be able to analyze gradient descent on the DDPM objective, we stress that one of the key takeaways of our work is precisely that our algorithm gets the best of both worlds of power method and EM. Note that the main limitation of EM is that it does requires a warm initialization in order to converge, and the main limitation of power method is that while it can converge from random initialization, it does not converge very quickly even when it is in a neighborhood of the ground truth. In contrast, we have shown that by varying the noise level with which one defines the DDPM objective, we get the advantages of both algorithms: 1) the large noise regime gives us the global convergence properties of power method, and 2) the small noise regime gives us the fast local convergence properties of EM. In fact we see precisely this phenomenon in our second set of experiments.

---

> > ### Comment · Reviewer_NKbJ · 2023-08-19
> > **Thanks for your reply**
> >
> > Thanks for your wonderful response, which addresses all of my concerns! To encourage your effort on this interesting direction. I will increase my score from 6 to 7.

---

### Official Review · Reviewer_R4kZ · 2023-07-03

**Soundness:** 2 fair
**Presentation:** 1 poor
**Contribution:** 2 fair
**Rating:** 3
**Confidence:** 3

**Summary:**

This paper presents a new approach to learning Gaussian mixture models using the denoising diffusion probabilistic model (DDPM) objective. The authors provide a 2-part algorithm that allows one to reconstruct the parameters of a mixture of Gaussians. The first part of the algorithm uses gradient descent with "high nose" scale of $t$ and random initialization to learn parameter, correlated to the true ones, while the second step refines these parameters, using those obtained in the previous step as a "warm start" and using the small $t$ mode.

**Strengths:**

The paper gives the relation between gradient descent with loss from denoising diffusion probabilistic model and a) power iteration in case of large $t$ and b) Expectation-Maximization algorithm in case of not large $t$.

Some number of theorems and lemmas are proved.

**Weaknesses:**

1. The presentation of the results is weak. The article is poorly structured, with no "conclusions" or "discussion" section. A lot of fuzzy language. Such wording may be acceptable in informal descriptions of results, but even in the section with formal descriptions phrases like "with high probability" are used. The main text of the article in the main pdf file differs from the version in the file with the Appendix.

2. The matrix $(1-\eta r)I +2\eta \mu^*_t {\mu^*_t}^T$ has eigenvalues close to one. The largest of them (for which the eigenvector in power iteration (6) is sought) differs only by an additional term proportional to $e^{-2t}\eta$. This term is very small, since by definition $t$ in this case is large, and the learning rate $\eta$ is usually chosen very small ($\sim10^{-2}-10^{-4}$). In such a situation power iteration can converge to an arbitrary vector. Moreover, as expression (6) is an approximation, this convergence is even more uncertain.

3. There are no numerical experiments, even for the simplest cases. However, the real convergence of power iterations with the matrix (7) is not obvious.

 4. There is no separate  comparison with other, including state-of-the-art, approaches to recovering Gaussian mixture parameters. Although paper has rather large literature review section.


Minor:

line 156 typo: "poweri iteration"

"EM" abbreviation is deciphered only on page 6, but widely used starting from the abstract.




**Questions:**

1. Have you performed numerical experiments to check the convergence of the method and compare it with other approaches?

2. In reality, for some model numerical parameter values, how many iterations are needed for power iteration described in (6)--(7) to converge?

3. line 246: (Sec. 2.1 with Formal statements) What is "high probability"?


Minor

What authors mean by "well-separated" (line 131)

What authors mean by _student_ network (line 142)

**Limitations:**

1. Only rough asymptotic estimates are given. Real constants, which can be in practical experiments are not given, however, they can be large.
2. The proof for the case of large $t$ (the relation of DDPM to power iteration) is based on an approximate, not exact, formula (6) and thus it is not conclusive.

---

> ### Author Rebuttal · Authors · 2023-08-10
>
> We thank the reviewer for the feedback and we address all the points raised by the reviewer. As the main weakness mentioned by the reviewer on the convergence of power method seems to be due to some misunderstanding and is thoroughly addressed both through theory and experiments that we performed for the rebuttal, we ask the reviewer to reconsider their score.
>
> - **Fuzzy language with informal phrases like "with high probability":** It is completely standard in the literature of learning theory to use “with high probability” to denote “with failure probability inverse polynomial in d” (for example, see Daskalakis et al. 2017, Diakonikolas et al. 2020, Liu-Li ‘22). The reviewer has not provided any other instance of “fuzzy” language, and we disagree that aspects of the paper are “fuzzy”. We will also make sure to include a conclusions section, but apart from this, we respectfully disagree that the presentation of the results is “weak” or “poorly structured.”
> - **power iteration can converge to an arbitrary vector** We strongly disagree that the convergence is uncertain. If a matrix’s top eigenvalue is larger than the others by some multiplicative $1+\tau$, then in $O(1/\tau)$ power iterations we will converge to the top eigenvector (see e.g. https://en.wikipedia.org/wiki/Power_iteration). So even if $\tau$ is small, as long as $\tau$ is inverse polynomial in the relevant parameters (which is the case for our $\tau = e^{-2t}\eta$, as $t$ is only logarithmic and $\eta$ is inverse polynomial), the complexity of power iteration is still polynomial. Furthermore, in Lemma 9 (page 7) and in Lemma B.3, we make completely precise the approximation error in (6) and rigorously show that this error is sufficiently low order that it does not affect convergence. We, therefore, encourage the reviewer to seriously reconsider their view that our proof is “inconclusive.”
> - **Numerical experiments and numerical convergence of power iterations:**
>    - We have performed a set of numerical experiments to validate our theory, as part of the rebuttal. Specifically, it shows that the constant factors in our analysis are benign so that our convergence guarantees are practical, and it demonstrates the utility of training using the DDPM objective in both high- and low-noise regimes.
>    - That being said, we would like to make a broader point. Please note that this is a theory paper. The main contributions are a set of mathematical theorems.  Experiments are orthogonal to the thrust of our work, and indeed orthogonal to the thrust of recent *theoretical* developments for diffusion models. The field of diffusion generative modeling is inundated with experimental works, and it is well-known that the algorithms used in this field are empirically successful. In the context of our work, there is not much to validate experimentally that would add to the empirical literature. In contrast, our theoretical understanding of diffusion models is seriously lacking, apart from a handful of works written over the last year which we have cited in Section 1.1. The whole point of our work is to give a rigorous theoretical justification for why these existing algorithms are effective.
> - **Comparison of results with other state-of-the-art approaches:** We make a comparison to the most relevant state-of-the-art approaches in the introduction section in Lines 52-55 and in the Related Work section in Lines 104-105. We also state the state-of-the-art EM guarantee of Segol and Nadler which clearly matches our guarantee, though we will add some words in the revision making this clear. In the last paragraph of Related Work, we also mention works on “general mixing weights and covariances” which is clearly more general than our setting. Again, we will add some words in the revision to make this clear.
> - **Asymptotic analysis and constant factors:** Please recall this is a theory paper where it is standard to use asymptotic analysis and suppress constant factors. Even the prior theoretical works on the diffusion models use asymptotic analysis (e.g., Lee et al. 2023, Chen et al. 2023a, b). And as the experiments we performed for the rebuttal make clear, the constant factors are benign.
>
> **Minor comments:**
> - By well-separated we mean that the means are separated by some absolute constant. We also handle the case of inverse polynomial separation.
> - Student network refers to the network that we train. This “student-teacher” terminology is standard in the deep learning theory literature, where the “teacher” is the ground truth network generating the examples (in our case, it is the true score function), and the “student” is the network whose parameters are optimized via gradient descent to minimize some training objective.

---

> > ### Comment · Reviewer_R4kZ · 2023-08-16
> >
> > I thank the authors for their answers and for additional experiments.
> >
> >
> > **1. About the style of the paper.**
> > The article is missing sections such as "Background", "Problem Statement", "Conclusion", "Discussions", comparison with other algorithms, etc. Of course, they don't all have to be at the same time. But the presented structure of the article is extremely difficult to understand the main ideas (which are quite simple). To correct the structure, to introduce correct definitions, to correct holes in the proofs, etc., requires substantial revisions that go far beyond the minor revisions allowed at this stage of reviewing the paper.
> >
> >
> > **2 About the convergence of power iteration:**
> >
> > > (see e.g. https://en.wikipedia.org/wiki/Power_iteration).
> >
> > a. The Wikipedia article you quoted says
> >
> > > If we assume $A$ has an eigenvalue that is strictly greater in magnitude than its other eigenvalues and the starting vector $b_0$ has a nonzero component in the direction of an eigenvector associated with the dominant eigenvalue, then a subsequence $(b_k)$ converges to an eigenvector associated with the dominant eigenvalue.
> > Without the two assumptions above, the sequence $(b_k)$ does not necessarily converge.
> >
> >
> > Your paper is positioned as a theoretical paper, but you have not checked these two conditions given, which are necessary to prove the convergence of power iteration.  The check of the second condition (about the non-zero component) can be omitted, since it is always possible to take several random vectors as an initial approximation.  But the condition that the first eigenvalue must be much larger than the others is crucial.
> >
> > The argument about asymptotics here cannot replace a rigorous study, since the numerical coefficients in this asymptotics are not known in advance, and it may turn out that the first eigenvalue is not sufficiently different from the others for convergence of power iteration.
> >
> > > If a matrix’s top eigenvalue is larger than the others by some multiplicative $1 + \tau$, then in power iterations we will converge to the top eigenvector
> >
> > b. Moreover, you don't just have $1 + \tau$. You have $1 + \tau + \epsilon$, where $\epsilon$  appears due to approximate equalities (6)--(7).  Is it possible that $\epsilon < 0$, and furthermore that $1 + \tau + \epsilon<1$? Theoretically, it is possible, and in this situation the convergence of power iteration will be to an arbitrary vector. The authors do not prove that this situation is excluded as there is no explicit comparison of these two values in the __main text__ of the paper. (Lemma 9, which turned into Lemma 8 in supplementary file, contains only the expression $poly(\frac1d)$ in the right-hand side, which can be arbitrary large when $d$ is fixed.)
> >
> >
> > **Minor:**
> > >Please note that this is a theory paper. The main contributions are a set of mathematical theorems. Experiments are orthogonal to the thrust of our work, and indeed orthogonal to the thrust of recent theoretical developments for diffusion models.
> >
> > Although the paper is positioned as theoretical, it describes a practical algorithm. If it turns out that in practice the number of iterations of such an algorithm will be equal to $10^9$ at real parameter values, but it will be "asymptotically small", the practical value of such an algorithm is insignificant
> >
> >
> > >By well-separated we mean that the means are separated by some absolute constant.
> >
> > It's still not clear how "well-separated" differs from just "separated" (or "poor separated", etc.).
> >
> >
> > > Student network refers to the network that we train. This “student-teacher” terminology is standard in the deep learning theory literature
> >
> > The concept of "student network" is connected with the concept of "teacher network". You answered that you do not have a teacher network as a network. Moreover, the words "teacher network" do not even appear in the paper. Thus, the concept of "student network" is misleading.
> >
> > **Conclusion.**
> > I think, that each of my two arguments above (about the structure of the paper and about the lack of justification for the convergence of power iteration) is separately a reason to reject the paper.

---

> > > ### Author Response · Authors · 2023-08-19
> > > **Response to Reviewer's Comment on the Rebuttal**
> > >
> > > > you have not checked these two conditions given
> > >
> > > The first condition directly follows from the eigenvalues of the matrix $Id (1 - 3 \eta || \mu_t ||^2 ) + 2 \eta \mu_t^* \mu_t^{\ast T}$ and is written in Lemma B.5. The first eigenvalue (corresponding to $\mu_t^*$) is $(1 - 3 \eta || \mu_t ||^2 ) + 2 \eta || \mu_t^* ||^2$, and the rest of the eigenvalues are $(1 - 3 \eta || \mu_t ||^2 )$. The second condition (written in Lemma B.4) is a direct consequence of the anti-concentration of the Gaussian random variable and does not require taking several random vectors as an initial approximation as mentioned by the reviewer.
> > >
> > >
> > > > Is it possible that $\epsilon < 0$, and furthermore that $1 + \tau + \epsilon < 0$? Theoretically, it is possible, and in this situation the convergence of power iteration will be to an arbitrary vector.
> > >
> > > No, $\epsilon$ is the approximation error when comparing gradient descent on the DDPM objective to the power method and it cannot be negative. We have also proved in Lemma B.5 and Lemma B.6 that the approximation error is sufficiently small such that it does not disturb the convergence of the power method. It is simply incorrect to say we do not carefully keep track of these issues.
> > >
> > > > contains only the expression $poly(\frac{1}{d})$ in the right-hand side, which can be arbitrarily large when $d$ is fixed.
> > >
> > > The parameter $d$ is never fixed.  For decades, $d$ has been the key asymptotic parameter of interest when learning mixtures of Gaussians. For example, if the failure probability of an event is $O(\frac{1}{d})$, then the failure probability of that event is treated as small (See cited literature Daskalakis et al. 2017, Diakonikolas et al. 2020, Kwon et al. 2020, Liu-Li ‘22). The reason is that in the regime where $d$ is a fixed constant, all of the learning problems considered are trivial: in this case because $d = O(1)$, brute-force enumeration over an epsilon-net suffices.
> > >
> > > > It's still not clear how ‘well-separated’ differs from just ‘separated’.
> > >
> > > We consider two regimes of separation in the paper for mixtures of two Gaussians. 1) the centers are separated by absolute constant or 2) the centers are separated by $poly(1/d)$. By well-separated centers, we mean the centers separated by an absolute constant and by poorly-separated centers, we mean the other case.
> > >
> > >
> > >
> > > To summarize, we strongly disagree with the reviewer’s repeated claims that there are “holes in the proof”, “the proof is based on an approximate formula” and “is not conclusive”. We have answered all the questions raised by the reviewers and are happy to discuss in further detail.

---

> > > > ### Comment · Reviewer_R4kZ · 2023-08-20
> > > >
> > > > > No, $\epsilon$ is the approximation error when comparing gradient descent on the DDPM objective to the power method and it cannot be negative.
> > > >
> > > > This doesn't follow from anywhere in the text of the article. I'm not sure it is true. There is no clear justification for this fact.
> > > >
> > > > >We have also proved in Lemma B.5 and Lemma B.6
> > > >
> > > > There are not even references to these Lemmas in the text of the main article. It is difficult for the reader to understand which lemma in the extensive appendix to look for in order to prove this or that fact.
> > > >
> > > > >  we strongly disagree with the reviewer’s repeated claims that there are “holes in the proof”
> > > >
> > > > By holes in the proof, I meant that, for example, the inequality under discussion $1+\tau+\epsilon > 1$ is not explicitly stated or proved in the text of the paper (which is necessary to prove convergence of power iteration). I admit that it is true, but only asymptotically, i.e. at some combination of parameters $d$, $t$ etc. The specific conditions under which this inequality is true are not specified. Thus, this paper does not meet the criteria of rigor required for theoretical papers and requires additional research.

---

> > > > > ### Author Response · Authors · 2023-08-21
> > > > >
> > > > > > I'm not sure it is true. There is no clear justification for this fact:
> > > > >
> > > > > The reviewer appears to be worried about whether the power iteration matrix with eigengap $\tau$ still has an eigengap after we replace it with an approximate version. This is definitely not an issue in our proof because we prove that the approximation error is always significantly smaller than the eigengap when $t=O(\log d)$ (See lines 576 and 583 of Lemma B.5).
> > > > >
> > > > > > the inequality under discussion $1 + \tau + \epsilon > 1$ is not explicitly stated….
> > > > >
> > > > > The existence of the eigengap of the power iteration matrix with the approximation can not be made precise by just proving $1 + \tau + \epsilon > 1$ because $1 + \tau$ is the *ratio* of first and second eigenvalues and on high level, both eigenvalues will be replaced with an approximate version. The actual steps needed to make precise that the eigengap is preserved are exactly made precise in lines 576 and 583 of Lemma B.5.
> > > > >
> > > > > > at some combination of parameters $d,t$ etc. The specific conditions under which this inequality is true are not specified.
> > > > >
> > > > > The condition under which the approximate power iteration has eigengap is $t = O(\log d)$ because when $t = O(\log d)$, the approximation error is significantly smaller than the eigengap and it has been repeatedly mentioned in section 2.2 (Part 1). If instead the reviewer is concerned that we have not stated the exact constant factors in our big-O notation, we do not agree that this equates to lack of rigor as this asymptotic language is the standard practice in learning theory and theoretical computer science more broadly. If the reviewer’s concern with big-O is instead purely practical, note that our experiments demonstrate that the constant factors in our proofs are not an issue.
> > > > >
> > > > >
> > > > > In summary, we believe that we have addressed all of the primary limitations and weaknesses mentioned by the reviewer in the original review (e.g., no experiments, the proof for the case of large $t$ is based on an approximate formula eq.(6)). We thus respectfully urge the reviewer to reconsider the original score.

---

### Official Review · Reviewer_Egk5 · 2023-07-06

**Soundness:** 3 good
**Presentation:** 1 poor
**Contribution:** 3 good
**Rating:** 3
**Confidence:** 2

**Summary:**

Proofs are given for showing that the true mean parameters of Gaussian mixture models (GMMs) with identity covariance matrices can be recovered when using gradient descent to optimize the DDPM objective. It is argued that it is not well understood if score-based models can provably estimate the parameters of the data distributions, which is the objective of this submission. The findings and results are connected to the EM algorithm.

**Strengths:**

The paper attempts contribute with fundamental understanding of diffusion models, focusing on an important class data distributions, GMMs. Diffusion models are clearly of high interest to the NeurIPS community and theoretical results of the type provided in the submission could be significant, or of interest to researchers in the field.

The literature review is thorough and it appears that the topic of the paper is original, in the sense that the theoretical results do not exist in the literature. It is good that the results are connected to other well-studied algorithms, like the EM algorithm.

The clarity of the purpose of the submission benefits from the bullet list in the abstract.

**Weaknesses:**

The structure of the submission is unfortunately not very strong, which makes the arguments in the submission tedious to follow. I am missing sufficient background to the power method and the EM algorithm, a background of the former is not provided, and there is no conclusion section or discussion. Such sections would help the reader to understand the contributions of the submission in terms of the lingo and terminology introduced in the paper. Also, it would give the paper a more concise ending, as it now ends with a proof sketch.

My score of the paper is mainly based on the structure and quality (presentation) of the paper, which I believe is below acceptable for a NeurIPS paper. However, my confidence level is relatively low as theoretical results for diffusion/score-based models is slightly outside my area of expertise. That is, am not sure how significant the results in the submission are--hopefully other reviewers can aid to evaluate this part. Nonetheless, as it stands, my opinion is that the inadequate presentation of the work outweighs (my understanding of) the significance of the results in the presented work.

To help the authors in the revision of the paper, I below enumerate some suggestions on how to improve the structure and some of my concerns. I hope they are helpful.

* GD is frequently used to denote gradient descent. But it is not defined as an acronym in the abstract nor in the paper. I inferred the definition of GD only after reading quite a bit of the paper. This confused me, as I did not know what GD referred to. I suggest introducing the acronym in the abstract and write gradient descent (GD) in line 35 where GD is first used.

* In line 35, what is the meaning of a "natural" data distribution? I could find an explanation of it and I could not derive what it ought to specify. Is "natural" an important distinction?

* Sentences like "we use this as our student network architecture when running gradient descent." (line 146) gives me expectation that some experiments have been carried out where these architectures have been used. As there are no experiments in the paper, this confused me. A potential reformulation could be "one could use" as you have in fact not used it. Similar confusion arouse at line 162 "we run projected gradient descent".

* Related to the previous bullet, why not run experiments to validate the results on synthetic data?

* Is the heading in line 164 supposed to be a paragraph or a section? Now it is an empty paragraph.

* I am not familiar with the power method and no background or explanation of it is given in the paper. I think this needs to be included to make the paper more stand alone.

* Projected gradient descent is introduced after the background section without proper explanation. Is projected gradient descent typically used to train diffusion models?

* The EM algorithm is central to the findings in the paper. As such, I think it should be more carefully explained. Especially, including a Background section earlier than Sec 1.3 would strengthen the paper and make it easier to follow the arguments made in the earlier sections. For instance, "We show that the gradient descent in this "small noise" regime corresponds to the EM algorithm" in line 158 would become easier to understand.

* In the setting where $K=2$, is there an additional, consistent assumption that the means of the Gaussian are centered around the origin? I.e. the means are $-\mu^*$ and $\mu^*$? I read the section above Eq. (2) as this being an example of a GMM "for the sake of intuition". But then this parameter setting is specified again in Eq. (13). Does this mean that the results in the paper are constrained to Gaussians centered around the origin? If so, this should be specified in the relating theorems? I.e. Thm 1 and 2 and their formal counterparts.

**Minor issues**
* line 46: the flow if the sentence could be improved by using punctuation after "$d$, the dimension."
* The acronym EM is most often used without definition. Although this is more clear what it refers to, "expectation-maximization" is spelled out in other parts of the paper. I suggest introducing the definition and sticking to it throughout the paper.

**Questions:**

* In Sec. 1.2, it would be nice to see a discussion or explanation of why the covariance matrices Id and not $\sigma^2$I_d. That is, the variance elements would not necessarily equal to one, but the Gaussians would still be spherical. Would your results still hold for this setting? Otherwise, shouldn't the setting itemized in the abstract specify that your results apply to unit-spherical Gaussians?

**Limitations:**

I had some questions on the assumptions of the parameters of the Gaussians in the GMMs. Otherwise I believe the limitations were addressed.

---

> ### Author Rebuttal · Authors · 2023-08-10
>
> We thank the reviewer for their helpful suggestions. We will make sure to include a discussion/conclusion section in the revision. Here we address their other points one-by-one. Given that it is straightforward to fix all of these concerns in a single round of cosmetic edits, and given that the only non-expository weakness mentioned stems from a misunderstanding about the generality of the origin-centering assumption, we don’t think that such a low score is appropriate and respectfully urge the reviewer to reconsider.
>
> **Weaknesses:**
>
> - *Confusion on GD:* In the updated draft, we will clarify that GD stands for gradient descent.
> - *The meaning of a "natural" data distribution:* The question in line 35 is intended to be informal, and in ML theory, “natural” is a commonly used term to signify a generative assumption which is “simple” and “non-contrived.” Given that Gaussian mixtures are one of the most studied distributions in the literature for provable algorithms for distribution learning, “natural” is a reasonable term in this context.
> - *Expectations about the experiments on the student network:* We are happy to incorporate the reviewer’s suggestions to avoid this confusion, but note that in theoretical works, it is standard to say “we perform algorithm X to solve task Y” without the literal implication that experiments will be done. For instance, if a paper says “we will maintain the following data structure while processing the edges of graph G,” it need not mean that they will carry out an experiment in which they implement said data structure.
> - *Experiments*: see below for a summary of the experiments we performed for the rebuttal, as well as a separate discussion on the role of experiments in theoretical works
> - *Is the heading in line 164 a paragraph or a section?:* It is a paragraph as intended. We will reformat the bolded subheaders beneath it as underlined or italicized subheaders to minimize confusion.
> - *Background on power method:* The power method is taught in introductory algorithms and linear algebra classes. We will add a paragraph reminding the reader of the simple intuition (to compute the top eigenvector of a matrix, repeatedly multiply a vector by the matrix, and the result will converge in angular distance to the top eigenvector).
> - *Background on projected gradient descent:* It is very common in deep learning to optimize with an L1/L2 regularizer on parameters, which implicitly plays the role, in a sense that can be made formal, of a projection step. Given that regularized gradient descent is thus a standard algorithm not just in diffusion modeling but in machine learning more broadly, and furthermore given that we explicitly state in Line 162 the set that we are projecting to, it is unclear what other aspects of projected gradient descent should be explained at that point in the text.
> - *Additional background on EM algorithm:* EM for Gaussian mixtures is taught in introductory machine learning at the undergraduate level. We will add some additional details to the paragraph starting in Line 58. Namely, we will include a centered equation reminding the reader of the form of the “M” step in the EM algorithm, and then elaborate on what we mean when we say that the gradient updates when minimizing the DDPM objective are performing this M step.
> - *Origin-centered mixtures of Gaussian:* We refer the reviewer to Eq. (13) where we make this assumption clear (When we said “For the sake of intuition” in Section 1.2, we meant that we focus on *K = 2* for the sake of intuition, not that we focus on origin-centered means). We stress that the origin-centered mean assumption is not a “constraint”, it is actually without loss of generality as mentioned in Line 192, 193 (just after Eq.(13)). One can always estimate the overall mean of the data distribution to high accuracy from samples and then recenter the dataset accordingly; one can check that the steps of our proof are robust to the error in estimating this mean.
>
> **Minor issues:**
>
> - We will use the suggested punctuation
> - We will make clear early on that EM stands for expectation-maximization
>
> **Questions:**
> - Yes, our results still hold for this setting. The reason is that if you scale the dataset by a factor of 1/sigma, you reduce it to the unit-spherical case. It is however true that if the covariance matrices have different variances across the different mixture components, then our analysis does not yet apply. Note however this should not be viewed as a weakness per se: it is common for works to focus on the equal covariance setting (see e.g. the state-of-the-art works of [Segol-Nadler ‘21] and [Liu-Li ‘22]). We will make this clear in the revision and reword the abstract accordingly.
>
> **Experiments:**
> - We have performed a set of numerical experiments to validate our theory, as part of the rebuttal. Specifically, it shows that the constant factors in our analysis are benign so that our convergence guarantees are practical, and it demonstrates the utility of training using the DDPM objective in both high- and low-noise regimes.
> - That being said, we would like to make a broader point. Please note that this is a theory paper. The main contributions are a set of mathematical theorems.  Experiments are orthogonal to the thrust of our work, and indeed orthogonal to the thrust of recent *theoretical* developments for diffusion models. The field of diffusion generative modeling is inundated with experimental works, and it is well-known that the algorithms used in this field are empirically successful. In the context of our work, there is not much to validate experimentally that would add to the empirical literature.
> In contrast, our theoretical understanding of diffusion models is seriously lacking, apart from a handful of works written over the last year which we have cited in Section 1.1. The whole point of our work is to give a rigorous theoretical justification for why these existing algorithms are effective.

---

> > ### Comment · Reviewer_Egk5 · 2023-08-15
> > **Response to rebuttal**
> >
> > First, I want to thank the authors for their rebuttal, they have clearly worked hard to address my comments. I also acknowledge that I have taken part of the submitted pdf, thanks for the complementary experiments.
> >
> > However, I disagree that the presentation edits are "straightforward" and adding one or two sections cannot be regarded as a "single round of cosmetic edits". To use the terminology of the NeurIPS reviewer guidelines (https://neurips.cc/Conferences/2023/ReviewerGuidelines#:~:text=well%2Dwritten%20summary.-,Strengths%20and%20Weaknesses,-%3A%20Please%20provide)
> >
> > *Quality: Is this a complete piece of work or work in progress?* I still think this is a work in progress, a view I appear to share with the other reviewers.
> >
> > *Clarity: Is the submission clearly written? Is it well organized? Does it adequately inform the reader?* I am sorry to conclude that these criteria are not met. Even if the promised revisions will certainly improve the clarity of the paper, I cannot increase my score to an acceptance level before reading the revised version of the paper with the added section(s). Instead I will increase my score to 4, displaying my appreciation of the efforts put in during the rebuttal, will keeping my low confidence score. I hope the authors will understand my reasoning.
> >
> > I want to be clear that I appreciate the improvements of the submission that have been made during the rebuttal period, and I applaud the authors for their efforts. By incorporating the feedback provided by the reviewers, a resubmission of the work will be competitive for acceptance in a future conference proceeding.

---

### Author Rebuttal · Authors · 2023-08-10

We thank the reviewers for their comments, which we have responded to individually. Here we note that we also performed numerical experiments (see attached pdf) to demonstrate that 1) the **constant factors in our analysis are quite benign**, and 2) as predicted by our theory, **training with the DDPM objective at different noise scales has a tangible benefit over power iteration and EM**, namely it gets the best of both worlds of power iteration and EM: global convergence from random initialization for the former and fast local convergence for the latter.

*In our individual responses to reviews, we also included some broader comments on the role of experiments in theory, especially in the theory of generative modeling where there is a wealth of empirical works and dearth of theoretical understanding.*

---

### Decision · Program_Chairs · 2023-09-21

**Decision:**

Accept (poster)

**Comment:**

The decision on this paper runs contrary to the the majority of the reviewers comments, after a discussion between AC and SAC.

The AC's opinion is that the paper is clearly above the bar for NeurIPS. It shows a very nice connection between the optimization dynamics for DDPM (Denoising Diffusion Probabilistic Models) and EM/power method (in different regimes for the amount of noise), when the underlying data distribution is a mixture of Gaussians in the well-separated regime.

Of course, the well-separated regime of mixture of Gaussians is the "easy" regime for the parameter learning problem, and classical strategies like EM + spectral initialization are long known to work. The purpose of the paper isn't to give new algorithms for this problem. Instead, the goal is to understand training dynamics of popular score-based/diffusion-based objectives.

While there's been a recent flurry of work trying to understanding the required discretization and training accuracy to achieve a good sampling quality (e.g. Block et al '22, Chen et al '23, Lee et al '23), as well as statistical properties of score-based losses (Koehler et al '23, Pabbaraju et al '23, Qin et al '23), training dynamics are challenging to analyze in the presence of non-convexity for any loss, and diffusion/score-based methods are very new territory. As such, the paper will be interesting to a large part of the theory of diffusion models community.